# A minimal self-organisation model of the Golgi apparatus

Quentin Vagne[1†], Jean-Patrick Vrel[2,3†], Pierre Sens[2,3]*

[1]Center for Systems Biology Dresden, Max Planck Institute of Molecular Cell Biology and Genetics, Dresden, Germany; [2]Institut Curie, PSL Research University, CNRS, UMR 168, F-75005, Paris, France; [3]UPMC Univ Paris 06, CNRS, UMR 168, F-75005, Paris, France

**Abstract** The design principles dictating the spatio-temporal organisation of eukaryotic cells, and in particular the mechanisms controlling the self-organisation and dynamics of membrane-bound organelles such as the Golgi apparatus, remain elusive. Although this organelle was discovered 120 years ago, such basic questions as whether vesicular transport through the Golgi occurs in an anterograde (from entry to exit) or retrograde fashion are still strongly debated. Here, we address these issues by studying a quantitative model of organelle dynamics that includes: de-novo compartment generation, inter-compartment vesicular exchange, and biochemical conversion of membrane components. We show that anterograde or retrograde vesicular transports are asymptotic behaviors of a much richer dynamical system. Indeed, the structure and composition of cellular compartments and the directionality of vesicular exchange are intimately linked. They are emergent properties that can be tuned by varying the relative rates of vesicle budding, fusion and biochemical conversion.

**\*For correspondence:**
pierre.sens@curie.fr

[†]These authors contributed equally to this work

## Introduction

The Golgi apparatus is an intracellular organelle at the crossroad of the secretory, lysosomal and endocytic pathways (*Heald and Cohen-Fix, 2014*). One of its most documented functions is the sorting and processing of many proteins synthesized by eukaryotic cells (*Lippincott-Schwartz et al., 2000*). Proteins translated in the endoplasmic reticulum (ER) are addressed to the Golgi, where they undergo post-translational maturation and sorting, before being exported to their final destination. The Golgi itself is composed of distinct sub-compartments, called cisternae, of different biochemical identities (*Stanley, 2011*). From the ER, cargo-proteins successively reside in cisternae of the *cis*, *medial* and *trans*-identities, after which they exit the Golgi via the *trans*-Golgi network (TGN). In some organisms such as the yeast *Saccharomyces cerevisiae* cisternae are dispersed throughout the cell (*Suda and Nakano, 2012*) and each cisterna undergoes maturation from a *cis* to a *trans*-identity (*Losev et al., 2006*; *Matsuura-Tokita et al., 2006*). In most other eukaryotes, cisternae are stacked in a polarized fashion, with cargo-proteins entering via the *cis*-face and exiting via the *trans*-face (*Boncompain and Perez, 2013*). This polarity is robustly conserved over time, despite cisternae constantly exchanging vesicles with each other, the ER and the TGN (*Lippincott-Schwartz et al., 2000*). Pharmacological treatments that alter the structure and stacking of Golgi compartments in mammalian cells affect the processing of certain proteins, and in particular glycosylation (*Hu et al., 2005*; *Shen et al., 2007*; *Xiang et al., 2013*; *Bekier et al., 2017*).

The highly dynamical nature and compact structure of a stacked Golgi makes it difficult to determine how cargo-proteins are transported in an anterograde fashion from the *cis* to *trans*-side of the stack and how this transport is coupled to processing. Several models have been proposed to explain the transport dynamics of Golgi cargo (See sketch in *Figure 1A*). They mostly belong to two classes: one involving stable compartments exchanging components through fusion-based

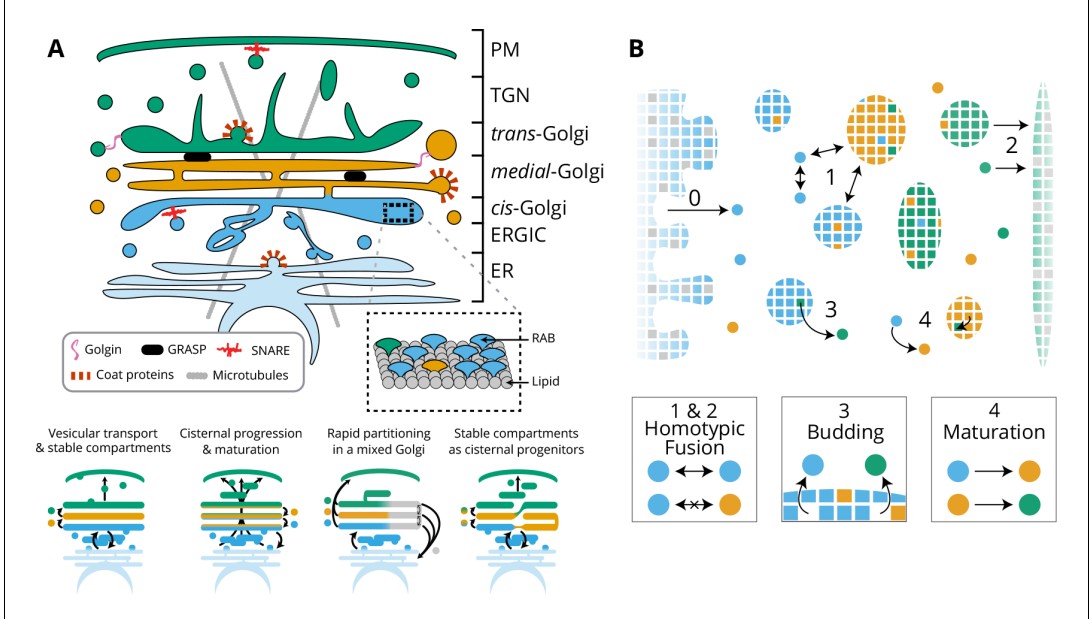

**Figure 1.** Model of golgi structure and transport. (**A**) Top: classical representation of the structure of the Golgi, showing some of the important protein actors. The three main membrane identities are shown in different colors (*cis*: blue, *medial*: orange, *trans*: green): ER = Endoplasmic Reticulum, ERGIC = ER Golgi intermediate Compartment, TGN = Trans Golgi-Network, PM = Plasma Membrane. Bottom: Sketches of the four main models of Golgi transport (see text). (**B**) Our quantitative model of Golgi self-organisation. The left boundary is the ER, composed of a *cis*-membrane identity, and the right boundary is the TGN, composed of a *trans*-membrane identity. Golgi compartments self-organise *via* three stochastic mechanisms: Fusion: (1) All compartments can aggregate using homotypic fusion mechanisms: the fusion rate is higher between compartment of similar identities. (2) Each compartment can exit the system by fusing homotypically with the boundaries. Budding: (3) Each compartment larger than a vesicle can create a vesicle by losing a patch of membrane. Biochemical conversion: (4) Each patch of membrane undergoes a conversion from a *cis* to a *trans*-identity. New *cis*-vesicles (0) bud from the ER at a constant rate. In the sketch, the boundaries also contain neutral (gray) membrane species that dilute their identity (impact of this dilution in Appendix 7).

mechanisms and one involving transient compartments undergoing *en-block* maturation and in which fusion mechanisms are dispensable for cargo transport. Historically (***Malhotra and Mayor, 2006***), the 'Vesicular transport' model postulated that cisternae have fixed identities and cargo progresses from one cisterna to the next via anterograde vesicular transport, while the 'Cisternal Maturation' model postulated that cisternae themselves undergo maturation from the *cis* to the *trans*-identity and physically move through the stack, while Golgi resident proteins remain in the stack by moving toward the *cis*-face by retrograde vesicular transport. Several extensions of these two models have been proposed, including cisternal maturation with tubular connections (***Trucco et al., 2004***; ***Rizzo et al., 2013***), rapid partitioning within Golgi cisternae between processing and exporting sub-compartments (***Patterson et al., 2008***), or the cisternal progenitor model (***Pfeffer, 2010***), in which stable cisternae exchange material by the fusion and fission of sub-compartments defined by their composition of Rab GTPases, which evolve over time through exchange with the cytosol. The strengths and weaknesses of these different models are nicely reviewed in ***Glick and Luini, 2011***. Variations of these models, such as the 'rim progression' model also exist (***Lavieu et al., 2013***). It is noteworthy that these models do not provide a quantitative analysis of the generation and maintenance of Golgi compartments, nor do they attempt to relate the Golgi structure (number and size of compartments) and transport dynamics. Therefore, much remains unknown regarding the mechanisms that dictate the organisation and dynamics of the Golgi.

Although there is a large diversity in Golgi structures and dynamics among different species, the physiological function of the organelle as a sorting and processing hub is common to all species, suggesting that important mechanisms controlling Golgi dynamics are conserved. Works in evolutionary biology and biophysics have attempted to describe these mechanisms (***Klute et al., 2011***; ***Sens and Rao, 2013***). Different classes of mathematical models have been proposed, from models of vesicle budding and fusion based on rate equations (***Binder et al., 2009***; ***Ispolatov and Müsch,***

*2013*; *Mukherji and O'Shea, 2014*) - some specifically including space (*Sachdeva et al., 2016*), to continuous, discrete and stochastic models of protein sorting in the Golgi (*Gong et al., 2010*; *Vagne and Sens, 2018b*) and computer simulations based on membrane mechanics (*Tachikawa and Mochizuki, 2017*). To account for the ability of the Golgi to reassemble after mitosis (*Wei and Seemann, 2017*), many of these studies have sought to describe the Golgi as a self-organized system (*Binder et al., 2009*; *Kühnle et al., 2010*). However, the manner in which the kinetics of the Golgi two main functions (namely the transport and biochemical conversion of its components) interplay with one another to yield a robust steady-state has so far received much less attention (*Sachdeva et al., 2016*; *Vagne and Sens, 2018b*).

Most existing models discuss transport through a pre-existing compartmentalised Golgi structure. The main goal of our model is to explain how this compartmentalised structure and the flux within it can spontaneously emerge through self-organisation. We propose here a model in which both Golgi self-organisation and vesicular transport are solely directed by (local) molecular interactions resulting in composition-dependent budding and fusion. In particular, we neglect the spatial structure of the Golgi, and assume that any two components of the system can interact with one another (see the Discussion section for the relevance of this approximation). Thus, the directionality of vesicular transport is an emergent property of the self-organisation process that co-evolves with the size and number of Golgi sub-compartments, rather than being fixed by arbitrary rules. This approach brings new conceptual insights that highlight the link between the large-scale steady-state organisation of the Golgi, the directionality of vesicular transport, and the kinetics of individual processes at the molecular scale. We show that a wide variety of organelle phenotypes can be observed upon varying the rates of the local processes. In particular, our model uncovers a correlation between the size and composition (purity) of Golgi sub-compartments. Besides, it emphasizes that composition-dependent exchange between compartments yields complex vesicular flux patterns that are neither purely anterograde (as in the 'vesicular transport' model) nor purely retrograde (as in the 'cisternal maturation' model).

## Model

The model, shown in *Figure 1* and fully described in the Appendix 1 Detailed model is a coarse-grained representation of the Golgi Apparatus. The parameters used in the model are summarised in *Appendix 1—table 1*, and the quantities used to charaterise a compartment and the steady-state of the system are summarised in *Appendix 1—table 2* and *Appendix 1—table 3*, respectively. The system is discretised at the scale of small vesicles, which define the unit size of the system. Vesicles can fuse together or with existing compartments to form larger compartments and can bud from compartments. Compartments consist of a number of membrane patches (of size unity) with distinct identities. The biochemical identity of a membrane patch is defined in a very broad sense by its composition in lipids and proteins that influence its interaction with other patches, such as Rab GTPases (*Grosshans et al., 2006*) or fusion proteins such as SNAREs (*Cai et al., 2007*). Consistently with the three main biochemical identities reported for Golgi membranes (*Day et al., 2013*), vesicles and membrane patches are of either *cis*, *medial* or *trans*-identity, and can undergo irreversible biochemical conversion from the *cis* to the *trans*-identity. The system is fed by a constant influx $J$ of *cis* vesicles coming from the ER, and vesicles and compartments can fuse homotypically with the ER (itself of *cis* identity) and the TGN (of *trans* identity).

Compartments are defined by their size $n$ (a number of patches) and their composition, the fractions $\phi_i$ (with $i = cis, medial, trans$) of patches of the three different identities composing it. These quantities are dynamically controlled by three, composition dependent, microscopic mechanisms – budding, fusion and biochemical conversion – as described below.

- Fusion: Homotypic fusion – the higher probability of fusion between compartments of similar identities – is a feature of cellular organelles in general (*Antonin et al., 2000*) and the Golgi apparatus in particular (*Pfeffer, 2010*; *Bhave et al., 2014*). This is controlled here by a fusion rate between compartments proportional to the probability that they present the same identity at the contact site. The total fusion rate for two compartments $(a)$ and $(b)$ with composition $\phi_i^{(a)}$ and $\phi_i^{(b)}$ for each identity $i$ (with $i$ equals *cis*, *medial* and *trans*) is then:

$$K_{\mathrm{f}} \times \sum_i \phi_i^{(a)} \phi_i^{(b)} \qquad (1)$$

where $K_{\mathrm{f}}$ is the fusion rate between two identical compartments. Fusion with the boundaries follow the same rules, with the ER containing a fraction $\alpha_{\mathrm{ER}}$ of *cis* patches and the TGN a fraction $\alpha_{\mathrm{TGN}}$ of *trans* patches. Numerical results described in the main paper are for $\alpha_{\mathrm{ER}} = \alpha_{\mathrm{TGN}} = 1$, and the role of the composition of the boundaries is discussed in the Appendix 7.

- Budding: Budding is the emission of a vesicle from a larger compartment. Composition-dependent vesicular budding is an important sorting mechanism in cellular traffic (***Bonifacino and Glick, 2004***). Owing to the high specificity of the budding machinery, each vesicle is assumed to be composed of a single membrane identity. The budding flux for any components $i$ (equals *cis*, *medial* or *trans*) present in a compartment is assumed not to depend on the number $n_i = n\phi_i$ of patches of identity $i$, but on the total size of the compartment:

$$J_{\mathrm{b},i} = K_{\mathrm{b}} \times n. \qquad (2)$$

This budding kinetics corresponds to the saturated regime of a Michaelis-Menten reaction kinetics, and is appropriate to the case where budding actors (e.g. coat proteins) bind non-specifically to the membrane of the compartment and find their budding partners (a patch of a particular identity) by diffusion (***Vagne and Sens, 2018b***). An alternative budding scheme where the budding flux is linear with the number of patches of a given identity ($J_{\mathrm{b},i} = K_{\mathrm{b}} n_i$) is discussed in Appendix 8.

- Biochemical conversion: Each membrane patch is assumed to undergo stochastic biochemical conversion from a *cis* to a *medial* to a *trans*-identity. This local mechanism of identity conversion is motivated by the Rab cascade, during which the membrane identity evolves over time through molecular exchange with the cytosol, as was observed in endosomes (***Rink et al., 2005***) and in the Golgi of Yeast (***Losev et al., 2006***; ***Matsuura-Tokita et al., 2006***). The Rab conversion mechanism has not been demonstrated in mammalian Golgi, but is thought to be universally important for the specificity of membrane trafficking (***Grosshans et al., 2006***), and many of the proteins involved in the Yeast Rab cascade are conserved in mammalian cells (***Klute et al., 2011***). Rab conversion is at the basis of the cisternal progenitor model (***Pfeffer, 2010***), and it is thus of fundamental importance to understand its potential implication on Golgi structure and dynamics with the use of a quantitative model. We stress that our model of biochemical conversion is not limited to Rab GTPases, but can also correspond to the modification of lipid components by enzymes in the Golgi (***McDermott and Mousley, 2016***). This is a local mechanism of identity conversion distinct from the maturation of entire compartments, which is also affected by the dynamics of budding and fusion. To limit the number of parameters, the biochemical conversion rate $K_{\mathrm{m}}$ is the same for the two reactions (*cis*→medial and medial→trans) and is independent of the concentration. Introducing cooperativity in the conversion process is expected to increase the robustness of the self-organisation process (***Vagne and Sens, 2018a***).

In its simplest form, our model contains only four parameters: the rates of injection, fusion, budding, and biochemical conversion ($J$, $K_{\mathrm{f}}$, $K_{\mathrm{b}}$, $K_{\mathrm{m}}$). By normalizing time with the fusion rate, we are left with three parameters: $j = J/K_{\mathrm{f}}$, $k_{\mathrm{b}} = K_{\mathrm{b}}/K_{\mathrm{f}}$ and $k_{\mathrm{m}} = K_{\mathrm{m}}/K_{\mathrm{f}}$. The dynamics of the system is entirely governed by these stochastic transition rates, and can be simulated exactly using a Gillespie algorithm (***Gillespie, 1977***), described in Appendix 1.

Codes used for the simulation can be found in Source code 1.

## Results

### Mean-field description of the system

The complexity of the system prohibits rigorous analytical calculation. Nevertheless, analytical results can be obtained for a number of interesting quantities in certain asymptotic regimes where simplifying assumptions can be made. We present below some of these derivations.

## De-novo formation and steady-state composition in the well-sorted limit

The composition of the system at steady-state is difficult to compute due to the fact that the exit of components from the system, through fusion with the boundaries, depends on the composition of the exiting compartments, which cannot be derived analytically in the general case. This calculation becomes straightforward if the system is well sorted and all compartments are pure. As explained below, this corresponds to the limit of high budding rate: $k_{\mathrm{b}} \gg k_{\mathrm{m}}$. In the well-sorted limit, only *cis* components exit through the *cis* boundary and only *trans* components exit through the *trans* boundary, and mean-field equations can be derived for the total amount $N_{cis}$, $N_{medial}$ and $N_{trans}$ of each species:

$$\frac{\partial N_{cis}}{\partial t} = j - N_{cis}(\alpha_{\mathrm{ER}} + k_{\mathrm{m}}) \ , \ \frac{\partial N_{medial}}{\partial t} = k_{\mathrm{m}}(N_{cis} - N_{medial}) \ , \ \frac{\partial N_{trans}}{\partial t} = k_{\mathrm{m}}N_{medial} - \alpha_{\mathrm{TGN}}N_{trans} \tag{3}$$

With $j$ the injection rate of new *cis*-vesicles in the system and $k_{\mathrm{m}}$ the biochemical conversion rates, both normalized by the fusion rate, and $\alpha_{\mathrm{ER}}$ ($\alpha_{\mathrm{TGN}}$) is the fraction of *cis* (*trans*) species promoting homotypic fusions with Golgi components in the ER (TGN). At steady-state, the total amounts of *cis*, *medial* and *trans*-species are fixed by the balance between influx, exchange, biochemical conversion and exit:

$$N_{cis} = N_{medial} = \frac{j}{\alpha_{\mathrm{ER}} + k_{\mathrm{m}}} \ , \ N_{trans} = \frac{k_{\mathrm{m}}}{\alpha_{\mathrm{TGN}}} \frac{j}{\alpha_{\mathrm{ER}} + k_{\mathrm{m}}} \tag{4}$$

To estimate the typical size of compartments, we assume that for each species, a single large compartment of size $n$ interacts through budding and fusion with a number $n_{\mathrm{v}}$ of vesicles of the same identity (so that $N = n + n_{\mathrm{v}}$). The compartments size for each species then satisfies:

$$\frac{\partial n}{\partial t} = n_{\mathrm{v}} - k_{\mathrm{b}}n \quad \xrightarrow{\text{steady state}} \quad n = \frac{N}{1 + k_{\mathrm{b}}} \tag{5}$$

To limit the number of parameters, we fix the average system's size $N = N_{cis} + N_{medial} + N_{trans}$ to a set value $N = 300$ in the main text, which is suitable for Golgi ministacks whose total area is of the order of 1 μm$^2$ (the area of a mammalian Golgi ribbon is much larger) (*Yelinek et al., 2009*), corresponding to about a few hundreds of vesicles of diameter ~ 10-50 nm. This is obtained by adjusting the influx to the variation of the other parameters according to:

$$j = \frac{N(\alpha_{\mathrm{ER}} + k_{\mathrm{m}})\alpha_{\mathrm{TGN}}}{2\alpha_{\mathrm{TGN}} + k_{\mathrm{m}}} \tag{6}$$

We show in *Appendix 1—figure 1* that the overall structure of the phase diagrams for the compartments size and composition is robust upon variation of the average system's size.

The de-novo formation of the system can be approached with *Equation 3*. Starting with an empty system with an influx $J (= K_f j)$ of *cis* vesicles, the system grows stochastically to reach the steady-state size after a time of order $1/K_{\mathrm{f}}$. The evolution of the size of the system with time obtained from numerical simulations, shown in *Appendix 4—figure 1* for different sets of parameters, agrees with *Equation 3*. At steady-state, the influx of vesicles is balanced by an outflux of material. The exit kinetics can be quantified by looking at the fate of a pulse of cargo released from the ER at a given time. *Appendix 4—figure 2* shows that the exit kinetics is exponential, in agreement with results of FRAP (*Patterson et al., 2008*) or pulse-chase (*Boncompain et al., 2012*) experiments.

Although *Equation 6* is only valid in the well-sorted limit, it permits a satisfactory control of the average system size over the entire range of parameters, with only a 10% variation for small budding rates (Appendix 5, *Appendix 5—figure 1*). At steady-state, the system size exhibits stochastic fluctuations around the mean value that depend on the parameters (Appendix 5, *Appendix 5—figure 1*). The large fluctuation (about 30%) for low budding rate $k_{\mathrm{b}}$ stem from the fact that compartments are large and transient, as explained below.

## Compartment size and composition in the maturation-dominated regime

For low values of the budding rate $k_{\mathrm{b}}$, compartments do not shed vesicles and evolve in time independently of one another in a way dictated by a balance between fusion-mediated growth and biochemical maturation. Their size and composition can be approximatively calculated by assuming that

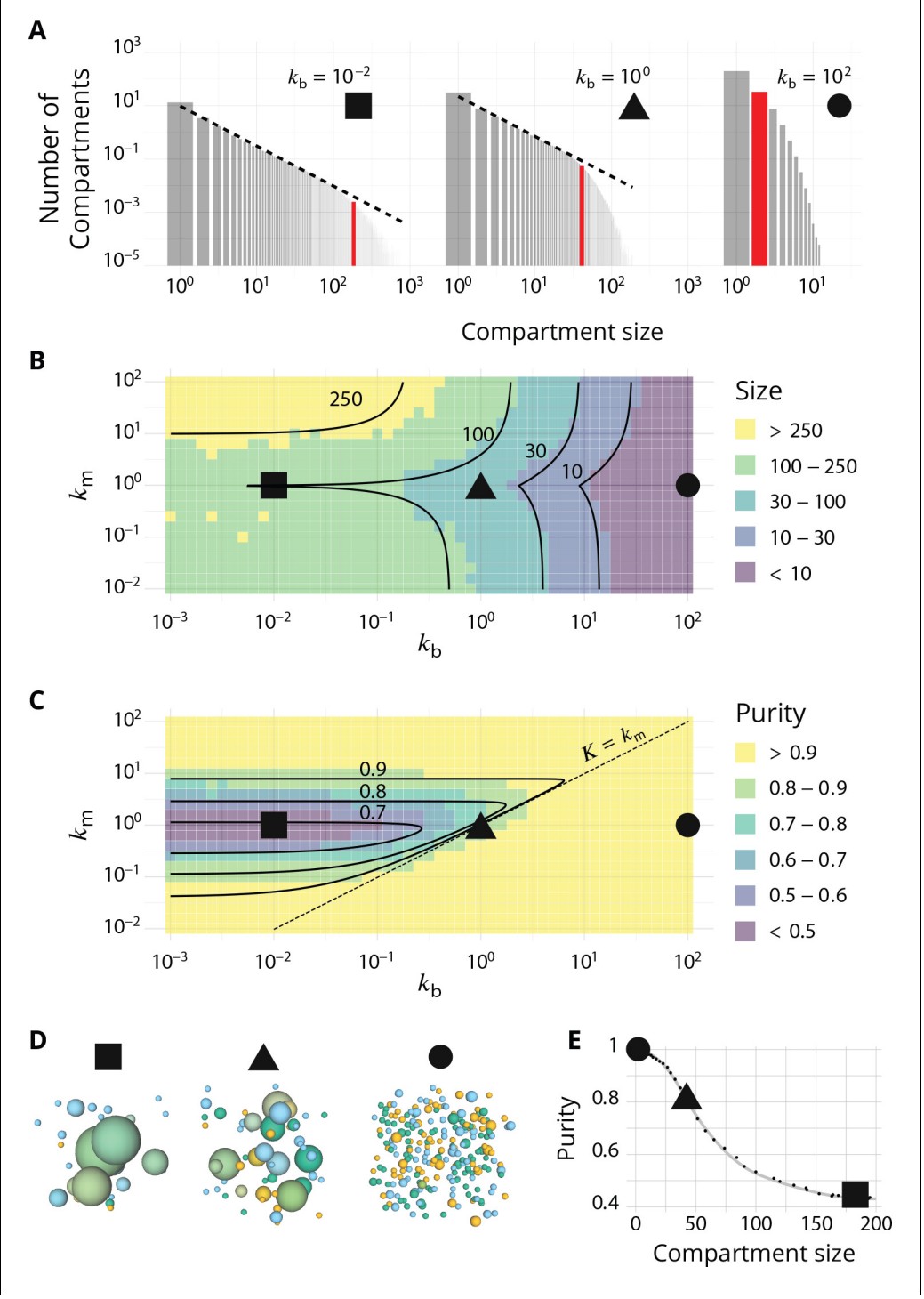

**Figure 2.** Steady-state of the self-organized model of Golgi apparatus. (A) Size distribution of compartments for a biochemical conversion rate $k_m = 1$ and different values of the budding rate $k_b$. The red bar shows the characteristic compartment size. (B) Phase diagram of the size of the compartments as the function of $k_b$ and $k_m$. Black lines: theoretical prediction (*Equations 4,6*) (C) Phase diagram of the purity of the system as a function of $k_b$ and $k_m$. Dashed line is $k_m = k_b$, black lines are a theoretical prediction (see Appendix 3). (D) System snapshots for $k_m = 1$ showing the *mixed regime* (square - $k_b = 10^{-2}$), the *sorted regime* (triangle - $k_b = 1$), and the *vesicular regime* (circle - $k_b = 10^2$) - see text. (E) Average compartment purity as a function of their average size, obtained by varying the budding rate $k_b$ for $k_m = 1$ (black dots: simulation results – gray line: guide for the eyes). For all panels, $\alpha_{ER} = \alpha_{TGN} = 1$. See also Appendix 5 for further characterizations of the steady-state organisation, *Figure 2 continued on next page*

*Figure 2 continued*

Appendix 7 for the role of composition of the exit compartment, and Appendix 8 for results with alternative budding and fusion kinetics.

---

compartments grow in size by fusion from a constant pool of vesicles containing the same number $n_v$ of vesicle for all three identities. Calling $n_{cis}(t)$, $n_{medial}(t)$ and $n_{trans}(t)$ the amount of components of the three identities in a given compartment of total size $n = n_{cis} + n_{medial} + n_{trans}$, and neglecting vesicle budding, the mean-field equations satisfied by these quantities are:

$$\partial_t n_{cis} = n_v \frac{n_{cis}}{n} - k_m n_{cis} \ , \ \ \partial_t n_{medial} = n_v \frac{n_{medial}}{n} + k_m(n_{cis} - n_{medial}) \ , \ \ \partial_t n_{trans} = n_v \frac{n_{trans}}{n} + k_m n_{medial}. \quad (7)$$

Starting with a vesicle of *cis* identity for $t = 0$: $n_{cis}(0) = 1$, $n_{medial}(0) = n_{trans}(0) = 0$, the compartment size evolves linearly with time: $n(t) = 1 + n_v t$, and the composition of each species satisfies:

$$
\begin{aligned}
n_{cis}(t) &= (1 + n_v t)e^{-k_m t} \\
n_{medial}(t) &= (1 + n_v t)k_m t \, e^{-k_m t} \\
n_{trans}(t) &= (1 + n_v t)\big(1 - (1 + k_m t)e^{-k_m t}\big)
\end{aligned}
\quad (8)
$$

The fraction of each species in the compartment is thus independent on $n_v$, and reads:

$$\phi_{cis}(t) = e^{-k_m t} \ , \ \phi_{medial}(t) = k_m t \, e^{-k_m t} \ , \ \phi_{trans}(t) = \big(1 - (1 + k_m t)e^{-k_m t}\big). \quad (9)$$

These results show that in this regime, a given compartment smoothly evolves from a mostly *cis* to a mostly *trans* identity over a typical time $1/K_m$. The maximum amount of medial identity is obtain for $k_m t = 1$ and is $\phi_{medial} = 1/e \simeq 0.37$. Therefore, the mechanism does not lead to pure *medial* compartments. These analytical results are confirmed by the full numerical solution of the system in the low budding rate regime shown in Appendix 5 and discussed in detail below.

## Steady-state organisation

The steady-state organisation and dynamics of the system is described in terms of the average size and purity of compartments (introduced here and detailed in the Appendix 2). The stationary size distribution of compartments is a decreasing function of the compartment size, which typically shows a power law decay at small size, with an exponential cut-off at large size (*Figure 2A*). This is expected for a non-equilibrium steady-state controlled by scission and aggregation (*Turner et al., 2005*) and means that there exist many small compartments and very few large ones. The typical compartment's size is defined as the ratio of the second over the first moments of the size distribution. This corresponds to a size close to half the size of the exponential cut-off (see Appendix 2) beyond which it is unlikely to find a compartment (*Vagne et al., 2015*).

The typical compartment size varies with parameters as shown in *Figure 2B*. Increasing the ratio of budding to fusion rate $k_b$ decreases the compartment size (*Turner et al., 2005*). The size depends on the biochemical conversion rate $k_m$ in a non-monotonic manner: for a given budding rate, the size is smallest for intermediate values of the biochemical conversion rate ($k_m \simeq 1$). The conversion rate controls the composition of the compartment and hence their fusion probability. This dependency is well explained by the simple analytical model discussed above (*Equations 3,4*) that approximates the system by three pure compartments interacting with a pool of vesicles (solid lines in *Figure 2B*).

The purity of a given compartment is defined as the (normalized) Euclidean distance of the composition of the compartment (the fractions $\phi_i$ in the different *i*-identities, $i = cis$, *medial* and *trans*) from a perfectly mixed composition. $P = 0$, $1/2$, $1$ correspond respectively to compartments that contain the same amount of the three identities, the same amount of two identities, and a single identity (Appendix 2). The purity of the system is the average purity of each compartment weighted by its size, ignoring vesicles. The dependency of the average purity with the parameters is shown in *Figure 2C*. As for the size of the compartments, it is non-monotonic in the biochemical conversion rate, and regions of lowest purity are found for intermediate biochemical conversion rates $k_m \simeq 1$, when all three identities are in equal amount. The purity increases with the budding rate $k_b$ in a sigmoidal fashion (see also *Figure 3A*). This can be understood by analyzing the processes involved in

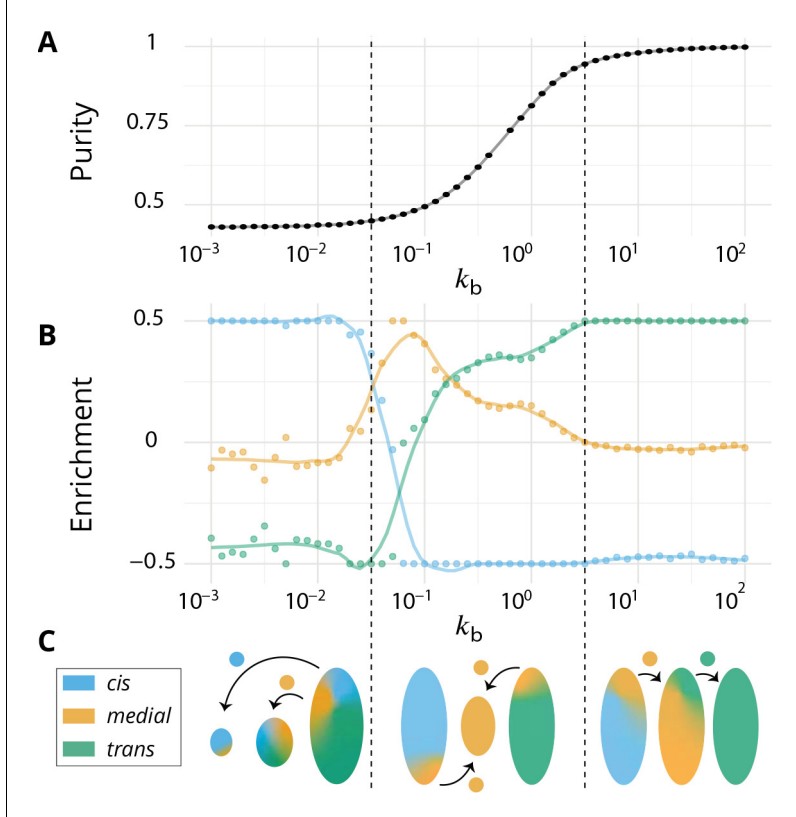

**Figure 3.** Directionality of vesicular transport, displayed varying the budding rate $k_b$ for a biochemical conversion rate $k_m = 1$ (equal amount of all species in the system). (**A**) System purity as a function of the budding rate – a cut through the purity phase diagram shown *Figure 2C*. (**B**) Normalized enrichment in cis (blue), *medial* (orange) and *trans* (green) identities between the acceptor and donor compartments during vesicular exchange (solid lines are guides for the eyes). See Appendix 6 for non-normalized fluxes. (**C**) Sketches showing the direction of the dominant vesicular fluxes – consistent with data of A and B and data from *Appendix 6—figure 1* – omitting less contributive transports. Before the purity transition (low values of $k_b$ - purity ~ 0.5) the vesicular flux is retrograde. After the transition (high values of $k_b$ - purity ~ 1) the vesicular flux is anterograde. Around the crossover ($k_b \sim 0.1$), the vesicular flux is centripetal and oriented toward *medial*-compartments. The centripetal flux disappears if inter-compartments fusion is prohibited, see Appendix 8.

mixing and sorting of different identities. In a first approximation, mixing occurs by biochemical conversion and sorting occurs by budding of contaminating species, suggesting a transitions from low to high purity when the budding rate reaches the biochemical conversion rate ($k_b \simeq k_m$). A 'purity transition' is indeed observed in an intermediate range of biochemical conversion rates ($0.1 < k_m < 10$, see *Figure 2C*). It is most pronounced for $k_m \simeq 1$. Beyond this range, the system is dominated by one identity and compartments are always pure. The purity variation can be qualitatively reproduced by an analytical model (Appendix 3) that accounts for the competition between biochemical conversion and budding, but also includes fusion between compartments and with the exits (solid lines in *Figure 2C*).

In summary, three types of organisation can be found, mostly controlled by the ratio of budding over fusion rate $k_b$: a *mixed regime* at low $k_b$, where compartments are large and contain a mixture of identities, a *vesicular regime* at high $k_b$, where compartments are made of only a few vesicles and are very pure, and a *sorted regime* for intermediate values of $k_b$, where compartments are both rather large, and rather pure. Snapshots of these three steady-states are shown in *Figure 2D*. The relationship between average size and average purity of compartments is shown in *Figure 2E*. The existence of a well-defined intermediate regime with large sorted compartments is promoted by our assumption of 'saturated budding' where the budding flux of any identity only depends on the

compartment size (*Equation 2*). If the budding flux is linear with the number of patches of a given identity, the purity transition occurs for larger values of the budding rate $k_b$, where compartments are smaller (see Appendix 8).

## Vesicular transport

Vesicular exchange between compartments is quantified by following the dynamics over time of passive cargo injected from the ER. Each cargo in a compartment of size $n$ has a probability $1/n$ to join a vesicle budding from this compartment. When this vesicle fuses with another compartment or a boundary, the composition difference $\Delta\phi_i$ between the receiving and emitting compartment (a number between $-1$ and 1) is recorded for the three $i$-identities ($i =$ *cis*, *medial*, *trans*). Averaged over all budding/fusion events, this defines the *enrichment vector* $\vec{E}$, whose three components $E_i$ are normalized for readability so that $\sum_i |E_i| = 1$ (see *Appendix 6—figure 1* for non-normalized enrichment). Vesicular flux is predominantly anterograde if $E_{cis} < 0$ and $E_{trans} > 0$, while it is predominantly retrograde if $E_{cis} > 0$ and $E_{trans} < 0$.

We focus on systems containing comparable amounts of each species at steady-state ($k_m = 1$ - see *Equation 6*). The enrichment in cis, *medial* and *trans*-identities are shown in *Figure 3B* as a function of the budding rate $k_b$. The most striking result is the high correlation between purity of the compartment and the directionality of vesicular transport. For low values of the budding rate ($k_b \ll 1$), compartments are well mixed (purity $\leq 1/2$) and the vesicular flux is retrograde, with a gain in cis-identities and a loss in trans-identities. For high values of the budding rate ($k_b \gg 1$), compartments are pure (purity $\simeq 1$) and the vesicular flux is anterograde, with a gain in trans-identities and a loss in cis-identities. The archetypal behaviors most often discussed in the literature are thus, within the limits of our model, asymptotic regimes for extreme values of the ratio of budding to fusion rates. Remarkably, the cross over between these two asymptotic behaviors is rather broad ($k_b \sim 0.1 - 1$) and displays a more complex vesicular transport dynamics, mostly oriented toward medial compartments.

In our model organelle, vesicle budding and fusion are biased solely by local compositions, which is subjected to irreversible biochemical conversions. The interplay between these microscopic processes gives rise to an irreversible flux of vesicles across the system. To better understand the directionality of vesicular exchanges, the net vesicular flux leaving compartments with a given composition is represented as a vector field in the compartments composition space in *Figure 4*. This is shown as a triangle plots in which each point corresponds to a fraction of *cis*, *medial* and *trans* components in a compartment, and the three vertices of the triangle correspond to pure compartments (see Appendix 2). The variation of the vesicular flux with the budding rate $k_b$ (*Figure 4*) is consistent with that of the composition enrichment in the different identities (*Figure 3B*). As $k_b$ increases, the flux evolves from being mostly retrograde at low $k_b$ (from *trans*-rich toward less mature compartments) to being anterograde at high $k_b$ (from *cis* to *medial*, and *medial* to *trans*-compartments). In between, the vesicular flux is centripetal toward mixed compartments, leading to a net enrichment in *medial*-identity. The relationship between structure and transport is explained below, and a dissection of this fluxes with respect to the composition of the vesicles is shown in Appendix 6. The net flux leaving a particular region of the triangular phase space is proportional to the total system's size (number of compartments times their size) with this composition, while the flux arriving at a particular region depends on the number of compartments with that composition. Both quantities are shown on *Figure 4*.

At low budding rate ($k_b \ll 1$ - top row in *Figure 4*), the system is in the mixed regime, where compartments are large (hundreds of vesicles in size) but mixed (purity $\leq 1/2$). The system is dominated by compartment maturation, and the compartment distribution is spread around a path going from purely *cis* to purely *trans*-compartments, consistent with a dynamics where each compartment maturates independently from the other and given by *Equation 9*. - see top left panel of *Figure 4*. The majority of the transport vesicles are emitted by *trans*-rich compartments, which concentrate most of the components of the system, leading to a retrograde vesicular flux. Note that although vesicular flux is clearly retrograde in the maturation-dominated regime, as seen from the increase in cis identity and the decrease in trans identity between emitting and receiving compartment (*Figure 3B*), there is no net flux of vesicles from *medial*-rich to *cis* compartments in this limit (*Figure 4*). This differs from the classical 'cisternal progression and maturation' mechanism sketched in

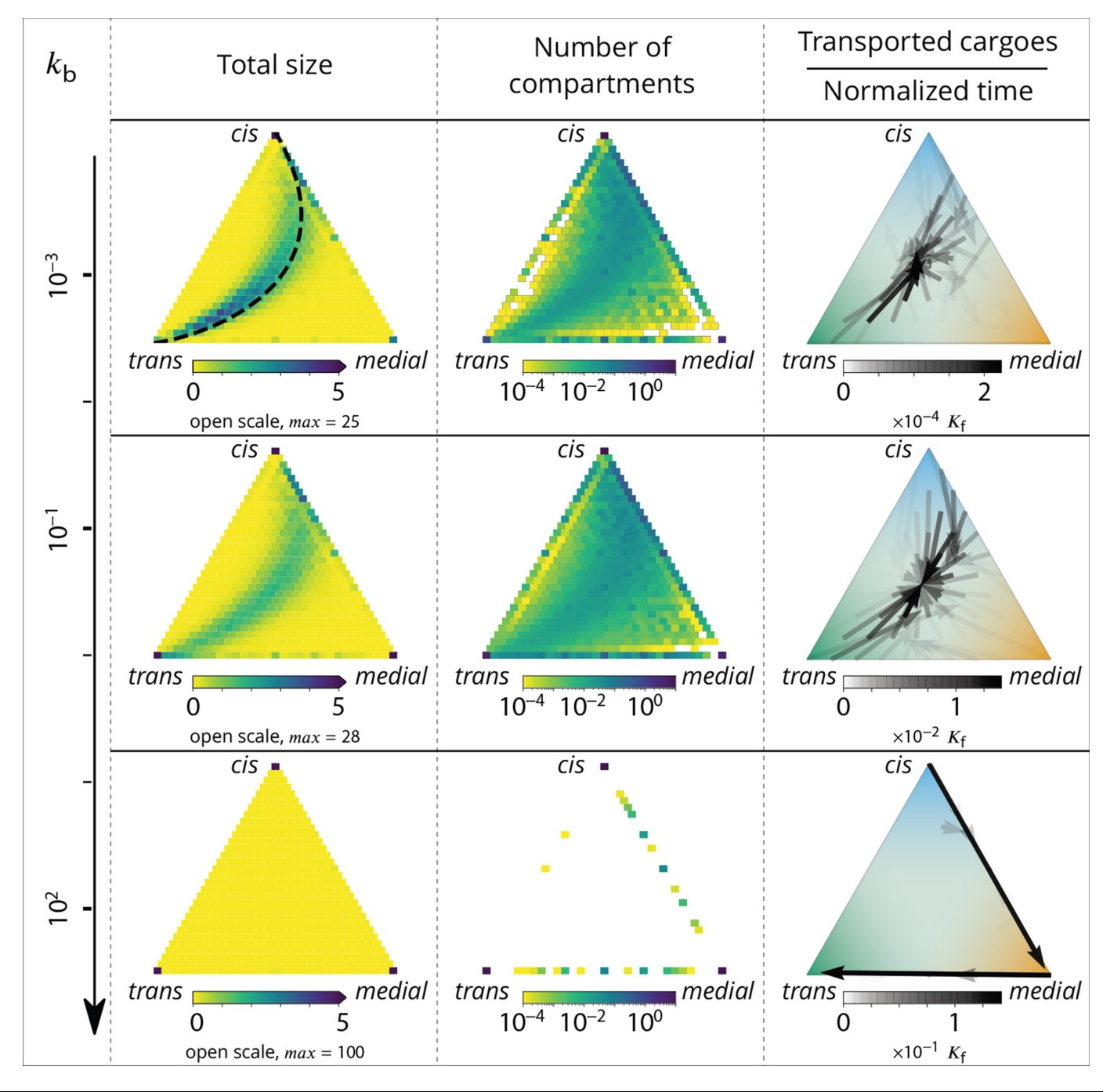

**Figure 4.** Relationship between the structure of the system and the vesicular fluxes. Distribution of the total size of the system (truncated scale, maximum value in brackets) and number of compartments, and the vesicular flux between them, shown in a triangular composition space. Each point represents a given composition of compartments and the triangle vertices are compartments of pure *cis*, *medial* and *trans* identity. Arrows show the vesicular flux. The base of each arrow is the composition of the donor compartment and the tip is the average composition of receiving compartments (ignoring back fusion). The opacity of the arrows is proportional to the flux of transported cargo going through this path per unit time ($1/K_f$), normalized by the total number of cargo-proteins. The dashed line on the top-left triangle is the theoretical prediction given by *Equation 9*. See also Appendix 6 for further characterizations of the vesicular transport, and Appendix 8 to discuss the impact of other budding and fusion implementations on the vesicular fluxes.

*Figure 1A*, and is due to the fact that the compartments that contain the highest fraction of *medial* identity in this regime are well mixed ($\phi_{cis} \simeq \phi_{medial} \simeq \phi_{trans} \simeq 1/3$). The vesicular flux leaving such compartments is equally split into *cis, medial* and *trans* vesicles, which fuse homotypically with compartments enriched in their identity, yielding a vanishing net flux averaged over the three identities. There is nevertheless a strong retrograde flux of *cis* vesicles toward *cis* compartments, as shown in *Appendix 6—figure 1*. In the classical 'cisternal maturation' mechanism, retrograde transport is needed to recycle *cis*-Golgi components from more mature compartments. Within the present model, this can be achieved by targeting such components to *cis* vesicles budding from more mature compartments.

At high budding rate ($k_b \gg 1$ - bottom row in *Figure 4*), the system is in the vesicular regime, where compartments are fairly pure, but rather small, with a size equivalent to a few vesicles. Compartments are exclusively distributed along the *cis/medial* and *medial/trans*-axes of composition, and accumulate at the vertices of the triangle. Membrane patches that just underwent conversion bud quickly from the donor compartment and fuse with pure compartments with the appropriate *medial* or *trans*-identity, explaining the clear anterograde vesicular transport. This feature is further reinforced under very high budding rate, where the vesicular flux is dominated by budding vesicles undergoing biochemical conversion *before* fusion, which prohibits their back fusion with the donor compartment.

In the intermediate regime (middle row in *Figure 4*), the distribution of mixed compartments is more homogeneous along the line describing the maturation of individual compartments - both in terms of size and number of compartments. *Cis*-rich and *trans*-rich compartments emit comparable amount of transport vesicles, while large *medial*-rich compartments are absent. This leads to a centripetal vesicular flux with an enrichment in *medial* identity (see *Figure 3B*). The lack of large *medial*-rich compartment is due to the fact that they can be contaminated by either *cis* or *trans*-species and fuse with *cis*-rich or *trans*-rich compartments. If inter-compartment fusion is prohibited, the enrichment in medial-identity in the intermediate regime is abolished, with a direct transition from anterograde to retrograde vesicular transport upon increasing $k_b$ (see Appendix 8 and the Discussion section).

A graphical sketch consistent with our numerical observations for the typical size and composition of different compartments and for the dominant vesicular fluxes between them in the different regimes is shown in *Figure 3C*.

## Discussion

The genesis and maintenance of complex membrane-bound organelles such as the Golgi apparatus rely on self-organisation principles that are yet to be fully understood. Here we present a minimal model in which steady-state Golgi-like structures spontaneously emerge from the interplay between three basic mechanisms: the biochemical conversion of membrane components, the composition-dependent vesicular budding, and inter-compartment fusion. As our model does not include space, we do not refer to the spatial structure of a stacked Golgi, but rather to the self-organisation of Golgi components into compartments of distinct identities undergoing vesicular exchange. We show that directional vesicular exchanges between compartments of different identities spontaneously emerge from local biochemical interactions, despite the absence of spatial information. This is in line with recent observations that the Golgi functionality is preserved under major perturbation of its spatial structure resulting from land-locking Golgi cisternae to mitochondria (*Dunlop et al., 2017*). Indeed, cargo transport still proceed, and is merely slowed down by a factor of two (from 20 min to 40 min) in land-locked Golgi despite the absence of spatial proximity between cisternae. This suggests that spatial information is less crucial to the Golgi function than biochemical information. Our main result is that the steady-state organisation of the organelle, in terms of size and composition of its compartments, is intimately linked to the directionality of vesicular exchange between compartments through the mechanism of homotypic fusion. Both aspects are controlled by a kinetic competition between vesicular exchange and biochemical conversion.

The results presented in *Figure 2* can be used to explain a number of experimental observations. The predicted role of the budding and fusion rates is in agreement with the phenotype observed upon deletion of Arf1 (a protein involved in vesicle budding), which decreases the number of compartments and increases their size, particularly that of *trans*-compartments which seem to aggregate

into one major cisterna (*Bhave et al., 2014*). A comparable phenotype has been observed upon mutation of NSF (a fusion protein), which produces extremely large, but transient, *trans*-compartments (*Tanabashi et al., 2018*), thereby increasing the stochasticity of the system. This is consistent with an increase of the fusion rate according to our model (see Appendix 5 for a quantification of the fluctuations in our model). Modifying the expression of *VPS74* (yeast homolog of *GOLPH3*) leads to an altered Golgi organisation, comparable to the *ARF1* deletion phenotype (*Iyer et al., 2018*). Both Δ*arf1* and Δ*vps74* present an enlargement of the Golgi cisternae and a disruption of molecular gradients in the system, which we interpret, in the limits of our system, as a lower purity due to slower sorting kinetics following an impairment of the budding dynamics. Importantly, the correlation between the size and purity of the compartments predicted by our model is in agreement with the observation that decreasing the budding rate by altering the activity of COPI (a budding protein) leads to larger and less sorted compartments (*Papanikou et al., 2015*). In yeast again, over-expression of Ypt1 (a Rab protein) increases the transition rate from early to transitional Golgi, and increases the co-localization of early and late Golgi markers (*Kim et al., 2016*). We interpret this as a decrease of the purity of Golgi cisternae upon increasing the biochemical conversion rate by unbalancing the ratio of budding to conversion rates. This suggests that the wild-type Yeast Golgi is closed to the line $k_{\mathrm{m}} = k_{\mathrm{b}}$ shown in *Figure 2*.

One crucial model prediction is that the size and purity of Golgi compartments should be affected in a correlated fashion by physiological perturbations. This could be tested experimentally by further exploring the phase diagrams of *Figure 2B–C* by simultaneously varying the ratios of biochemical conversion over fusion rate $k_{\mathrm{m}}$ and of budding over fusion rate $k_{\mathrm{b}}$. Compartment purity can be altered without changing their size by acting on $k_{\mathrm{m}}$, while it should be correlated with a change of size by acting on $k_{\mathrm{b}}$. We thus predict that the purity decrease concomitant with the size increase observed in *Papanikou et al., 2015* upon impairing COP-I activity (decreasing $k_{\mathrm{b}}$) could be reversed *without a change of cisterna size* upon decreasing the biochemical conversion rate $k_{\mathrm{m}}$, for example by impairing Ypt1 activity (as done in *Kim et al., 2016*). Along the same line, we predict that the decrease of purity observed by *Kim et al., 2016* upon Ypt1 over-expression (which increases the early to transitional conversion rate) should not be associated to change of cisternae size (at steady-state) if altering Ypt1 does not modify the budding rate $k_{\mathrm{b}}$. We can predict further that the loss of purity phenotype could be reversed upon increasing the budding rate by over-expressing COP-I, but that this would be accompanied by an decrease of cisternae size. This experimental proposal is represented graphically on the size and purity phase diagrams in *Appendix 9—figure 1*, with somewhat arbitrary arrows, to give a feel for the way our model may be used to design experimental strategies.

The directionality of vesicular transport is intimately linked to the steady-state organisation of the organelle; vesicular transport is anterograde when compartments are pure, and it is retrograde when they are mixed. This can be intuitively understood with the notion of 'contaminating species'. If a compartment enriched in a particular molecular identity emits a vesicle of that identity, the vesicle will likely fuse back with the emitting compartment by homotypic fusion, yielding no vesicular exchange. One the other hand, an emitted vesicle that contains a minority (contaminating) identity will homotypically fuse with another compartment. If budding is faster than biochemical conversion, the contaminating species is the one that just underwent conversion, and its budding and fusion with more mature compartment leads to an anterograde transport. In such systems, compartments are rather small and pure. If biochemical conversion is faster than budding, the contaminating species is the less mature one, leading to a retrograde transport. Such systems are rather mixed with large compartments. Thus, the directionality of vesicular exchange is an emergent property intimately linked to the purity of the compartments. The *medial*-rich compartments are special in that regard: for intermediate values of the purity ($k_{\mathrm{b}} \sim k_{\mathrm{m}}$), they may be contaminated both by yet to be matured *cis*-species and already matured *trans*-species and can fuse both with *cis*-rich and *trans*-rich compartments. If inter-compartment fusion is allowed, large *medial*-rich compartments are relatively scarce, and emit few vesicles. On the other hand, *medial*-vesicles emitted by *cis/medial* and *medial/ trans*-compartments may fuse together to form (small) *medial*-rich compartments. For intermediate values of the budding rates, this vesicular flux dominates vesicular exchange and leads to a centripetal vesicular flux towards *medial*-compartments (details in Appendix 6). If inter-compartment fusion is prohibited, which for instance corresponds to a situation where Golgi cisternae are immobilized (*Dunlop et al., 2017*), *medial*-rich compartments are present at steady-state, the centripetal

vesicular flux is less intense, and the purity transition is accompanied with a direct transition from retrograde to anterograde vesicular flux (Appendix 8). Note that vesicular transport of cargo through the Golgi is not bound to follow the net vesicular flux between compartments discussed above. Indeed, the net flux is the average of the flux of vesicles of the three identities. A given cargo follows this flux - on average - if it does not interact preferentially with membrane of particular identity. On the other hand, a cargo that is preferentially packaged into vesicles of *trans* identity (for example) will be transported toward *trans*-rich compartment even if the net vesicular flux is mostly retrograde.

Regardless of the details of the model, we find that systems showing a well defined polarity with well sorted cisternae exhibit anterograde vesicular fluxes, whereas systems with mixed compartments exhibit retrograde fluxes. The former dynamics is expected when biochemical conversion is the slowest kinetic process and compartments are long-lived, while the latter is expected when vesicular exchange is slow and the system is composed of transient compartments undergoing individual maturation. One can relate this prediction to the difference in organisation and dynamics between the Golgi of *S. cerevisiae* and the more organized Golgi of higher organisms such as vertebrates. Maturation of Golgi cisternae has been directly observed in *S. cerevisiae* (*Losev et al., 2006*), with colocalization of different identity markers within single cisternae (low purity) during maturation, whereas vesicular transport phenotypes have been indirectly observed (*Dunlop et al., 2017*) or inferred through modeling (*Dmitrieff et al., 2013*) in mammalian cells. Consistent with our predictions, Golgi dynamics is one to two orders of magnitude faster in *S. cerevisiae* Golgi cisternae (as measured by the typical maturation rate of cisternae $\sim 1/\mathrm{min}$ [*Losev et al., 2006*; *Matsuura-Tokita et al., 2006*]) than in mammalian cells (as measured by the typical Golgi exit rate of cargo $\sim 1/20 \ \mathrm{min}$ to $\sim 1/40 \ \mathrm{min}$ *Bonfanti et al., 1998*; *Hirschberg et al., 1998*; *Patterson et al., 2008*; *Boncompain et al., 2012*]). At this point, one may speculate a link between structure and function through kinetics. A well sorted and polarized Golgi is presumably required to accurately process complex cargo. The glycosylation of secreted cargo is key to the interaction of a vertebrate cell with the immune system of the organism it belongs to *Ryan and Cobb, 2012*, and glycans appear to be more diverse in higher eukaryotes - which also possess a highly organized Golgi - than in unicellular eukaryotes like yeast (*Wang et al., 2017*). The Golgi organisation in *S. cerevisiae* could thus be the result of an adaptation that has favored fast transport over robustness of processing, leading to a less organized Golgi characterized by cisternal maturation and retrograde vesicular transport. Remarkably, the slowing down of Golgi transport when *S. cerevisiae* is starved in a glucose-free environment coincides with the Golgi becoming more polarized (*Levi et al., 2010*), strengthening the proposal derived from our model of a strong connection between transport kinetics and steady-state Golgi structure.

The present model is designed to account for crucial dynamical processes at play in the Golgi with a limited number of parameters. It is sufficient to yield robust self-organised structures which can reproduce a number of experimental observations, but cannot be expected to account for the full richness of Golgi dynamics, especially under severe perturbation. The model includes a single kind of exchange mechanism based on vesicles. Exchange could also proceed via the pinching and fusion of larger cisterna fragments (*Pfeffer, 2010*; *Dmitrieff et al., 2013*), or via tubular connection between compartments (*Trucco et al., 2004*), although the relevance of the latter to Golgi transport is still debated (*Glick and Luini, 2011*). Exchange mechanisms that do not establish stable connections able to relax concentration gradient between compartments by diffusion could be included as extension of our vesicle exchange mechanism (at the expense of introducing additional parameters to characterise their composition and size-dependent nucleation, scission and fusion rates). Stable Golgi tubulation without tubule detachment is observed under brefeldin A treatment, which prevent the membrane association of COPI coat. This eventually leads to the disappearance of the Golgi by quick resorption into the ER whenever a tubular connection is established between them, in a way suggesting a purely physical (tension-driven) mechanism (*Sciaky et al., 1997*). According to our model, preventing COPI vesicle budding should lead to larger cisternae, as discussed above. The brefeldin A phenotype can be reconciled with our framework by invoking the fact that larger cisternae should have a higher surface to volume ratio and hence a lower membrane tension, decreasing the force required to nucleate membrane tubules and increasing the probability of direct tubular connection with the ER. Such tension effects are clearly beyond the scope of the present model, but this phenotype could be qualitatively reproduced by invoking an increase of the ER fusion parameter $\alpha_{\mathrm{ER}}$ with the size of the compartment due to membrane mechanics considerations.

The model does not account for the possibility of lateral partitioning between processing and exporting domains within compartments which could play an important role in regulating cisterna composition and cargo processing (*Patterson et al., 2008*). A detailed kinetic model of organisation based on this concept and supported by extensive quantitative live cell imaging experiments is proposed in *Patterson et al., 2008* to account for the distinct composition of the different Golgi cisterna and the kinetics of cargo transport. This rapid partitioning model (RPM) is very different in scope with the model we propose here, since it focuses on the transport of lipids and cargo molecules in a pre-established Golgi apparatus with a prescribed structure, while the present model describes de-novo Golgi formation and maintenance, through self-organised fusion and scission mechanisms between dynamic compartments. The two models share important similarities, such as the fact that no directionality is assumed a priori for intra-cisternal vesicular transport, and that export is allowed from every cisterna, which is crucial to reproduce the exponential kinetics observed in experiments. The RPM is much more precise in terms of the microscopic description of the biochemical processes, and consequently involves many parameters. Combining our simple model of Golgi self-organisation with the detailed description given by the RPM for the kinetics of lipid, enzymes and cargo within the Golgi is the logical next step to push forward our understanding of Golgi self-organisation.

In summary, we have analyzed a model of self-organisation and transport in cellular organelles based on a limited number of kinetic steps allowing for the generation and biochemical conversion of compartments, and vesicular exchange between them. Although we kept the complexity of individual steps to a minimum, our model gives rise to a rich diversity of phenotypes depending on the parameter values, and reproduces the effect of a number of specific protein mutations. We identify the concomitance of a structural transition (from large and mixed to small and pure sub-compartments) and a dynamical transition (from retrograde to anterograde vesicular exchange). Within our model, anterograde vesicular transport is accompanied by many spurious events of vesicle back-fusion. For very high budding rates, compartments are very pure and anterograde transport is dominated by vesicles undergoing biochemical conversion *after* budding from a compartment. In our model, such vesicle biochemical conversion events, which have been described in the secretory pathway as a way to direct vesicular traffic and to prevent vesicles back-fusion (*Lord et al., 2011*), only occur when compartments are very small (high budding rate). Adding composition-dependent feedback involving specific molecular actors to tune budding and fusion rates - for instance to reduce or prevent the budding of the majority species - will displace the purity transition to lower budding rates and thus extend the regime of anterograde transport to larger sub-compartment steady-state sizes. Cisternal stabilizers like GRASP or vesicles careers like Golgins, known to already be present in the ancestor of eukaryotes (*Barlow et al., 2018*), are obvious candidates. Although more complex models, and in particular the inclusion of spatial dependencies, are surely relevant to dynamics of the organelle, the fundamental relationship between kinetics, structure and transport highlighted by our model is a universal feature of the interplay between biochemical conversion and vesicular exchange in cellular organelles.

## Acknowledgements

We acknowledge stimulating discussions with Bruno Goud, Cathy Jackson, Grégory Lavieu, Franck Perez and Stéphanie Miserey-Lenkei, and critical reading of the manuscript by Matthew S Turner.

## Additional information

### Competing interests

Pierre Sens: Reviewing Editor, eLife. The other authors declare that no competing interests exist.

### Funding

| Funder | Grant reference number | Author |
| --- | --- | --- |
| Université Paris Descartes | Ecole Doctorale 474 FIRE - Programme Bettencourt | Jean-Patrick Vrel |

The funders had no role in study design, data collection and interpretation, or the decision to submit the work for publication.

## Author contributions
Quentin Vagne, Conceptualization, Software, Formal analysis, Funding acquisition, Validation, Investigation, Visualization, Methodology; Jean-Patrick Vrel, Conceptualization, Data curation, Software, Formal analysis, Funding acquisition, Validation, Investigation, Visualization, Methodology; Pierre Sens, Conceptualization, Resources, Formal analysis, Supervision, Funding acquisition, Validation, Investigation, Visualization, Methodology, Project administration

## Author ORCIDs
Pierre Sens https://orcid.org/0000-0003-4523-3791

## Decision letter and Author response
Decision letter https://doi.org/10.7554/eLife.47318.sa1
Author response https://doi.org/10.7554/eLife.47318.sa2

# Additional files

## Supplementary files
• Source code 1. Simulation source code.

• Transparent reporting form

## Data availability
All data generated from computer simulations and analysed during this study are included in the manuscript and supporting files. Source code has been provided.

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

**Appendix 1**

## Detailed model

### General overview

The nomenclature we use in this section is presented in the *Appendix 1—tables 1–3*. We propose a coarse-grained, discrete model of the Golgi Apparatus. The smallest element is a vesicle, which defines the unit size of the system. Other compartments are assemblies of fused vesicles; their size is an integer corresponding to an equivalent number of vesicles that have fused to build this compartment. Each unit of membrane is of either *cis*, *medial* or *trans*-identity. A vesicle has a unique identity, and a compartment of size $n$ is a assembly of $n$ discrete membrane patches of different identities.

**Appendix 1—table 1.** System's parameters.

| | |
|---|---|
| $K_b$ | **Budding rate per patches of membrane in a compartment** |
| $K_m$ | biochemical conversion rate for each patch of membrane (*cis→medial→trans*) |
| $K_f$ | Maximum fusion rate between two compartments |
| $J$ | Injection rate of *cis*-vesicles from the ER |
| $k_b$ | Budding rate normalized by the fusion rate ($K_b/K_f$) |
| $k_m$ | biochemical conversion rate normalized by the fusion rate ($K_m/K_f$) |
| $j$ | Injection rate normalized by the fusion rate ($J/K_f$) |
| $\alpha$ | Fraction of active species (driving homotypic fusion) in the ER (*cis*) or the TGN (*trans*) |
| $\alpha_i$ | When different between boundaries, $\alpha$ in the boundary $i$ ($i$ equals ER or TGN) |

**Appendix 1—table 2.** Compartments' description.

| | |
|---|---|
| $\phi_i$ | **Fraction of a given species $i$ (*cis*, *medial* or *trans*) in a compartment** |
| $\vec{C}$ | Composition of a compartment, whose components are the fractions $\phi_i$ |
| $n$ | Size of a compartment, (i.e. number of membrane patches in the compartment) |
| $n_i$ | Number of membrane patches of the $i$ identity, in a compartment |
| $n_{id}$ | Number of different identities (from 1 to 3) in a single a compartment |
| $n_v$ | Number of vesicles surrounding a compartment that share its composition |
| $P$ | Purity of a compartment |
| $J_{b,i}$ | Budding flux of vesicles decorated by the identity $i$, from a compartment |

**Appendix 1—table 3.** Steady-state description.

| | |
|---|---|
| $c_n$ | **Size distribution of compartments in the system** |
| $n_c$ | Cutoff in the size distribution, namely the typical size of big compartments |
| $N$ | Total number of membrane patches, in all compartments, in the steady-state system |
| $N_i$ | Total number of patches of the $i$ species (*cis*, *medial* or *trans*) |

We let the system self-organizes between two boundaries, the ER and the TGN, that are two biological boundaries of the Golgi (*Presley et al., 1997*; *Klemm et al., 2009*). This theoretical Golgi is self-organized as we do not impose any topological or functional constraints, such as the number of compartments or the directionality of vesicular trafficking. The evolution of the system is only dictated by three mechanisms that are budding, fusion and biochemical conversion (short description in the main text, complete description bellow). In order to keep the model as simple as possible, we neglect all mechanical and spatial dependencies. This means that we do not take into account motion of compartments in space, or mechanical properties of the membranes. Any two compartments have the

possibility to fuse together, with a rate that depends on their relative composition (see below), but not on their size. An alternative model where compartments cannot fuse with one another is presented in Appendix 8.

### Biochemical conversion

In the Golgi, it is known that cargo (*Kelly, 1985*) and cisternae (*Day et al., 2013*) undergo biochemical modifications over time. Changes in membrane identity are driven by maturation cascades of proteins such Rab GTPases (*Suda et al., 2013*). These biochemical conversions can be directly observed using live microscopy, and have been best characterized in the yeast Golgi (*Matsuura-Tokita et al., 2006*). Biochemical conversion of membrane identity is a complex and possibly multi-step process involving different kinds of enzymes. As we are mostly interested in the interplay between biochemical conversion (which mixes different identities within a single compartment) and sorting mechanisms, we coarse-grain the biochemical conversion events into simple, one step, stochastic processes: *cis→medial* and *medial→trans*. To simplify the model, we choose the same rate $K_m$ for both conversions. Each patch of membrane (in a vesicle or a compartment, and whatever the surrounding patches are) has the same transition rate.

### Fusion

Intracellular transport strongly relies on vesicular trafficking (*Takamori et al., 2006*). This involves at least two steps: the budding of a vesicle from a donor compartment (described in the next section), and its fusion with a receiving compartment. The fusion process itself requires close proximity between the receiving compartment and the vesicle, followed by the pairing of fusion actors resulting in membrane fusion. In the current model, the rate of encounter of two compartments is constant, and equal to $K_f$, while the actual fusion event depends on the biochemical composition of the fusing compartments. In vivo, some of the key proteins involved in the fusion process, such as the SNAREs or tethering proteins are known to closely interact with membrane markers like Rab proteins (*Cai et al., 2007*). This is thought to accelerate fusion events between compartments of similar biochemical identities and decrease it between compartments of different identities, a process often called homotypic fusion (*Marra et al., 2007*). To take this into account, fusion is modulated by the probability that both compartments exhibit the same identity at the contact site, corresponding to *Equation 1* in the main text.

The fusion rate is maximum ($K_f$) for two identical compartments, and vanishes between two compartments with no common identity. The exchange of vesicles between compartments of different identities (*Glick and Luini, 2011*), or the fusion of compartments of different identities (*Marsh et al., 2004*), are sometimes regarded as heterotypic fusion. Such processes are allowed within our homotypic fusion model, provided the different compartments share at least some patches of similar identities.

### Budding

Budding is the formation of a new vesicles from a large compartment. We assume that each budding vesicle is composed of a single membrane identity, which is consistent with the high specificity of the in vivo budding machinery (*Bonifacino and Glick, 2004*). In our model, the flux of vesicles budding from a compartment depends on the size and composition of the compartment. Compartments smaller than 2 cannot bud a vesicle. Compartments of size 2 can split into two vesicles. For larger compartments, we consider a general budding flux for vesicles of identity $i$ (for $i = cis$, *medial* and *trans*) of the form:

$$J_{b,i} = K_b \times n \times f(\phi_i) \tag{10}$$

with $n$ the size of the compartment', $K_b$ a constant and $f(\phi_i)$ a function of the composition of the compartment.

One expects the budding flux to follow a Michaelis-Menten kinetics; linear with the number of patches of a given identity ($f(\phi) = \phi$) at small concentration, and saturating to a constant at high concentration (*Vagne and Sens, 2018b*). This scenario is consistent with the fact that budding proteins interact very dynamically with the membrane, attaching and detaching multiple times before budding a vesicle (*Hirschberg et al., 1998*). Previous theoretical works suggest that vesicular sorting of different membrane species is more efficient in the saturated regime (*Dmitrieff and Sens, 2011*). Since one of the main features of the Golgi is to segregate different biochemical species into different compartments, we assume a very high affinity between membrane patches and their budding partners, setting $f(\phi_i) = 1$ if $\phi_i > 0$ and $f(\phi_i = 0) = 0$ (corresponding to *Equation 2* of the main text). The budding flux for a given identity thus depends on the total size of the compartment rather than on the amount of that particular identity. This leads to a total budding flux $K_b \times n \times n_{id}$ for a compartment carrying $n_{id}$ different identities. The results of simulations with this linear budding scheme are discussed in Appendix 8.

### Exits

In cells, the Golgi is placed between two intra-cellular structures that are the ER and the TGN (*Presley et al., 1997*; *Klemm et al., 2009*). Fluxes of material leaving the Golgi thus include retrograde fluxes toward the ER, carrying immature components such as *cis* and *medial*-Golgi enzymes or recycling ER enzymes, and anterograde fluxes toward the TGN of mature components, such as processed cargo that properly underwent all their post-translational maturation steps (*Boncompain and Perez, 2013*). The exiting fluxes are accounted for by allowing the different compartments in the system to fuse with the boundaries. All compartments, from vesicles to the largest ones, can fuse homotypically with the ER or the TGN to exit the system. Thus, these boundaries are modelled as stable compartments, containing a fraction $\alpha_{ER}$ of *cis* components for the ER and $\alpha_{TGN}$ of *trans* components for the TGN. These fraction are simply written $\alpha$ when they are assumed to be the same. This allows immature (*cis*) compartments to undergo retrograde exit, and mature (*trans*) components to undergo anterograde exit. In the main text we focus on the case $\alpha = 1$. Lower values of $\alpha$ reduce fusion with the boundaries and increase the residence time of components in the system. This increases the average size of compartments, and increases fluctuations in the system, as large compartments exiting the system induce large fluctuations in the instantaneous size and composition of the system. The impact of $\alpha$ is studied in some details in Appendix 7.

### Injection of components

#### Influx to the Golgi

The Golgi receives material from different compartments like the endosomal network, lyzosomes, etc... (*Boncompain and Perez, 2013*). As we are primarily interested in characterizing the relationship between the structure and dynamics of the Golgi, and in relating these to the rates of individual biochemical conversion and transport processes, we focus here on the secretory role of the Golgi, and only account for the incoming flux of immature components coming from the ER. We define a rate $J$ of injection of *cis*-vesicles in the system. As the parameters of the system are varied, the injection rate is varied as well in order to keep the total system's size to an almost constant value. This constraint is only approximately enforced using *Equation 6*, which is only strictly valid when all compartments are pure.

The impact of the system's size on the structure of the Golgi is shown on *Appendix 1—figure 1A*. Increasing the total size increases the number of compartments and hence the total fusion flux between compartments. Consequently, compartments are larger in larger systems, and tend to be (slightly) less pure, as fusion increases mixing. In the main text, we restrict ourselves to a system size of $N = 300$. One should remember that *Equation 6*, used

to predict the system's size $N$, assumes that compartments are pure (perfectly sorted). *Appendix 1—figure 1B*, shows that this assumption fails to predict $N$ for systems where the budding rate is low compared to the fusion rate (highly interacting compartments). In this regime, compartments have a great probability to fuse together, creating hybrid *cis/medial/trans*-structures. This makes *medial*-patches sensitive to the interaction with the ER and the TGN, and creates an exit flux for these patches that is not observed in a pure regime. Such systems exhibit a larger exit flux and thus a lower $N$. Note this is only true for biochemical conversion rates lower than the fusion rate, as the system is saturated with *trans*-patches for high $k_m$ (see *Equation 4*).

## Transport of passive cargo-proteins

The direction of vesicular transport is assessed by following the transport of passive cargo injected from the ER as part of incoming *cis*-vesicles. Cargo molecules are passive in the sense that their probability of joining a vesicle is insensitive to the vesicle membrane identity: when a compartment of size $n$ buds a vesicle, each cargo in this compartment has a probability $1/n$ to join the budding vesicle. In the simulation, the number of cargo molecules in the system is kept to a fixed number (typically 20) by injecting a new one each time one leaves the system trough fusion with the boundaries.

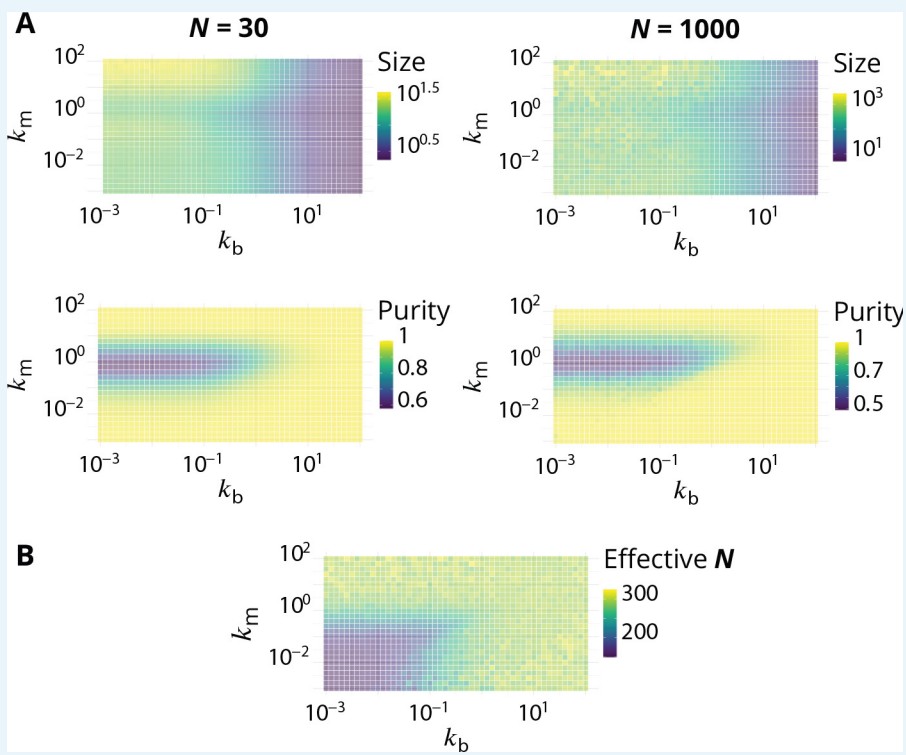

**Appendix 1—figure 1.** Link between the total size of the Golgi and its steady-state organisation. (**A**) Size dependence of the steady-state organisation of the Golgi (to be compared with *Figure 2* in the main text). We look at two different systems that only differ in term of the average total size $N$, that equals 30 or 1000. Typical size of compartments and average system purity are shown as a function of $k_m$ and $k_b$. In both cases $\alpha = 1$ (see Appendix 1). (**B**) Average steady-state total size $N$, as a function of $k_m$ and $k_b$, for simulation in which the influx is set to have $N \simeq 300$ (using *Equation 6*).

## Gillespie algorithm and software implementation

All rates in the system are normalized by the fusion rate: $k_b = K_b/K_f$, $k_m = K_m/K_f$ and $j = J/K_f$. The dynamics of the system can be exactly simulated (to yield the correct trajectories, stochastic noise, etc...) using a Gillespie algorithm (*Gillespie, 1977*), also known

as stochastic simulation algorithm. After choosing the initial state of the system (see below), the algorithm is implemented as follows:

- The rates of all possible events (be it the input of a new vesicle in the system, the fusion between two compartments or between a compartment and one boundary, the budding of a vesicle, or the biochemical conversion of a membrane patch) are calculated. The time interval $\delta t$ for the next reaction to occur is randomly picked from an exponential distribution with mean $1/R_{\mathrm{TOT}}$, where $R_{\mathrm{TOT}}$ is the sum of the reaction rates of all possible reactions. The reaction that actually occurs is randomly picked with a probability equal to the rate of this reaction, normalized by $R_{\mathrm{TOT}}$.
- The state of the system is modified according to the picked reaction. The rates of all events in the modified system are calculated, and the current time of the simulation is incremented by $\delta t$.
- The program comes back to step 1 IF the predefined maximum number of iterations has not been reached, AND there is at least one reaction that has a non-zero rate ($R_{\mathrm{TOT}} \neq 0$).

With the Gillespie algorithm, there is no direct dependency between the number of simulation steps and the actual time that is simulated. The physical time between two consecutive steps decreases with increasing reaction rates. Since we are interested by steady-state quantities, we average over a fixed number of steps rather than a given physical time. As shown in Appendix C, the steady-state of the system is reached after a physical time of order $1/K_{\mathrm{f}}$. Depending on the parameters, the steady-state is reached after between $10^3$ and $10^6$ simulation steps. To characterise the steady-state, we disregards the transient regime and data are recorded after $10^6$ steps. The full simulation typically last at least $10^7$ steps. This arbitrary amount is a good compromise between computation time and the need to accumulate sufficient amount of data to obtain enough statistics on all the measured quantities. In practice, the time needed to reach steady-state can be shortened by starting the simulation from a vesicular system with the predicted amount of *cis*, *medial* and *trans*-species (with **Equation 4**).

Because simulation steps have different durations, one should be careful when computing time averages. Two different approaches can be used. Either we weight each configuration by the duration between two consecutive time-steps, or we re-sample data to get a fixed time-step between observations, $\Delta t$. We checked that both procedures give the same results, and adopted the second approach for the analysis. As the steady-state of the system is reached after a physical time of order $1/K_{\mathrm{f}}$, we take $\Delta t = 1/K_{\mathrm{f}}$.

## Appendix 2

### Quantification and statistical analysis

#### Computation of the purity

The purity of a compartment $P$ is defined such that its value 0 is for a perfectly mixed compartment containing the same amount of *cis*, *medial*, and *trans*-species, and it is equal to 1 for a pure compartment containing a single species. With $\phi_i$ the fraction of the species $i$ in the compartment, the purity $P$ is defined as:

$$P = \sqrt{\frac{3}{2}\sum_i (\phi_i - 1/3)^2} \tag{11}$$

On a triangular composition space where each corner corresponds to a pure compartment, $P$ is the distance from the center of the triangle, see **Appendix 2—figure 1**. When we show snapshots of the system, each compartment is represented as a sphere whose area is proportional to the compartment size (defined as an equivalent number of vesicles). The composition of a compartment is represented as a color following the color code shown in **Appendix 2—figure 1**. The purity of the system is a global average (over time and over all compartments) of the purity of individual compartments, weighted by their sizes and ignoring vesicles (that are always pure as they are composed of one unique identity).

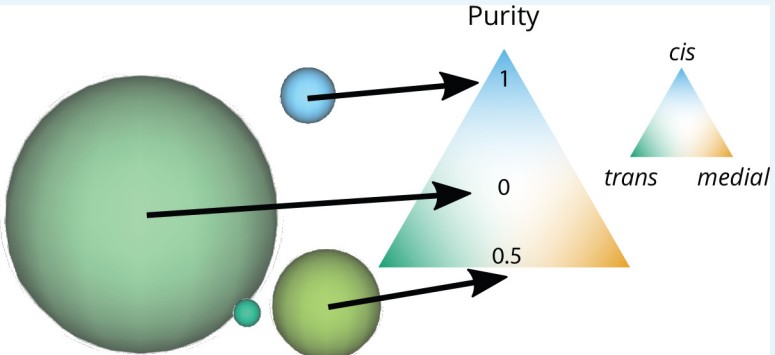

**Appendix 2—figure 1.** Graphical representation of the size and purity of compartments. Each compartment is represented as a sphere with an area equal to the area of a vesicle (the smallest elements in this snapshot) times the number of vesicles that fused together to create the compartment. The color of the compartment reflects its composition, according to the color code on the triangular composition phase space: 100% *cis* at the top, *medial* at the bottom right, *trans* at the bottom left. The purity of each of these compartments is its distance from a perfectly mixed compartment at the center of the triangle (middle arrow). This distance is normalized so that it equals 1 for a perfectly pure compartment (bottom arrow).

#### Computation of the typical size

Among other mechanisms, this theoretical organelle self-organizes by budding and fusion of components. In that sense, it is a scission-aggregation structure which should follow the laws dictating the behavior of this class of systems. One of these laws is the fact that the size distribution for small compartments should follow a power-law. Because of the scission, compartments cannot grow indefinitely and the power-law ends by an exponential cutoff (**Turner et al., 2005**; **Vagne et al., 2015**). Thus, the size distribution $c_n$ of compartments of size $n$ should, on a first approximation, follow this general formulation:

$$c_n \sim \mathrm{Const}\; n^{-\beta} \exp\left(-\frac{n}{n_c}\right) \tag{12}$$

where $n_c$ is the cut-off size and $\beta$ is an exponent that has been calculated to be $\beta = 3/2$ in a similar system (**Turner et al., 2005**). There are multiple ways to characterize the average size of the distribution, using ratios of moments $k+1$ over $k$ of the distribution:

$$\langle n \rangle_k \equiv \frac{\sum_{n=1}^{\infty} n^{k+1} c_n}{\sum_{n=1}^{\infty} n^k c_n} \tag{13}$$

It turns out that for $n_c \gg 1$, $\sum_{n=1}^{\infty} n^{k-\beta} e^{-n/n_c} \sim 1$ if $1+k-\beta < 0$ and $\sum_{n=1}^{\infty} n^{k-\beta} e^{-n/n_c} \sim n_c^{1+k-\beta}$ if $1+k-\beta > 0$. In order to have $\langle n \rangle_k \sim n_c$, we need to choose the exponent $k$ such that $k > \beta - 1 = 1/2$ in the present case. In the main text, we adopt $k=1$ and define the characteristic size of the distribution as:

$$\langle n \rangle \equiv \frac{\sum_{n=1}^{\infty} n^2 c_n}{\sum_{n=1}^{\infty} n c_n} \xrightarrow[n_c \gg 1]{} \frac{n_c}{2} \tag{14}$$

As shown in **Figure 2A** of the main text, the calculated average is in good agreement with simulations' data.

## Computation of the mean enrichment

To discriminate between the two models of Golgi's dynamics that we can find in the literature, we need to quantify whether the vesicular transport is anterograde or retrograde. Indeed, the 'Vesicular transport' model assumes that cargo is transported sequentially from *cis* to *medial* to *trans*-compartments while resident enzymes remain in place, meaning that the vesicular flux is anterograde. On the other hand, 'Cisternal maturation' models assume that cargo remain inside cisternae, and resident enzymes are recycled, thus requiring a retrograde vesicular flux. One way to measure this flux in our simulations is to follow passive cargo molecules and quantify whether they move to more or less mature compartments, as they get carried by vesicles. To do so, we record all events that affect cargo molecules. Every time a compartment buds a cargo, we store the composition of this compartment and compare it to the one in which the vesicle later fuses. Both compositions are a vector $\vec{C}$, with three components that are the fraction $\phi_i$ ($i$ equals *cis*, *medial* and *trans*) of the donor and acceptor compartments. Defining $\vec{C}_d$ the composition of the donor compartment, and $\vec{C}_a$ the composition of the acceptor, we can compute the enrichment $\vec{E}$ as:

$$\vec{E} = \vec{C}_a - \vec{C}_d \tag{15}$$

The sum of both $\vec{C}$ components equals 1 (as they are fractions of each identities), and the sum of $\vec{E}$ components equals $\vec{0}$, which simply means one cannot gain in fraction of any identity without loosing the same amount of the others. We can now compute $\langle \vec{E} \rangle$ to calculate the mean enrichment in cis, *medial* and *trans*-species, between a budding event and the next fusion event.

However, and because of back fusion events (fusion into the same compartment that previously budded the vesicle), $\langle \vec{E} \rangle$ components can be close to 0. This is particularly true for pure, sorted systems. In that case, a budded vesicle has the same identity as its donor compartment, and thus has a great probability to fuse back with the same compartment or one of very similar composition. That is why, non-treated data do not allow to discriminate well between an anterograde and a retrograde regime when the purity (and thus $k_b$) is high (**Appendix 6—figure 1**). As we are primarily interested in the sign of these vectors' components, we normalize $\vec{E}$ using the $L_1$ norm ($\sum_i |E_i| = 1$).

## Computation of the vesicular flux vector field

The vesicular flux discussed in the previous section can be directly displayed on the triangle of composition we introduced in *Appendix 2—figure 1*. Instead of computing $\langle \vec{E} \rangle$ for all $\vec{E}$, we can bin data with respect to the donor compartment composition (typically triangular bins of size $\Delta\phi = 0.1$) and calculate the average enrichment for each bin of donor compartments. To get rid of back fusion effects in this quantification, we remove the vectors $\vec{E}$ for which all components are smaller (in absolute values) than the binning mesh-grid. For each binned composition we can now compute the mean enrichment vector, and plot this vector on the 2D triangular composition space. To emphasize the dominant fluxes in this vector field, the opacity of vectors is set proportionally to the flux of transported cargo per unit time (normalized by the total amount of cargo-proteins in the system). This can be seen on *Figure 4* - main text.

## Appendix 3

# Analytical model for the purity of compartments

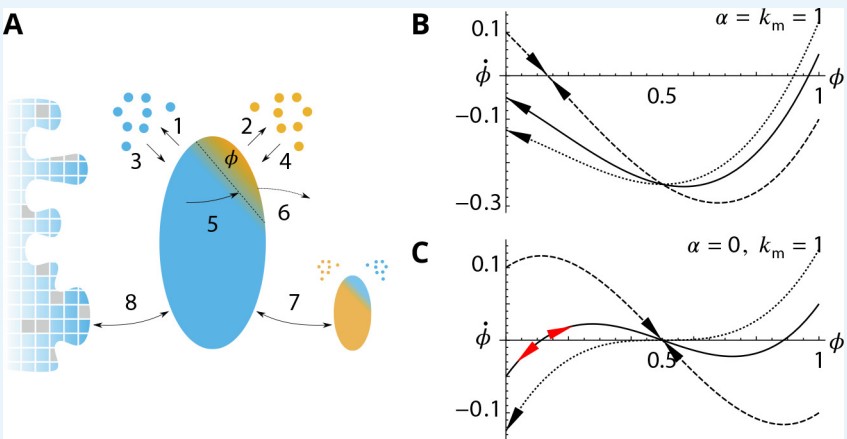

**Appendix 3—figure 1.** Analytical description of the *cis*-compartment purity. Related to *Figure 2* - main text. (**A**) Analytical model of the steady-state contamination by a *medial*-species (orange) of a *cis*-compartment (blue). The fraction occupied by the *medial*-species is named $\phi$. The eight events (1–8) impacting $\phi$ are listed in the text. (**B-C**). Time variation $\dot{\phi}$ as a function of $\phi$, with $k_{\mathrm{m}} = 1$ and for different $k_{\mathrm{b}}$ (0.9 dashed line, 1.05 solid line, and $1 + 1/8$ dotted line). $\phi$ has a stable value when $\partial_\phi \dot{\phi} < 0$ and $\dot{\phi} = 0$ or $\dot{\phi}|_{\phi=0} < 0$ (black arrows). It is potentially unstable when $\partial_\phi \dot{\phi} > 0$ and $\dot{\phi}_{\phi \sim 0} = 0$ (red arrows). (**B**) For $\alpha = 1$ : $k_{\mathrm{b}} = 0.9$ (dashed) exhibits a stable fixed-point for $\phi > 0$ but with $\phi < 1/2$ as $\alpha > 0$; $k_{\mathrm{b}} = 1.05$ (solid) and $k_{\mathrm{b}} = 1 + 1/8$ (dotted) exhibit a stable concentration for $\phi = 0$. (**C**) For $\alpha = 0$; $k_{\mathrm{b}} = 0.9$ (dashed) exhibits a stable fixed-point for $\phi = 1/2$; $k_{\mathrm{b}} = 1.05$ (solid) exhibits an unstable fixed-point near $\phi = 0$; $k\mathrm{b} = 1 + 1/8$ (dotted) does not display this unstable fixed-point as $k_{\mathrm{b}} \geq k_{\mathrm{m}} + 1/8$ (see text).

We want to build a simple model to explain how the purity of the system is affected by the parameters. This is challenging as all events (budding, fusion and biochemical conversion) influence the purity of a compartment. We seek to determine the purity transition, namely the values of $k_{\mathrm{b}}$ and $k_{\mathrm{m}}$ for which a pure system becomes impure.

We make the following approximations, based on the assumption that the system is almost pure: (i) *trans*-compartments are pure as they cannot be contaminated by biochemical conversion. (ii) *cis* and *medial*-compartments share the same purity. (iii) The total amount of the different species $N_{cis}$, $N_{medial}$ and $N_{trans}$ follows the steady-state repartition of *cis*, *medial* and *trans*-species in a pure limit, given by **Equation 4**. Most of the components of the system are within compartments.

The purity of a compartment - defined in the main text - is $P = \sqrt{\left( 3/2 \sum_i (\phi_i - 1/3)^2 \right)}$. We consider that *cis*-compartments (*medial*-compartments) are contaminated by an average fraction $\langle \phi \rangle \ll 1$ of *medial*-patches (*trans*-patches), while *trans*-compartments are pure. The purity of *cis* and *medial*-compartments is thus $P_{cis,medial} = \sqrt{1 - 3\langle \phi \rangle (1 - \langle \phi \rangle)}$ and $P_{trans} = 1$. Weighting these with the fraction of the different species in the system (from **Equation 4**) gives

$$\langle P \rangle = \frac{2\alpha_{\mathrm{TGN}} \sqrt{1 - 3\langle \phi \rangle (1 - \langle \phi \rangle)} + k_{\mathrm{m}}}{2\alpha_{\mathrm{TGN}} + k_{\mathrm{m}}} \tag{16}$$

To compute the average contamination $\langle \phi \rangle$, we consider the different events that affect the purity of a *cis*-compartment of size $n$ contaminated by a fraction $\phi$ of *medial*-patches,

surrounded by a number $n_v$ of *cis*-vesicles, and a number $n_v'$ of *medial*-vesicles. These events are sketched in **Appendix 3—figure 1**, and listed below:

1.
Budding of *cis*-patches (increases $\phi$)

2.
Budding of *medial*-patches (decreases $\phi$)

3.
Fusion of *cis*-vesicles (decreases $\phi$)

4.
Fusion of *medial*-vesicles (increases $\phi$)

5.
Conversion of *cis*-patches (increases $\phi$)

6.
Conversion of *medial*-patches (decreases $\phi$)

7.
Fusion with a *medial*-compartment ($\phi \rightarrow \frac{1}{2}$)

8.
Fusion with the ER ($\phi \rightarrow 0$)

The rate $p(i)$ (normalized by the fusion rate) for each mechanism, and the extent $\delta\phi$ to which it affects $\phi$, are:

$$
\begin{aligned}
p(1) &= nk_b , & \delta\phi(1) &= \tfrac{n\phi}{n-1} - \phi ; & p(2) &= nk_b , & \delta\phi(2) &= \tfrac{n\phi-1}{n-1} - \phi \\
p(3) &= n_v(1-\phi) , & \delta\phi(3) &= \tfrac{n\phi}{n+1} - \phi ; & p(4) &= n_v'\phi , & \delta\phi(4) &= \tfrac{n\phi+1}{n+1} - \phi \\
p(5) &= n(1-\phi)k_m , & \delta\phi(5) &= \tfrac{1}{n} ; & p(6) &= n\phi k_m , & \delta\phi(6) &= -\tfrac{1}{n} \\
p(7) &= \phi(1-\phi) , & \delta\phi(7) &= \tfrac{1}{2} - \phi ; & p(8) &= \alpha(1-\phi) , & \delta\phi(8) &= -\phi
\end{aligned}
$$

We can now compute the temporal variation of $\phi : \dot{\phi} = \sum p(i)\delta\phi(i)$. We concentrate on the limit of large compartments: $n \gg 1$. For simplicity, we also assume that $n \gg (n_v - n_v')$, so that the net contribution of vesicle fusion to the purity variation, which scales like $(n_v' - n_v)/n$, is negligible. Neglecting stochastic fluctuations, the mean-field evolution of the fraction of contaminating species is:

$$\dot{\phi} = (k_m - k_b)(1 - 2\phi) + \phi(1-\phi)(1/2 - \phi - \alpha) \tag{17}$$

Below, we discuss separately the case where $\alpha = 1$ (fast fusion with the ER boundary - discussed in the main text), and the case where $\alpha = 0$ (no exit through the ER - discussed in Appendix 7).

- $\alpha = 1$. A pure compartment; $\phi = 0$, is stable ($\dot{\phi}|_{\phi=0} < 0$) when $k_m < k_b$. If $k_m > k_b$, the compartment is contaminated by a stable fraction of *medial*-patches given by the single stable root of **Equation 17** (see **Appendix 3—figure 1B**). The purity transition lines shown in **Figure 2C** of the main text are obtained by inserting this root in **Equation 16**.
- $\alpha = 0$. If $k_b < k_m$, the single root of **Equation 17** is for $\phi = 1/2$ and is stable. The system is always impure with mixed *cis-medial* compartments. If $k_b > k_m$, pure compartments ($\phi = 0$) are stable at the mean-field level, but there exist an unstable fixed point for small values of $\phi$. Stochastic fluctuations may bring the system passed the unstable fixed point toward the stable mixed solution $\phi = 1/2$. For $k_b > k_m + \frac{1}{8}$, $\phi = 1/2$ is an unstable fixed point and the only stable solution is for $\phi = 0$ (see **Appendix 3—figure 1C**). The purity transition is thus with the range $k_m < k_b < k_m + \frac{1}{8}$.

A more precise characterization of the transition in the case $k_b > k_m$ requires to include stochastic effects. The different stochastic events are associated to very different fluctuations amplitudes. Budding and fusion of vesicles and biochemical conversion of membrane patches represent small fluctuations for a large compartment, while fusion between compartments and with the boundaries represent a very large alteration of the purity. In what follows, we propose a simplified treatment of this process, focusing on the case $\alpha = 0$ for simplicity, and neglecting the contribution of vesicle fusion as discussed above. For a compartment of size $n$ contaminated by $n_{medial}$ patches, we identify 2 types of mechanisms (depending on their amplitude). First, a smooth drift in the contamination due to biochemical conversion (with a rate $k_m * (n - n_{medial})$) and budding (with a rate $k_b n$). Second, a jump in the contamination due to fusion with the neighbor compartment (rate $n_{medial}(n - n_{medial})/n^2$). We compute below the mean contamination, assuming that the *cis*-compartment spends a time $\tau_{conta}$ undergoing small fluctuations $\delta\phi \simeq 0$ close to the pure state, before fusing with a *medial*-compartment of composition $1 - \delta\phi$. The resulting compartment with $\phi \simeq 1/2$ then spends a time $\tau_{deconta}$ undergoing a decontamination process, which depends on the balance between budding and biochemical conversion.

We first calculate the average value $\delta\phi$ of the contamination around $\phi = 0$ due to biochemical conversion and budding, disregarding inter-compartment fusion. The probability $\mathbb{P}(n_{medial})$ of finding $n_{medial}$ in the cisterna satisfies:

$$\begin{aligned}
\dot{\mathbb{P}}(n_{medial}) =\ & -\mathbb{P}(n_{medial})(k_m(n - n_{medial}) + k_b \times n) \\
& + \mathbb{P}(n_{medial} - 1)\, k_m(n - n_{medial} + 1) \\
& + \mathbb{P}(n_{medial} + 1)k_b \times n
\end{aligned} \tag{18}$$

In the limit $n \gg n_{medial}$, this becomes:

$$\dot{\mathbb{P}}(n_{medial})/n = -\mathbb{P}(n_{medial})(k_m + k_b) + \mathbb{P}(n_{medial} - 1)k_m + \mathbb{P}(n_{medial} + 1)k_b \tag{19}$$

The steady-state solution is

$$\mathbb{P}(n_{medial}) = (k_m/k_b)^{(n_{medial})} / \sum_{i=0}^{n/2}(k_m/k_b)^i = \frac{(k_b - k_m)(k_m/k_b)^{(n_{medial})}}{k_b - k_m(k_m/k_b)^{n/2}} \tag{20}$$

and the average contamination $\delta\phi$ due to budding and biochemical conversion is:

$$\delta\phi = \sum_{i=0}^{n/2} \frac{i}{n} \times \mathbb{P}(i) \simeq \frac{k_m}{n(k_b - k_m)}, \quad \text{(for } k_b > k_m) \tag{21}$$

These fluctuations in composition allow fusion between compartments, at a rate $\delta\phi(1 - \delta\phi) \simeq \delta\phi$ (for $\delta\phi \ll 1$). The average waiting time for an inter-compartment fusion event is thus $\tau_{conta} = 1/\delta\phi$.

After inter-compartment fusion, the compartment is at $\phi = 1/2$, which is an unstable fixed point according to *Equation 17*. Any small fluctuation in composition leads to a smooth decontamination that satisfies $\dot{\phi} = (k_m - k_b)(1 - 2\phi)$, disregarding inter-compartment fusion. The decontamination time $\tau_{deconta}$ and the average contamination during the process $\langle\phi\rangle_{deconta}$, can be estimated by integrating $\dot{\phi}$, from $\phi = 1/2 - 1/n$ to $\phi = 0$:

$$\tau_{deconta} = \frac{\log(n/2)}{2(k_b - k_m)}, \quad \langle\phi\rangle_{deconta} = \frac{2 + n[\log(n/2) - 1]}{2n\log(n/2)} \tag{22}$$

We can now calculate $\langle\phi\rangle$ as a temporal average of $\phi$:

$$\langle\phi\rangle = \frac{\tau_{conta} \times \delta\phi + \tau_{deconta} \times \langle\phi\rangle_{deconta}}{\tau_{conta} + \tau_{deconta}} \tag{23}$$

When $\log(n/2)$ is close to 1 (typically true for the range of $n$ that is interesting here, $n \sim 100$), $\langle\phi\rangle$ can be simplified to:

$$\langle \phi \rangle \simeq \frac{k_{\mathrm{m}} + 2k_{\mathrm{m}}(k_{\mathrm{b}} - k_{\mathrm{m}})}{2k_{\mathrm{m}} + 2n(k_{\mathrm{b}} - k_{\mathrm{m}})^2} \tag{24}$$

Injecting this result into **Equation 16** gives the approximate purity boundary for $\alpha_{\mathrm{ER}} = 0$:

$$\langle P \rangle \simeq 1 - \frac{3\alpha_{\mathrm{TGN}}}{2\alpha_{\mathrm{TGN}} + k_{\mathrm{m}}} \frac{k_{\mathrm{m}} + 2k_{\mathrm{m}}(k_{\mathrm{b}} - k_{\mathrm{m}})}{2k_{\mathrm{m}} + 2n(k_{\mathrm{b}} - k_{\mathrm{m}})^2} \tag{25}$$

Despite numerous simplifications, the model gives good predictions on the purity, both for $\alpha_{\mathrm{ER}} = \alpha_{\mathrm{TGN}} = 1$ (**Figure 2** - main text) and for $\alpha_{\mathrm{ER}} = 0$, $\alpha_{\mathrm{TGN}} = 1$ (**Appendix 7—figure 1**). Depending on the normalized biochemical conversion rate $k_{\mathrm{m}}$, we can discriminate three regions in the purity phase diagram:

- $k_{\mathrm{m}} \gg 1$, the system is saturated in trans-species and is thus always pure.
- $k_{\mathrm{m}} \sim 1$, compartment's contamination occurs by biochemical conversion, matured species are sorted by budding. The purity transition occurs for $k_{\mathrm{b}} \simeq k_{\mathrm{m}}$ (compartments are impure system if $k_{\mathrm{b}} < k_{\mathrm{m}}$).
- $k_{\mathrm{m}} \ll 1$, the system is saturated by *cis* and *medial*-compartments, and the transition of purity occurs for $k_{\mathrm{b}} \ll 1$: the fusion rate is dominant in this regime. Thus interaction with the boundaries or other compartments are crucial to understand the transition of purity. For $\alpha_{\mathrm{ER}} = 1$, fusion with the ER decontaminates the system, thus the transition occurs for $k_{\mathrm{b}} < k_{\mathrm{m}}$. The transition becomes independent of $k_{\mathrm{b}}$ for very small biochemical conversion rates, where the system is saturated in cis-species. For $\alpha_{\mathrm{ER}} = 0$, purifying the system by recycling compartments is not possible, and inter-compartment fusion decreases the purity, thus the transition of purity requires large values of the budding rate $k_{\mathrm{b}} > k_{\mathrm{m}}$, as given by **Equation 25**.

## Appendix 4

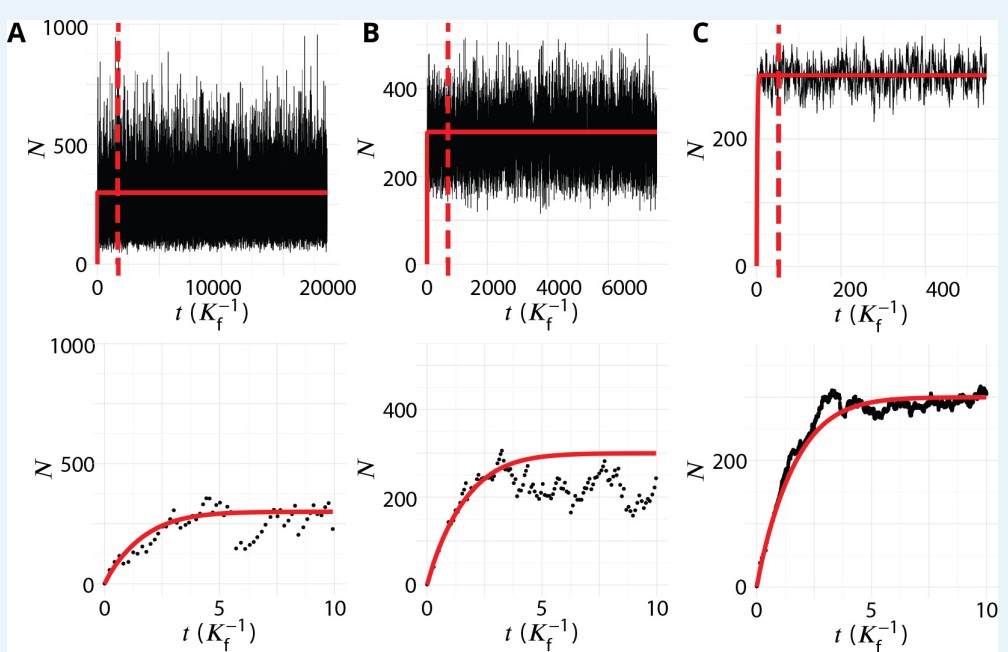

### de novo formation and cargo export kinetics

**Appendix 4—figure 1.** Transient regime towards the steady-state. System's size (total number of membrane patches in the system) as a function of time (in unit $1/K_f$) for three different values of the budding rate (A: $k_b = 10^{-2}$, B: $k_b = 1$, C: $k_b = 10^2$). The top row shows the full simulations, with a vertical dashed bar when the number of simulation steps reaches $10^6$. The steady-state data analysed and discussed in this paper are recorded after this time, and the full simulation typically lasted $10^7$ time-steps. The bottom row shows the evolution at early time (de-novo formation). The red lines are the analytical solution given by Appendix 4 *Equation 3*.

### Transient evolution toward the steady-state: De novo formation

The evolution of the system's size with time for different values of the dimensionless budding rate $k_b$ in shown in **Appendix 4—figure 1**. The organelle forms de-novo, and reaches the steady-state after a time of order a few $1/K_f$. The steady-state is reached under any parameter values, but the fluctuations around the steady-state depend on the parameter, as discussed in Appendix 5. The analytic calculation presented in **Equations 3** (red line in **Figure 1** above) is very accurate for high budding rate, but gives a 10% error for low budding rate.

### Transit kinetics of passive cargo

The cargo used as marker of vesicular transport directionality can be used to quantify the transit kinetics of a passive cargo across the self-organised organelle. **Appendix 4—figure 1** shows the amount of cargo present in the system as a function of time during a typical 'pulse-chase' experiment, in which a given number of cargo molecules is allowed to exit the E.R. at $t = 0$, by joining the vesicles constituting the influx $j$. The amount of cargo in the system initially increases, then decreases in a close-to-exponential fashion. The export kinetics resemble the one observed by iFRAP (Figure 2i of *Patterson et al., 2008*) or using the RUSH technics (Figure 3e of *Boncompain et al., 2012*).

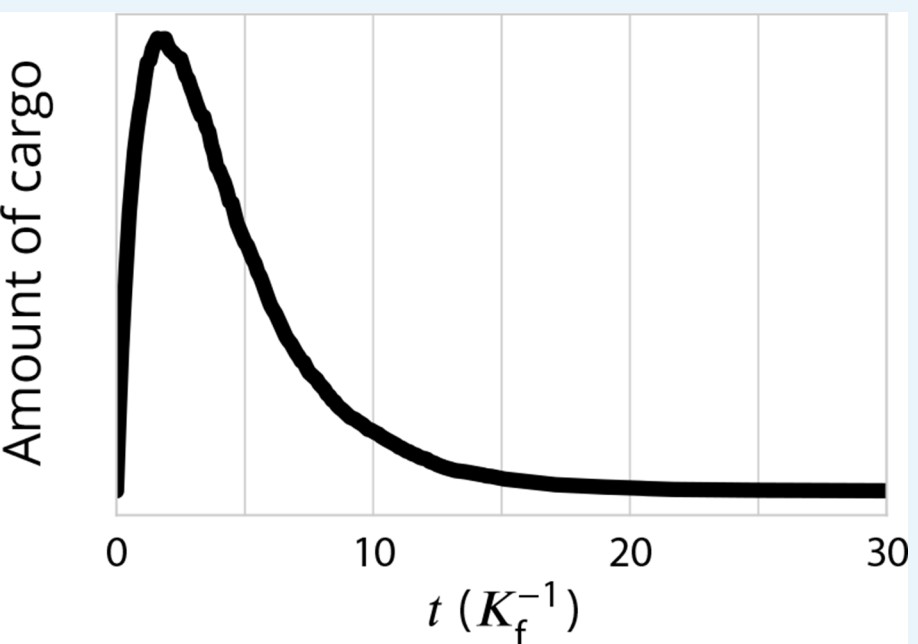

**Appendix 4—figure 2.** Exit kinetics of passive cargo. Simulated amount of cargo in the Golgi following a pulse-chase like simulation in which 1000 cargo molecules are trapped in the ER and allowed to be packaged into ER exiting vesicles at $t = 0$. This is expected to mimick the iFRAP experimental setup used in *Patterson et al., 2008* and to the RUSH setup used in *Boncompain et al., 2012*. The decay of the amount of cargo is close to a single-time exponential decay, is agreement with experimental observations for transiting cargo (Figure 2i of *Patterson et al., 2008* and Figure 3e in *Boncompain et al., 2012*). Parameters are $k_b = k_m = \alpha = 1$, corresponding to the 'sorted regime' (the triangle of *Figure 2*) which we expect to be the most reasonable for mammalian Golgi.

## Appendix 5

### Detailed organisation

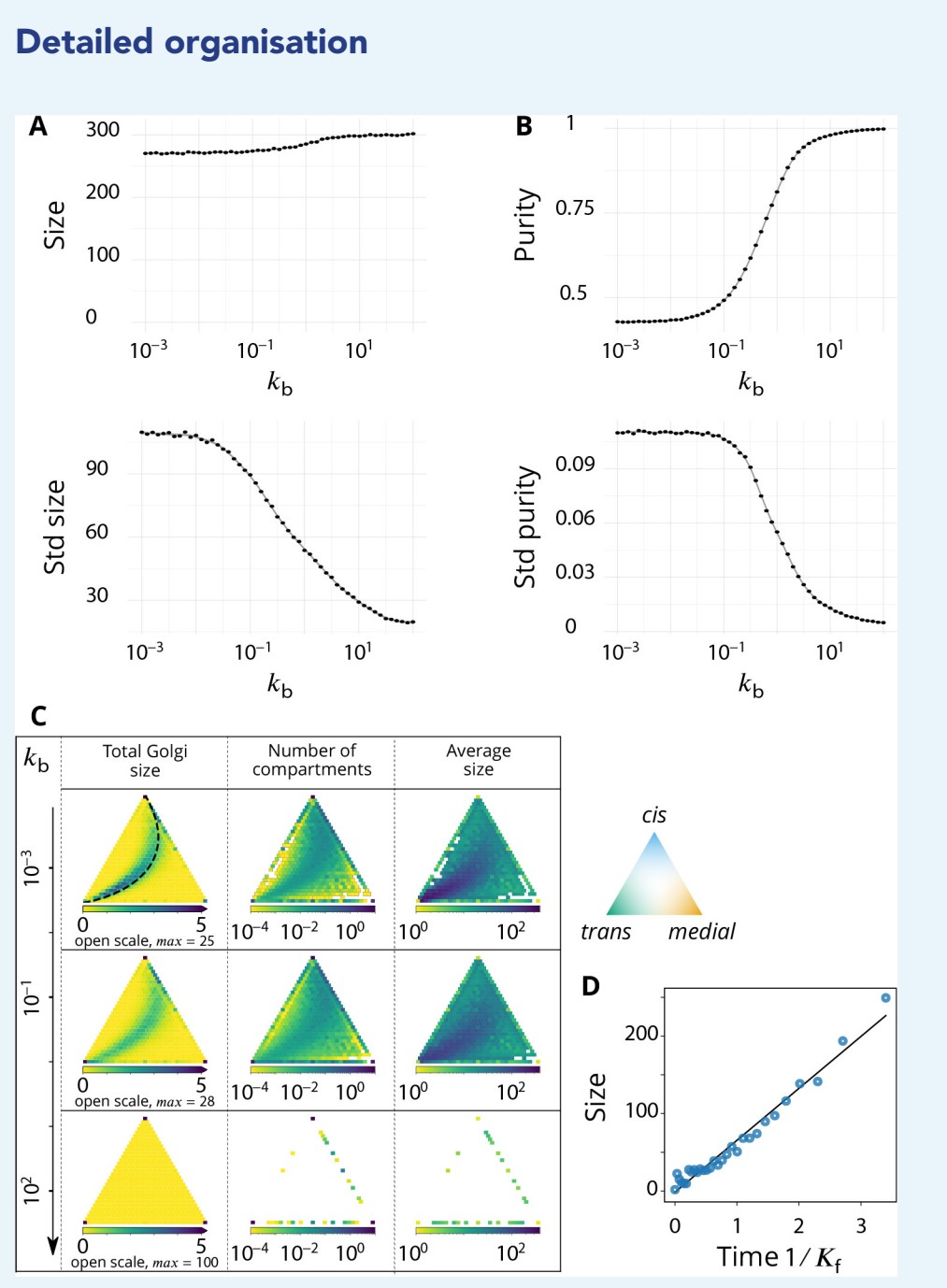

**Appendix 5—figure 1.** Steady-state composition and fluctuation of compartments. Related to *Figure 2* - main text. (**A–B**) Characterization of the temporal fluctuations of the system as a function of the budding rate $k_b$. (**A**) Average values of the system total size (top panel) and standard deviation of the size (bottom panel). (**B**) Average values of the system purity (top panel) and standard deviation of the purity (bottom panel). (**C**) Distribution of the total size, number of compartments and their average size, as a function of the composition of the compartments, shown in a triangle plot (see Appendix 2) for different values of the (normalized) budding rate $k_b$. The average size for a given composition is calculated as the average size divided by the average number of compartments. Plots are shown for four different values of $k_b$, with $k_m = \alpha = 1$. The dashed black line on the top, left panel (low $k_b$

regime) corresponds to the theoretical prediction of *Equation 9*. (**D**) The evolution of the compartment's size as a function of time, in the low $k_b = 10^{-3}$ limit, agrees with the linear prediction of *Equation 7*.

The fluctuations of the system around its steady-state are characterized by computing the temporal standard deviation of the total system's size and purity, shown in *Appendix 5— figure 1*. Fluctuations decrease as the budding rate $k_b$ increases, owing to the fact that compartments are on average smaller for larger budding rate, so that the removal of a compartment by fusion with the boundaries has a smaller impact on the state of the system. We note in *Appendix 5—figure 1*, that the total size of the system depends on the budding rate $k_b$, despite the fact that the influx $j$ is varied according to *Equation 6* to limit variations of the system's size. This is due to the fact that *Equation 6* is valid only in the limit of pure compartments (large $k_b$). For smaller values of the budding rate, *medial*-compartments contaminated by *cis* and *trans*-species and may thus exit the system by fusing with the boundaries, which decreases the total size for low $k_b$. For the regime we are interested in, namely $k_m \sim \alpha \sim 1$, this phenomenon is practically negligible, but its impact on the average size is more severe for lower $k_m$ (Appendix 1).

We also characterize the variation of the size of compartments and their abundance as a function of their composition. In the main text (*Figure 4*), variations of the total size (number of membrane patches) and the number of compartments is shown as a function of their composition, in the composition triangular space defined in Appendix 2. This is obtained by binning the composition space in such a way that the characterization is both relatively precise and statistically significant (binning $\phi$ with a mesh of $1/30$). The size and number of compartments are averaged over the simulation time (once the steady-state is reached) for each composition bin. These results are reproduced in *Appendix 5—figure 1*, where the average size of compartment, computed as the ratio of average size over average number of compartments, is also shown.

For low values of the budding rate $k_b$, both the size and the number of compartments follow the theoretical line obtained from the 'cisternal maturation' limit (*Equation 9*). Compartments also grow linearly in time, in agreement with the linear prediction of *Equation 7*. Their sizes are shown in *Appendix 5—figure 1*, as a function of their theoretical lifetime. The sizes are computed by measuring the average compartment size for each composition bin along the theoretical line. The compositions are then used to estimate the lifetime of compartments using *Equation 9*.

## Appendix 6

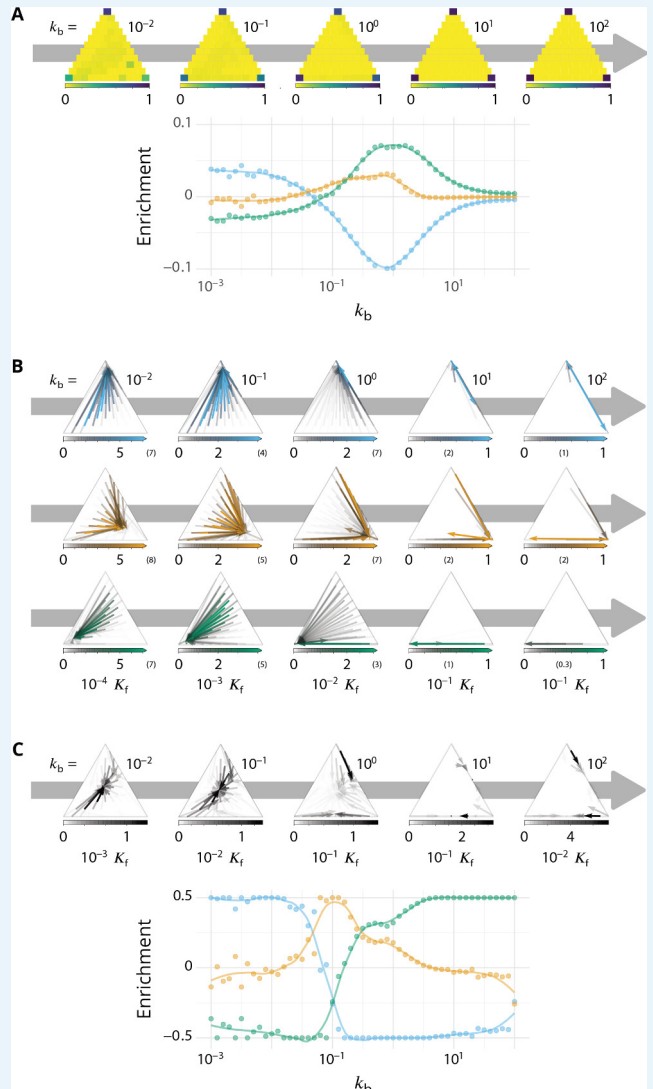

### Detailed characterization of vesicular transport

**Appendix 6—figure 1.** Detailed characterizations of the vesicular transport. Related to *Figures 3–4* - main text. $\alpha = k_m = 1$ for all panels. (**A**) Characterization of vesicle back-fusion (fusion of vesicles with the same compartment or with a compartment of comparable composition). Top panel: fraction of vesicular flux undergoing back-fusion, as a function of the composition of the budding compartments, for different values of the budding rate $k_b$. Bottom panel: non-normalized enrichment in cis (blue), *medial* (orange) and *trans*-species (green) seen by a passive cargo, between a budding and the next fusion event, as a function of $k_b$. (**B**) Vesicular fluxes of *cis* (blue), *medial* (orange) and *trans* (green) vesicles for different values of the budding rate $k_b$. The base of each arrow gives the composition of the donor compartment and the tip gives the average composition of receiving compartments - ignoring back-fusion - as in *Figure 4*, main text. The opacity of the arrows represents the normalized number of cargo (divided by the total number of cargo in the system) transported per unit time (normalized by the fusion rate). (**C**) Net vesicular flux, excluding vesicle biochemical conversion, for different values of the budding rate $k_b$. Top panel: vesicular flux is represented

as a vector field as in *Figure 4*, main text. Bottom panel: vesicular flux is represented as the enrichment between the donor and the acceptor compartment, excluding back-fusion, as in *Figure 3*, main text.

## Quantification of the back-fusion

In the main text, we discussed the importance of back-fusion when the budding rate $k_b$ increases. To determine the likelihood of a vesicle to fuse back with the compartment it budded from, or a compartment of similar composition, we analyzed the trajectories of passive cargo-proteins.

Back fusion is defined as a vesicular transport event where the donor and acceptor compartments fall within the same composition bin. The fraction of the total vesicular flux leaving a compartment that undergoes back fusion is shown as a function of the composition of compartments in a triangular composition space in *Appendix 6—figure 1*. As expected, back-fusion is dominant near the vertices of the composition triangle, where compartments are pure and mostly bud vesicles of identity similar to their own identity. These vesicles are very likely to fuse back with the donor compartment by homotypic fusion. The back fusion flux increases with the budding rate $k_b$, going from $\sim [90\%, \ 10\%, \ 20\%]$ for respectively [*cis*, *medial* and *trans*]-compartments when $k_b = 10^{-2}$, to $\sim 100\%$ when $k_b = 10^2$ for all compartments.

The mean enrichment (composition difference between donor and acceptor compartments) seen by a transport vesicle is plotted in the main text in a normalized fashion. The non-normalized enrichment is plotted in *Appendix 6—figure 1*. The mean enrichment for all three identities vanishes when $k_b$ is large. This is explained by the contribution of back fusion. When $k_b$ is large, the compartments are very pure, and vesicles have a high probability to undergo back fusion, leading to a small value of the net enrichment.

## Vesicular transport by composition

*Figure 4* of the main text shows the average flux of vesicular transport as a vector field on the triangular composition phase space. The main conclusion is that the vesicular flux is retrograde (toward *cis*-compartments), when the compartments are mixed (low budding rate $k_b$), anterograde when the compartments are pure (high $k_b$), and centripetal (toward compartments of *medial*-identity) for intermediate budding rates. In the main text, only the total flux is described, which is the sum of the fluxes of vesicles of the three different identities. To better understand this net enrichment, the total flux can be dissected into the flux of of *cis*, *medial* and *trans*-vesicles. These results are shown in *Appendix 6—figure 1*. For low budding rate ($k_b \ll 1$), compartments are mixed with all three identities, with a composition distributed around a trajectory given by *Equation 9* and shown in *Appendix 6—figure 1*. These mixed compartments emit vesicles of all identities. The *cis*-vesicles predominantly fuse with pure *cis*-compartments, which always exist at steady-state, since pure *cis*-vesicles are continuously being injected in the system. On the other hand, *medial* and *trans*-vesicles fuse with *cis/medial* and *cis/trans*-compartments, as pure *medial*-compartments are absent, and pure *trans*-compartments are few and transitory. In this limit, the net enrichment is completely dominated by the retrograde transport of *cis*-vesicles, and is positive in cis-identity and negative in trans-identity.

For intermediate budding rate ($k_b \sim 0.1 \to 1$), and as $k_b$ increases, components can be efficiently sorted as they mature, since the budding and biochemical conversion rates are of the same order. Medial compartments are rather unstable, as they can fuse with both *cis*-rich and *trans*-rich compartments. Their reformation involves *medial* vesicles leaving *cis*-rich and *trans*-rich compartments to fuse with each other or small *medial*-rich compartments. *Appendix 6—figure 1*, shows that this is the main contribution to the overall vesicular flux, thereby defining the centripetal net vesicular flux shown in *Figure 4* - main text. Note that they primarily exit *trans*-compartments for $k_b \sim 0.1$, and *cis*-compartments for $k_b \sim 1$, denoting a transition from retrograde to anterograde vesicular fluxes.

For high budding rate ($k_b \gg 1$), compartments are very pure, and most budding events lead to vesicle back fusion from the donor compartment (*Appendix 6—figure 1*). A new phenomenon can be observed, which is the anterograde transport of *cis* and *medial*-vesicles

fusing with *medial* and *trans*-compartments, respectively. This is due to vesicle biochemical conversion *after* budding, as discussed below.

## Role of vesicles biochemical conversion

The biochemical conversion of vesicles after their budding from an immature compartment has been presented as a mechanism to prevent back fusion with the donor compartment and to promote anterograde vesicular transport (see Discussion - main text). This mechanism is naturally included in our model, as vesicles have the same biochemical conversion rate than any membrane patch that belong to bigger compartments. One can notice on **Appendix 6— figure 1**, that there is a major increase of the anterograde *cis*-to-*medial* and *medial*-to-*trans* vesicular fluxes between $k_b = 10$ and $k_b = 10^2$. This is due to the biochemical conversion of the vesicles *after* their budding. The budding vesicle has the same identity than the donor compartment, but undergoes biochemical conversion before undergoing back fusion and fuses with a more mature compartment.

   **Appendix 6—figure 1**, shows the net vesicular flux (for the same simulation than in the main text) ignoring the events where vesicles undergo biochemical conversion after their budding. The net vesicular flux vanishes in the vector field for high budding rates when these events are removed. We interpret this result considering dimers and trimers are dominant when $k_b$ is large. Disregarding vesicle biochemical conversion, such compartments emit as many immature vesicles fusing with immature compartments as mature vesicles fusing with mature compartments, yielding a vanishing net vesicular flux. However, this is not true for the net enrichment which still displays an anterograde signature of the vesicular transport. Indeed, the explanation of the vesicular transport considering the minority identity (see Discussion section in the main text) is still relevant in this regime; compartments larger than three vesicles can generate an anterograde transport by budding patches of membrane that just underwent biochemical conversion. Consequently, the net enrichment displays the characteristics of an anterograde vesicular flux even in the absence of vesicle biochemical conversion.

## Appendix 7

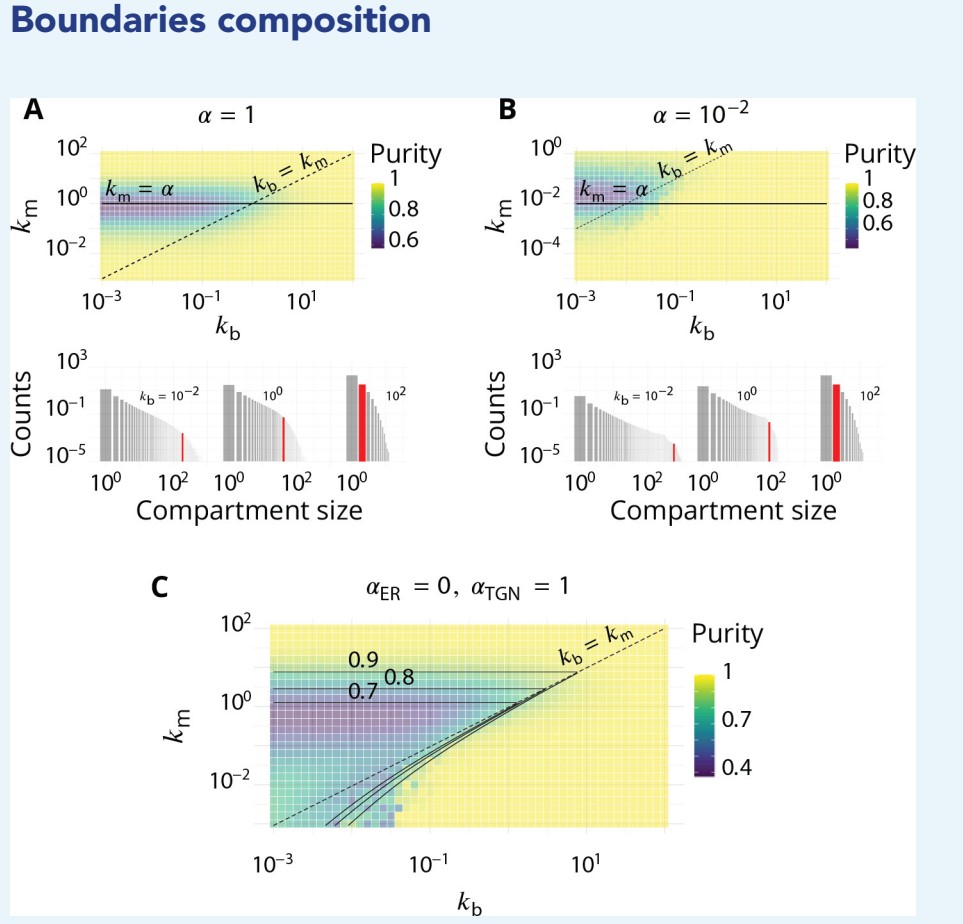

### Boundaries composition

**Appendix 7—figure 1.** Impact of the composition of the boundaries of the system (ER and TGN) on the steady-state organisation. Related to *Figure 2* - main text. (**A–B**). The fraction $\alpha$ of active species driving homotypic fusion with the boundaries are identical for the ER and the TGN. A. $\alpha = 1$ (same value as in the main text) B. $\alpha = 10^{-2}$. For both cases, the top panel is the purity diagram as a function of the normalized biochemical conversion rate $k_m$ and budding rate $k_b$. Dashed lines correspond to $k_m = k_b$ and solid lines to $k_m = \alpha$. For both cases, the bottom panel shows some examples of the size distribution for $k_m = \alpha$ and different values of $k_b$. (**C**) Purity diagram as a function of $k_m$ and $k_b$ when fusion with the ER is abolished: $\alpha_{ER} = 0$ whereas $\alpha_{TGN} = 1$. Solid lines are the predictions of the analytical model developed in Appendix 3.

Our system contains two boundaries: the *cis* face (the ER) and the *trans* face (the TGN). *cis* vesicles are injected via the *cis* face, and compartments may exit the system by fusing with either boundary. The way the different compartments fuse with the boundaries has a strong impact on their lifetime and average size. In the model, this is characterized by the parameter $\alpha$, which represents the fraction of the boundary composed of species able to elicit homotypic fusion of Golgi's components, that is, the fraction of *cis* species in the ER and of *trans* species in the TGN, and might be different for the two boundaries. In the main text, we focus on the case $\alpha = 1$ (arguably the maximum possible value) for both *cis* and *trans* boundaries. Here, we discuss the impact of this parameter on the structure of the steady-state, and we compare the results of simulations with the analytical model derived in Appendix 3.

Decreasing this parameter has two major impacts on the steady-state organisation: it decreases the biochemical conversion rate for which the steady-state fraction in cis, *medial* and *trans*-species are equal and it increases the residence time of compartments in the

system. This increases their size, as they have more time to aggregate, and increases the fluctuations in the system, as larger compartments have a stronger impact each time they exit or fuse together.

The impact of $\alpha$ on the fraction of *cis*, *medial* and *trans*-species in the system is rather straightforward if compartments are pure (*Equation 4*). In this case, the exit flux can be exactly computed, and *Equation 4* shows that the different species are in equal amount at steady-state when $k_m = \alpha$. If $k_m \gg \alpha$, the membrane patches have time to undergo biochemical conversion before exiting the system, which is then dominated by *trans*-species. The purity of the compartment is high, but this is a rather uninteresting limit as the system is dominated by a single species. If $k_m \ll \alpha$, compartments are recycled out of the system before *trans*-membrane patches appear, and the system is dominated by *cis* and *medial*-species, which are of equal amount at steady state if the system is perfectly sorted.

The same conclusion can be reached in the slow budding regime when compartments are not pure. Let's consider a *cis*-rich compartment of size $n$ contaminated by a fraction $\phi_{medial}$ of *medial*-species, and let's disregard, for simplicity, budding and fusion with other compartments. The number of *medial*-patches increases by biochemical conversion with a flux $k_m n(1 - \phi_{medial})$, and decreases by compartment fusion with the boundary, which removes all *medial*-patches with an average flux $\alpha(1 - \phi_{medial}) \times n \, \phi_{medial}$. At steady-state, one thus expects $\phi_{medial} \simeq k_m/\alpha$, suggesting that the lowest purity will be observed for $k_m \simeq \alpha$. The results of simulations, shown in *Appendix 7—figure 1*, confirm this prediction.

The transition of purity occurs when sorting mechanisms become more efficient than mixing mechanisms. As a first approximation mixing occurs by biochemical conversion and sorting by budding, yielding the prediction that the purity transition occurs when $k_b \simeq k_m$. Additional mixing mechanisms include the fusion of two slightly impure compartments of different compositions. Such slow events become relevant if compartments remain in the system for a long time, that is, if fusion with the boundaries is slow (small values of $\alpha$). In this case, the transition of purity occurs for $k_b > k_m$. Phase diagram of the purity of the system for a system where $\alpha = 10^{-2}$ can be found in *Appendix 7—figure 1*. As expected the diagram is centered around $k_m = \alpha$, with a transition of purity between $k_b \sim 10^{-2}$ and $k_b \sim 10^{-1}$ (and thus $k_b > k_m$).

As it tunes the residing time of compartments in the system, $\alpha$ also impact the average size of compartments. Decreasing the value of $\alpha$ increases the residence time and leads to larger compartments, as predicted by *Equation 4*. Note however that for small budding rates, when compartments are mixed and their exit through the boundary is difficult to estimate, the size distribution deviates from the single-component ideal distribution.

The Golgi being a highly polarized organelle, it could be argued that the parameters controlling the fusion with the *cis* face (ER) and the *trans* face (TGN), $\alpha_{ER}$ and $\alpha_{TGN}$ could be different. An obvious way to reduce the retrograde exit of material from the ER is to reduce $\alpha_{ER}$. To investigate the role of this asymmetry, we use an extreme regime of $\alpha_{ER} = 0$ and $\alpha_{TGN} = 1$. This prevents the recycling of impure compartments by fusion with the ER and broadens the low purity region of the parameter space, as shown in *Appendix 7—figure 1*, and explained by the analytical model presented in Appendix 3.

## Appendix 8

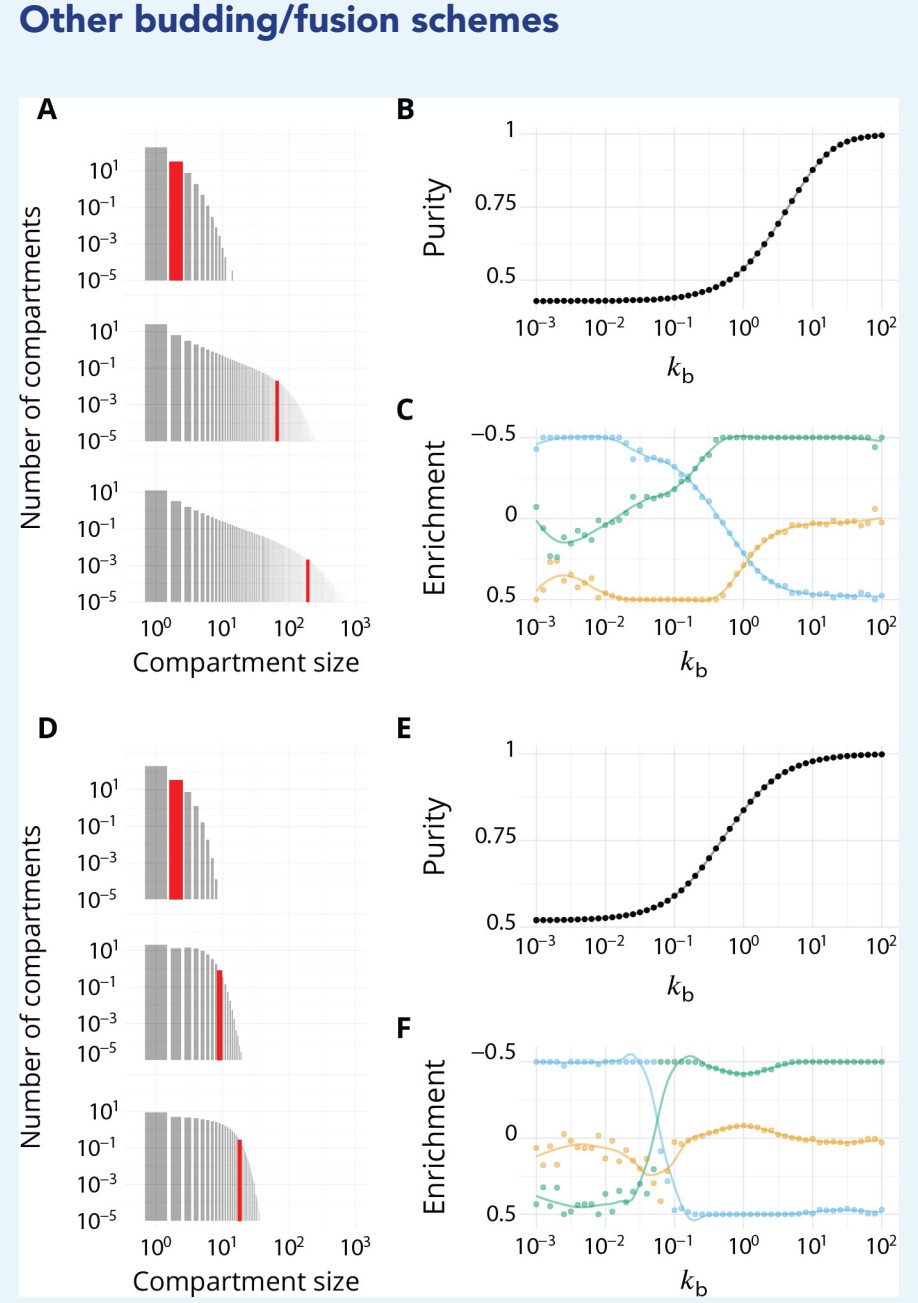

### Other budding/fusion schemes

**Appendix 8—figure 1.** Impact of different budding and fusion kernels on the system's organisation and vesicular transport. Related to *Figures 2–4* - main text. $k_\mathrm{m} = \alpha = 1$. (**A–C**) Linear budding scheme. (**D–F**) Inter-compartment fusion is prohibited. (**A** and **D**) Size distribution of compartment for $k_\mathrm{b}$ equals $10^2$ (top), 1 (middle) and $10^2$ (bottom). (**B** and **E**) Steady-state purity as a function of $k_\mathrm{b}$. (**C** and **F**) Enrichments in cis (blue), *medial* (orange) and *trans*-species (green) during vesicular transport (details of computations of this enrichment presented in Appendix 2) as a function of $k_\mathrm{b}$.

### Linear budding

To verify whether the assumption of saturated budding (*Equation 2*) has a strong impact on the results, we performed simulations with a linear budding mechanism ($f(\phi) = \phi$ in *Equation 10*), for which the budding flux for a given identity is linear with the number of patch

with this identity ($J_{b,i} = K_b n \phi_i = K_b n_i$). As shown in **Appendix 8—figure 1**, the average compartment size and purity follow the trends discussed in the main text for the saturated budding regime. However, the purity transition occurs for larger $k_b$, since the total budding flux is merely proportional to the size of the compartment for linear budding, and is thus smaller than for saturated budding, where it is proportional to the size times the number of species. Saturated budding thus promotes systems that are at the same time well sorted, and with large compartments.

Even though the purity transition is shifted toward higher values of $k_b$ the link between purity and vesicular flux is qualitatively conserved (**Appendix 8—figure 1**). For low purity, the system is retrograde with an enrichment in cis-species and a depletion of more mature species, and for high purity it is anterograde. However the retrograde flux is less marked than with a saturated budding rate (no clear depletion in trans-species). Indeed, efficient sorting relies on the capacity to export the minority component out of a compartment. Within the saturated budding scheme, the rate of export is proportional to the size of the compartment and does not depend on the number of minority components to export. Within the linear budding scheme however, it is proportional to the number of minority components. In the low $k_b$ regime, *trans*-rich compartments are large. They emit a large vesicular flux of immature components within the saturated budding scheme, but this flux is much smaller within the linear budding scheme. This explains the qualitative difference between the enrichment curves in the low $k_b$ regime for the saturated budding scheme (large depletion in trans identity - **Figure 3** - main text) and the linear budding scheme (almost no change in trans identity - **Appendix 8—figure 1**).

## Non-fusing compartments

Fusion between compartments is an important ingredient of the model discussed in the main text. It could be argued that the fusion of large cisternae with one another could be a much slower process than fusion involving much smaller transport vesicles. We have performed simulations where fusion between compartments is prohibited if both compartments are larger than a vesicle. Note that in the model, such compartments are still able to fuse with the boundaries, so that the composition of the system can still be predicted by **Equation 4** The results of this model are shown in **Appendix 8—figure 1**.

The size distribution of non-fusing compartment shows a broad peak, instead of the power-law seen in the case where compartments can fuse. As compartments can only grow by vesicle fusion and not by compartments fusion, their size is smaller than in the previous case. The compartments size is results from a balance between the rate at which compartments are recycled (fuse with one boundary) and the rate at which they grow by vesicle fusion. Consequently, the size distribution only weakly depends on $k_b$, as compartments have very little time to fuse with vesicles before exiting the system (**Appendix 8—figure 1**).

As expected, and because we remove the possible interactions between compartments, the centripetal flux we observe between the previously described anterograde and retrograde is abolished. This is particularly visible if we plot the (normalized) mean enrichment in cis, *medial* and *trans*-species seen by cargo as they are transported via vesicular fluxes (**Appendix 8—figure 1**). As before, the system goes from a purely retrograde flux, characterized by a depletion in trans-species and an enrichment in cis-species, to an anterograde flux, characterized by an enrichment in trans-species and a depletion in cis-species. However, the transitional centripetal regime we previously described, characterized by an enrichment in *medial*-species, vanishes. This suggests this particular regime is indeed resulting from the possible fusion between compartments, as it destabilizes *medial*-compartments that have a great probability to fuse with slightly impure *cis* or *trans*-compartments. Thus the transport is anterograde as soon as the purity transition occurs (**Appendix 8—figure 1**).

## Appendix 9

### Experimental proposal

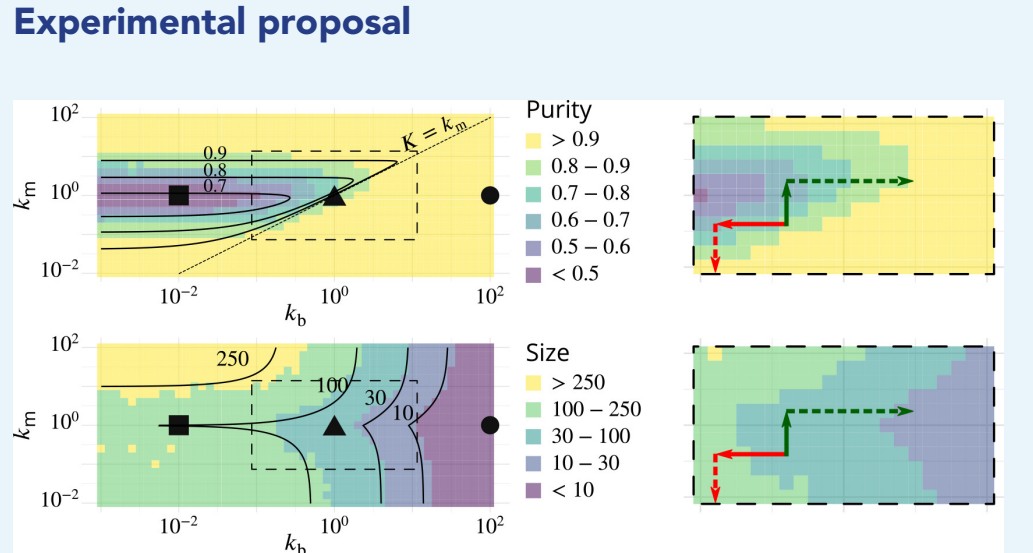

**Appendix 9—figure 1.** Possible experimental strategies to test the predicted correlation between size and purity of Golgi cisternae. The left column reproduces **Figure 2B–C**, showing the phase diagrams for the purity (top) and size (bottom) of Golgi cisternae upon varying the ratio of budding to fusion rates $k_b$ and the ratio of biochemical conversion to fusion rate $k_m$. The right column shows a zoom of the dashed square area, showing the observed result of physiological perturbations of cisternae purity and size (solid arrows) and prediction for combined perturbations (dashed arrows). Red arrow: observed result of a decrease of the budding rate $k_b$ by impairing COP I (**Papanikou et al., 2015**). Dashed red arrow: prediction for the combined decrease of the biochemical conversion rate $k_m$ by Ypt1 down-regulation. Green arrow: observed result of an increase of the biochemical conversion rate $k_m$ by Ypt1 over-expression (**Kim et al., 2016**). Dashed green arrow: prediction for the combined increase of the budding rate $k_b$ by COP I up-regulation.

