## [Decision Letter]

**Acceptance summary:**

Vagne et al. study a minimal model for the self-organisation of the Golgi, showing how the traditional models of "vesicular transport" and "cisternal maturation" arise as limiting cases of rich dynamical behaviour in a model of vesicle budding, maturation, fusion that leads to self-organised Golgi structure and cargo transport.

**Decision letter after peer review:**

[Editors’ note: the authors submitted for reconsideration following the decision after peer review. What follows is the decision letter after the first round of review.]

Thank you for submitting your work entitled "A minimal self-organization model of the golgi apparatus" for consideration by *eLife*. Your article has been reviewed by three peer reviewers, and the evaluation has been overseen by a Reviewing Editor and a Senior Editor. The following individual involved in review of your submission has agreed to reveal their identity: Jane Kondev (Reviewer #3).

You will see that the reviewers had differing views on the paper, with two highly critical and one supportive. Our decision has been reached after consultation between the reviewers. Based on these discussions and the individual reviews below, we regret to inform you that your work will not be considered further for publication in *eLife*.

The consensus among the reviewers is that there is no doubt in the potential utility of the kind of minimal model that you have developed, but there were serious questions about the connection between that model and existing data, about the presentation of the model itself, and the structure of the paper.

Reviewer #1:

The Sens group has been developing mathematical models of the Golgi apparatus since 2011 and this paper represents a continuation of this effort. Here, theoretical roles of vesicle budding, maturation and self-organization are used to describe the Golgi's various activities. While the mathematical computations and simulations are not in question, the paper provides no experiments on the Golgi to support or verify the proposed model, and the study fails to incorporate significant features of the Golgi (including the Golgi's unique lipid composition, its tubular-vesicular character, and its ability to form de novo from ER). The paper reads as an exercise for modeling a minimal, self-organizing system that, while attempting to capture some features of organelles (i.e., fusion, budding and maturation), is far removed from the actual complex system comprising the Golgi apparatus. Because the proposed model does not significantly further our understanding of the Golgi apparatus, I cannot support its publication in *eLife*. Among the concerns of this study is that the model is built on faulty or unproven assumptions regarding the Golgi apparatus (some of these are listed below). It is also worrisome that several papers providing quantitative experimental data detailing Golgi dynamics are never cited, perhaps because these data cannot easily be explained by their model.

1) In their Introduction, the authors state that proteins passing through the Golgi interact with Golgi enzymes in a pre-determined order dictated by the position of the enzyme in the Golgi stack (i.e., cis first and trans last). There is no evidence for this in the literature. Indeed, Farquhar et al., showed many years ago that some trans-acting enzymes are, in fact, in the cis-most cisternae. We also know that carbohydrate processing by Golgi enzymes occurs in a sequential manner whether or not the enzymes are in the ER (i.e., in BFA-treated cells) or Golgi. In both cases, it's substrate availability that dictates the processing events.

2) The authors claim in their Introduction that there are only two major classes of models for explaining the Golgi's activities (i.e., the vesicle transport and maturation models). However, a third model involving tubular continuities and lipid partitioning has also been proposed by the groups of Luini and Lippincott-Schwartz. This third model can account for key experimentally observed features of the Golgi that neither the vesicle transport or maturation models explain. These features include: the finding that secretory cargo rapidly mixes throughout the Golgi stack upon arrival; that different types of cargo exhibit different export kinetics from the Golgi; and that cargo export from the Golgi follows mass action laws. Sens' model cannot account for any these observed features of the Golgi.

3) The authors claim that each membrane patch in their model undergoes maturation from a cis to medial to trans identity. This may have some type of support in yeast (in which membrane patches budding out from the ER clearly “mature” by exchanging cytosolic components over time), but it has no support in mammalian Golgi systems. There, cargo transport through the Golgi stack has never been visualized due to the extremely compact nature of the stack, whose individual cisterna are separated by <20 nm, well below the diffraction limit of imaging.

4) The authors' model predicts that there will be more cis/medial components associated with Golgi structures having low maturation rates, and more trans components for Golgi with high maturation rates. There is no evidence in the literature supporting this prediction. If anything, when one lowers the temperature to slow movement through the Golgi (which in the Sens' model would slow maturation), molecules tend to accumulate at the trans face of the Golgi, the opposite of their prediction.

5) The authors state that cargo flux through the Golgi in yeast is two orders of magnitude faster than in mammalian Golgi. This is unlikely. Bonfanti's paper examining flux of VSVG through the Golgi by EM, which they cite as supporting evidence, is inappropriate for this calculation at many levels- from using repeated temperature shifts to counting a few immuno-gold particles per Golgi (a super crude quantification) to having to compare different fixed cells at different time points. The authors ignore the quantitative live cell imaging data of Golgi export of VSVG-GFP provided by Hirschberg et al., 1998, which measured VSVG export out of the Golgi in real time, with thousands of data points per cell. There, VSVG was shown to be transported out the Golgi at an enormous rate (~7,000 molecules leaving the Golgi per sec) at peak flux after release from the ER. Importantly, the Hirschberg paper also revealed that the amount of VSVG cargo leaving the Golgi is directly correlated with the amount of cargo in the Golgi, with a single rate constant characterizing VSVG efflux kinetics. Sens' paper does not integrate any of these findings into their analysis.

6) The few predictions made by the author's model are not supported by findings in the literature. For example, the authors' state that their model predicts that deletion of Arf1 will decrease the number of Golgi compartments and increase their size. Experimentally, this has already been tested with brefeldin A (which inactivates Arf1) or with Arf1 mutants. Contrary to the Sens' model prediction, when Arf1 is inactivated in cells, the Golgi disappears, being resorbed quickly into the ER with no increase in any Golgi subcompartment. This occurs through retrograde transport back to the ER, a pathway not incorporated into the Sens' model.

I appreciate the efforts made by the Sens group in using modeling to describe self-organizing systems, but their attempt to describe the Golgi apparatus with this approach is missing key aspects of Golgi organization and dynamics. This has resulted in predictions by the model that are counter to experimental data.

Reviewer #2:

Poorly written, hard to understand.

I was excited to read this paper since the work of one of the authors is known to me and I expected an understandable and well-crafted paper. I was rather crestfallen to find a poorly written paper, and one where I could not figure out what the authors were actually doing.

One major concern is that this paper is not at all easy to read. It is either i) not self-contained, or ii) sloppily put together. The authors refer to a "steady state" but do not at any time show that such a state exists, as far as I can see. There is no plot with the time evolution of any quantities being measured. In fact, they admit that there is no steady state, since the size of the system is not constant. They state that "the maximum number of time-steps is typically set to 10^6^ or 10^7^ in order to reach steady state and accumulate enough statistics on all the measured quantities", but that in and of itself is meaningless if one doesn't correctly specify the conditions underlying the measurement.

The paper as a whole is an odd combination of being verbose AND telegraphic at the same time. Echoing my above comment, I do not feel that I am getting enough information, enough detail, despite the verbosity. An example: it takes a long time to get to the point of how heterogeneous compartments can form; they say "this homotypic fusion mechanism (relying on local interactions) allows vesicular transport between compartments of different identities – a process that may be regarded as heterotypic fusion. It merely requires that the receiving compartment contains some membrane patches of identity similar to that of the emitted vesicle". Why not just give the rule, rather than the metaphysical description? It would take up less space!

Does seemingly everything have to be in the Appendix? One can understand page limits, but this is ridiculous. More explanation of the model in the main text would have been very welcome, because in its present form, the paper is extremely difficult to understand.

The question of system size should appear in the main text. As mentioned above, they cannot really keep it constant, so they just adjust the injection rate so it is more or less constant when they change K. They say that they aim for N ~ 300, but the actual system size (sometimes called size, sometimes mass) in the "steady state" changes significantly with K (see Appendix 1—figure 1B and Appendix 4). Moreover, the fluctuations are very large (since no time evolution plot is ever shown, one suspects that they are not really fluctuations…). Of course, the results would be highly dependent on system size, and one gets no justification for their choice of N ~ 300 until the Appendix.

The plots that are shown in the paper are not very useful. They rather tend to obfuscate. Figure 2A intends to show a power law for the size of the compartments with an exponential cutoff, but by plotting a bar diagram (even though it is a log-log scale), it is very difficult to see how well this works. They should plot actual points and a fit, or a parallel line showing an actual power law. In a similar vein, they overuse density plots that are not really quantitative (even in the Appendices). In Figure 2BC, they could at the very least plot some contour lines (perhaps dotted) for the same values that are plotted with a continuous line for the analytical approximation – from the colour map, it is difficult to judge.

Nomenclature:

It is confusing to keep track for reasons that are fully avoidable. For instance, their parameters are (K_i_,K_f_,K_b_,K_m_): the rates of injection, fusion, budding and maturation. All of them are normalised by Kf, which sets the time unit. But then they define k_m_ = K_m_/K_f_ and, instead of maintaining the use of lowercase k for the others, they then define K=K_b_/K_f_, J=K_i_/K_f_. As if the other problems with the paper weren't enough, now the reader has to remember what each means.

Main result is obfuscated:

The main result, in my opinion, is the existence of an intermediate "sorted" regime. It would be trivial to have a regime with a large mixed compartment or with small pure ones. The potentially interesting aspect of the model is that it can create pure compartments of at least intermediate size, if the parameters are chosen correctly. This is highly dependent on the way budding is modeled -- only explained in the Appendix, in an unclear way. In short, the budding flux for species i is defined as

J_b,_i = K_b_ \times n \times f(\phii)

where n is the size of the compartment and phi_i_ the proportion of vesicles of type i (phi_i_ = n_i_ /n). This is already confusing nomenclature, since they use J_b_ for this flux, when J is also the normalised injection rate. They then say that if the budding rate for species i depends linearly on its concentration, so

J_b,i_ = K_b_ n \phi_i_ = K_n_ n_i_

then you cannot get the intermediate regime. For that you need what they call "non-linear budding". In their words: "In order to reproduce this feature we choose a highly non-linear budding scheme f(phi_species_) = 1 if phi_species_ > 0 and f(phi_species_) = 0 otherwise". First of all, they are now using phi_species_ for what was before phi_i_.

I can only imagine what biological readers might make of this labeled "highly non-linear scheme". It is, after all, just a constant!

phi_i_ cannot be negative and if phi_i_ = 0, then there are no vesicles to bud. So the budding rate is just a constant, independent of everything and the same for all species in the same compartment.

Problematic statistical analysis:

The statistical analysis is problematic. First of all, because we don't know whether there is a steady state, can one even define averages? But even if there is a steady state, there are things like the figure in Appendix 4. There, they measure the "temporal standard deviation" of the mass and purity. How are they defining this standard deviation, when the individual measurements come from a correlated time series and are, therefore, not independent? Note that there is a very large dependence with K, which could be trivial for this reason (as the correlation time would obviously depend on K).

Conclusion:

I cannot recommend publication of the paper – it is too flawed at present – and importantly, it is not possible to evaluate whether the research presented is even correct. A revised version, should the editors wish to solicit one, at the very least, must show some evidence of a steady state and explain how averages are calculated. The authors should pay special attention and care to make the paper more readable. Much more readable. In particular, the model (the actual model) could be fit in the text and justification/discussion of terms could be left for the Appendix.

Reviewer #3:

This paper explores a simple model for the dynamics of vesicles as they progress through the Golgi apparatus and shows that a number of observed phenomena can be understood based on a small number of biologically, and physically, well-motivated assumptions.

I thoroughly enjoyed reading this paper, even though I am not an expert on the Golgi. I think it's a great example of theory at its best, applied to an interesting biological problem. The authors start with some simple observations that have been reported in the literature, which they turn them into a mathematical model that upon analysis yields some interesting and experimentally testable predictions. I am optimistic that their results will lead to some new experiments.

The key result of the paper is the connection between the directionality of the vesicular transport and the chemical composition of the Golgi compartments, for which there seems to be experimental evidence. The connection is established within a very simple model whose assumptions seem reasonable. The paper also describes the emergence of a number of structural and dynamical properties of the Golgi, from the same small set of assumptions. A particularly interesting feature of the model is how different structures and dynamics can be accommodated within the model by changing one or more parameters (of which there are only four). For example, the model predicts that there is an inverse relationship between the speed of vesicle transport and fidelity of processing, which is consistent with published comparisons between the structure of Golgi in normal and starved yeast.

The paper is very clearly written, and there were only a few places where I was not quite sure about what the authors had in mind.

1) When describing the Budding mechanism, there is reference to a non-linear budding scheme that depends on the compartment size and contamination state. These are described in detail later, but I would recommend that the authors say a few words here to give the reader an intuitive sense of the thing without having to skip ahead.

2) In describing maturation, the chemical transformations are described as going only one way. How is this justified? Is it known that the reverse reactions are very slow? Also, is the maturation rate independent of the composition of the compartment and if so, what is the justification for that?

3) A typical compartment size is defined as the ratio of the second and the first moment of the size distribution. I was struck by the fact that this would give a "typical size" even for an exponential distribution of sizes, which I would not say defines a typical size (at least not from the point of view that the mean and standard deviation of such a distribution are equal). Might be worth a short comment on why this is a good definition of "typical size".

4) In the "Steady-state organization" section there is reference to a cut-off size, which I don't think was defined.

5) In the Discussion there is a conclusion that the spatial information is less crucial than biochemical composition for the organization of the Golgi. I am not sure this is warranted since a comparison between the two cannot be made in a model lacking spatial information.

In general I was left wondering whether simple testable quantitative predictions can be made from the model. I very much liked the qualitative observations that the model neatly explains, but I would be even more excited if there was a prospect for putting it to a much more stringent test provided by quantitative experiments, which the authors might propose for an intrepid experimentalist to do.

[Editors’ note: further revisions were suggested prior to acceptance, as described below.]

Thank you for submitting your article "A minimal self-organization model of the golgi apparatus" for consideration by *eLife*. Your revised article has been reviewed by two peer reviewers, and the evaluation has been overseen by a Reviewing Editor and Suzanne Pfeffer as the Senior Editor. The reviewers have opted to remain anonymous.

The reviewers have discussed the reviews with one another and the Reviewing Editor has drafted this decision to help you prepare a revised submission.

Following your submission of a revised manuscript we sought the opinion of the original reviewers and, more recently, that of an additional reviewer (a theorist chosen specifically to be outside the field of Golgi research). As you can appreciate, purely theoretical papers in *eLife* need to be accessible to the biological readership of the journal and, if on the abstract side, may require extra effort to be suitable for publication. Such is the case here. As detailed below, we fully appreciate the significance of your major conclusions regarding the emergence of two distinct models from a singe underlying dynamics. At the same time, the reviewers request changes in the structure of the paper to improve readability, and that you address some key questions about the role of the actual structure of the Golgi.

Summary:

Vagne et al. study a minimal model for the self-organisation of the Golgi, showing how the traditional models of "vesicular transport" and "cisternal maturation" arise as limiting cases of rich dynamical behaviour in a model of vesicle budding, maturation, fusion that leads to self-organised Golgi structure and cargo transport.

Revisions:

1) Regarding prior work, the reviewers note that as the work of Patterson et al., 2008, provided a detailed quantitative model of the Golgi using one set of parameters, and furthermore assessed and tested those parameters by doing extensive quantitative live cell imaging experiments, it would be appropriate to seek a comparison between your model (with its dimensionless parameters) and those results. This would likely enlarge the scope of potential predictions for direct experimental tests. Workers in the Golgi field would benefit from some suggested experiments expected to skew the pathway in one direction or another, with predicted outcomes on the size of the compartment or the speed of traversal etc. For example, the cisternal progression model predicted that larger cargo export rates would be more sensitive to nocodazole dissolution of microtubules and resulting creation of ministacks, a prediction that turned out to be correct: PMID 25103235.

2) To help the general reader understand the context of this work better, we would like to see an introductory figure illustrating the structure of the Golgi (and perhaps some of the models for transport in the Golgi).

3) The reviewers suggest that putting more of the model equations in the main text would streamline the argument; one way of making things more understandable might be to start by writing down simple mean-field equations (as in the early parts of Appendix 3), before delving into the analysis of the detailed model. The back-and-forth between the main text and the appendices makes for difficult reading.

4) An important concern is that the model shows how Golgi structure and vesicular transport can arise in a self-organised fashion, but it remains unclear to what extent the authors can claim that vesicular transport in actual Golgi is dominated by this transport arising from self-organisation. When reading the paper, we wondered about the contribution of spatial structure or biochemical processes (that might bias which vesicles cargo goes into, or which compartments a vesicle fuses into). It was only pages later, in the Discussion, that one finds a paragraph on the role of spatial structure: even if experiments show that "Golgi functionality is preserved under major perturbation of its spatial structure", how is transport affected? From a quick look at the reference (Dunlop et al., 2017), it seems that transport can be slowed down massively, hinting that spatial structure is just as important as self-organisation. Are there similar experimental results on biochemical effects?

5) One of the main results in the paper is the existence of different regimes of self-organisation, and in particular the existence of an intermediate, sorted regime with large pure compartments. However, the definition of these regimes remains rather qualitative. Perhaps some plots of mean compartment size against mean purity, to complement Figure 2, would put this part of the analysis on a clearer footing.

6) The model shows very clear anterograde transport of cargo in the limit k_b_>>1, but the retrograde transport in the limit k_b_<<1 is less clear; in particular, the model appears to show transport from the trans to well-mixed compartments, but not into the cis compartments: rather, there is weak flow from the cis compartments to well-mixed ones, i.e. anterograde flow in part of the system. Is it possible to show that flow cannot be purely retrograde? – This might be an important constraint on the "cisternal maturation" mechanism. (Perhaps part of this issue is addressed in Appendix 6.)

---

## [Author Response]

We thank you and the referees for taking the time and effort to review our paper. After carefully reading through the referee reports, we are convinced that we can provide satisfactory answers to all the criticisms offered by reviewers 1 and 2, and that we can use the comments from the three reviewers to improve our paper.

Reviewer #1:The Sens group has been developing mathematical models of the Golgi apparatus since 2011 and this paper represents a continuation of this effort. Here, theoretical roles of vesicle budding, maturation and self-organization are used to describe the Golgi's various activities. While the mathematical computations and simulations are not in question, the paper provides no experiments on the Golgi to support or verify the proposed model, and the study fails to incorporate significant features of the Golgi (including the Golgi's unique lipid composition, its tubular-vesicular character, and its ability to form de novo from ER). The paper reads as an exercise for modeling a minimal, self-organizing system that, while attempting to capture some features of organelles (i.e., fusion, budding and maturation), is far removed from the actual complex system comprising the Golgi apparatus.

Our model does not provide new experimental data, but discusses existing data is a new way, in particular regarding the correlation between size and purity of Golgi compartments. The reviewer is correct that our model is far from encompassing the complexity of the Golgi. This was stated explicitly in the original version, including in the title, which promises a “minimal” model. There will undoubtedly be experimental observations that our model will not be able to explain. In the revised version, we discuss the few observations mentioned by the reviewers that are indeed not reproduced by our model, we explain why our model fails to reproduce these data and how it can be improved to do so. But as we discuss below, many of the observations said to be at odds with our model are in fact entirely consistent with it, and none contradict the model. These observations are also discussed in the revised version of the manuscript.

Notwithstanding, we think that our model offers fundamental conceptual advances that are important to convey to the Golgi community, and more generally to cell biologists interested in intracellular transport and organelle dynamics. Most existing models discuss transport through a pre-existing compartmentalised Golgi structure. The main goal of our model is to explain how this compartmentalised structure can spontaneously emerge through self-organisation. Our aim was and still is to investigate how the interplay between the biochemical maturation of membrane components and vesicular transport can lead, at the same time, to compartmentalisation and directed fluxes. The model involves three main parameters, so it definitely misses some of the Golgi’s complexity, but we believe that we have succeeded in answering that question from a theoretical point of view, and that our conclusions are very much relevant to Golgi dynamics.

Regarding lipid (and protein) composition of the Golgi, we only consider three types of membrane identity. It could be argued that this is not enough to account for the complex lipid composition of the Golgi, but we show that this is definitely enough to produce a rich and versatile dynamics. Our model Golgi is certainly able to form de novo and reaches a steady-state that does not depend on its initial state. Although this paper focusses on the study of the steady-state, the de novo formation is now explicitly shown in Appendix 4 (see also our response to reviewer 2 regarding the existence of a steady-state). Regarding the tubular-vesicular nature, we agree that we have disregarded the existence of tubular connections between cisternae. We note that the relevance of tubules in Golgi dynamics is still debated, except in some extreme conditions such as Brefeldin A treatment (see our response to point (6) below). In the revised version, a paragraph has been added discussing in more details the limitation of our model, in particular regarding tubular transport and lateral segregation and partitioning within cisternae, and many references (provided below) have been added.

Because the proposed model does not significantly further our understanding of the Golgi apparatus, I cannot support its publication in eLife. Among the concerns of this study is that the model is built on faulty or unproven assumptions regarding the Golgi apparatus (some of these are listed below). It is also worrisome that several papers providing quantitative experimental data detailing Golgi dynamics are never cited, perhaps because these data cannot easily be explained by their model.

We strongly disagree that our model does not further the understanding of Golgi structure and dynamics. Whether Golgi cisternae are long-lived structure exchanging material, or dynamical structures evolving over time is still fiercely debated after many decades. Our model is the first to show that, for the same set of individual mechanisms, there is a transition from one dynamics to the other upon varying the relative rates. Furthermore, we provide a complete analytical understanding of why this happens. We argue that the competition between sorting and maturation dictates the Golgi structure and dynamics. We have never seen this discussed before. This is an achievement, which allowed us to explain a number of experimental observations, in particular relating the size and the purity of Golgi compartments, which have been barely discussed and certainly not been explained before. This has been made clearer in the revised version of the manuscript.

We also disagree that our model is based on faulty assumptions. The reviewer criticised our treatment of (molecular) biochemical maturation (question 3). This is a fundamental assumption, which is indeed unproven for mammalian Golgi. This is explicitly mentioned as a hypothesis in the revised version. Our response to question 3 explains why we believe this is a reasonable assumption. We note that many influential models of the Golgi are based on it, so it is certainly worthwhile to quantitatively assess its potential role on Golgi dynamics (references are provided below). The other criticisms do not address model assumptions, but rather the discussion of model predictions. We hope that the reviewer will be convinced by our responses below.

Notwithstanding, we agree with the reviewer that important papers were not properly cited in the original version of the manuscript. This has been corrected in the revised version, where many references (listed in our response below) have been added. We now discuss how our model relates to the data provided in these papers, and how it should be improved (at the expense of increasing the number of parameters) when it fails to explain them.

1) In their Introduction, the authors state that proteins passing through the Golgi interact with Golgi enzymes in a pre-determined order dictated by the position of the enzyme in the Golgi stack (i.e., cis first and trans last). There is no evidence for this in the literature. Indeed, Farquhar et al., showed many years ago that some trans-acting enzymes are, in fact, in the cis-most cisternae. We also know that carbohydrate processing by Golgi enzymes occurs in a sequential manner whether or not the enzymes are in the ER (i.e., in BFA-treated cells) or Golgi. In both cases, it's substrate availability that dictates the processing events.

A more precise reference to the work of Farquhar et al. would be helpful to know which of her work the reviewer refers to. Nevertheless, many papers show that disrupting Golgi structure and dynamics affects the processing of proteins, and in particular glycosylation. Among those: (Hu et al., 2005), reporting that ceramide, which induces Golgi fragmentation, also induces glycosylation defects of integrin, disrupting cell adhesion. A similar phenotype with observed upon Brefeldin A treatment, strongly suggesting that BFA also induces glycosylation defects. (Shen et al., 2007) showed that siRNA suppression of BIG1 (a GEF also inhibited by BFA) disrupted Golgi structure and altered *β*1 integrin glycosylation, leading to aberrant cell adhesion and migration. (Xiang et al., 2013) and (Bekier et al., 2017) showed that knockout of the Golgi stacking proteins GRASP55 and GRASP65 impairs glycosylation and sorting of several proteins and lipids. In any case, this introductory sentence has no impact on the layout of our model what so ever, and can be removed if problematic. In the revised version, this sentence was modify to specifically mention glycosylation and references (Hu et al., 2005, Shen et al., 2007, Xiang et al., 2013, Bekier et al., 2017) are cited.

2) The authors claim in their Introduction that there are only two major classes of models for explaining the Golgi's activities (i.e., the vesicle transport and maturation models). However, a third model involving tubular continuities and lipid partitioning has also been proposed by the groups of Luini and Lippincott-Schwartz. This third model can account for key experimentally observed features of the Golgi that neither the vesicle transport or maturation models explain. These features include: the finding that secretory cargo rapidly mixes throughout the Golgi stack upon arrival; that different types of cargo exhibit different export kinetics from the Golgi; and that cargo export from the Golgi follows mass action laws. Sens' model cannot account for any these observed features of the Golgi.

There are indeed different qualitative models, including Luini’s cisternal maturation with tubular connections (Trucco et al., 2004, Rizzo et al., 2013), Lippincott-Schwartz’s rapid partitioning between processing and exporting sub-compartments (Patterson et al., 2008), or Pfeffer’s cisternal progenitor model (Pfeffer, 2010). The strengths and weaknesses of these different models are reviewed by Glick and Luini in (Glick and Luini, 2011). Variations of these models, such as the “rim progression” model also exist (Lavieu et al., 2013). These papers should have been cited and are cited in the revised version.

Notwithstanding, none of these models provide a quantitative analysis of the generation and maintenance of Golgi compartments, which are generally assumed to be of fixed number and size. The strength of our model is that the structure spontaneously emerges from the dynamics. Our model is in that respect fairly close to the concepts qualitatively exposed in the cisternal progenitor model (Pfeffer, 2010), but it is quantitative and can explain de-novo Golgi formation, as we now show in Appendix 4. One difference with the cisternal progenitor model is that we restrict intercisternae exchange to be vesicle-based, while Pfeffer suggests that larger cisterna fragments could also be exchanged. We have rephrase our description of existing models to explain that they mostly belong to two classes: one involving stable compartments exchanging components through fusion-based mechanisms, to which the vesicular transport, rim progression, rapid partitioning and cisternal progenitor models belong, and one involving transient compartment undergoing *en-block* maturation and in which fusion mechanisms are dispensable for cargo transport. The papers by Luini, Lippincott-Schwartz, Pfeffer and others (refs. (Trucco et al., 2004,Rizzo et al., 2013,Patterson et al., 2008,Pfeffer, 2010,Lavieu et al., 2013)) are now cited.

The features referred to the reviewer pertain to the dynamics of cargo transport through the Golgi, and our model can account for them. This is clearly a very important topic of much interests to us, but we emphasise that the present manuscript only discusses the steady-state organisation and vesicular exchange of Golgi components. The present article is already fairly dense, and we found it important to have a full description and quantification of the self-organisation model before tackling the issue of cargo transport. The dynamics of cargo sorting and transit through the Golgi is also very rich and deserve a focussed publication. This will be the subject of a follow-up “Research Advance” article which will be submitted to *eLife* later this year. Nevertheless, the simulations described in the present paper already includes cargo, which we use as marker of vesicular transport directionality. We emphasise that these cargo are “passive” in the sense that they do not feel the biochemical identity of the membrane patch they are bound to. Their dynamics thus correlates with membrane patches dynamics, averaged over all identities. Our simulations can already predict the Golgi export kinetics of such passive cargo. We have added a figure (Appendix 4—figure 2) showing that the export kinetics of cargo is indeed exponential. This dynamics is similar to the one observed by iFRAP (Figure 2 i of (Patterson et al., 2008)) or using the RUSH technics (Figure 3 e of (Boncompain et al., 2012)). The exponential exit kinetics was one of the main arguments in (Patterson et al., 2008) to dismiss a “strictly cisternal maturation” dynamics, a conclusion we fully support. The revised version now cites (Patterson et al., 2008, Boncompain et al., 2012).

The specific features mentioned by the reviewer are entirely consistent with our model.

1) Rapid mixing of cargo: The kinetics of cargo distribution amongst cisternae of different identities following their release from the ER is at present not available from our simulations, and we plan to do these simulations in the follow-up paper. Nevertheless, the fact that incoming cargo can distribute throughout the spectrum of compartment identity in a time scale not limited by biochemical maturation can already be inferred from Figure 4 (main text) and Appendix 6—figure 1B of our paper. Although these are steady-state distributions of compartment identity and intercompartment exchange (vesicular in our model), comparing the distribution of the system’s size and the distribution of vesicular exchange arrows show that compartments of all identity are dynamically connected through inter-cisternal exchange. Furthermore, compartments can fuse with one another in our model, which further speeds-up the kinetics of cargo mixing. This leads to the presence of cargo in compartments of all identity within a time scale that is not limited by biochemical maturation. We realize that this remains qualitative at this point, but we hope that this will convince the reviewer that rapid mixing of cargo (which is not the topic of the present paper) is not at odd with our model.

2) Cargo export follow law of mass action: this is directly demonstrated by the exponential decay of cargo abundance in the Golgi following a pulse-chase-like simulation. This is now shown in Appendix 4—figure 2 and discussed in the main text.

3) Different export kinetics for different cargo: our model naturally leads to different export kinetics for cargo that interact differently with the various membrane identities (which will be called “ colored cargo” in the follow-up article), related to the fact that these cargo would reside in different membrane domains (patches of different identities in our framework). Trans cargo (residing preferentially in Trans membrane patches), exit faster than Medial and Cis cargo because they can enter Trans vesicles that can fuse directly with the exit. Cis cargo are often recycled to the ER via their packaging into Cis vesicles and show the slowest exit rate.

We can make the data regarding colored cargo available to the reviewer, although additional work is needed to fully understand them theoretically. We stress that cargo dynamics is beyond the scope of the present work, which focusses on Golgi self-organisation. We strongly feel that adding results on cargo dynamics to the present paper could obfuscate our main message.

3) The authors claim that each membrane patch in their model undergoes maturation from a cis to medial to trans identity. This may have some type of support in yeast (in which membrane patches budding out from the ER clearly “mature” by exchanging cytosolic components over time), but it has no support in mammalian Golgi systems. There, cargo transport through the Golgi stack has never been visualized due to the extremely compact nature of the stack, whose individual cisterna are separated by <20 nm, well below the diffraction limit of imaging.

The maturation of membrane patches, which is one of the three fundamental mechanisms in our model, refers to the local biochemical maturation of membrane components. Importantly, it does not refer to the global maturation of entire Golgi cisternae. In fact, we have replaced the term maturation (which bears strong connotation in the Golgi field) by biochemical conversion in the revised version to avoid confusing the molecular mechanism with its putative large-scale consequence.

Our inspiration for this model draws from the so-called Rab cascade which, as the reviewer correctly points out, has not yet been demonstrated in Mammalian Golgi. However, many of the proteins involved in the Yeast Rab cascade are conserved in mammalian cells (Klute et al., 2011), so it is important to explore the possibility that this mechanism is also relevant to mammalian Golgi. Furthermore, the Rab cascade concept is at the basis of the cisternal progenitor model (Pfeffer, 2010), and it is fundamentally important to test this model quantitatively, which is what we do here. One strong result of the present work is that this molecular biochemical conversion mechanism might lead to the overall maturation of cisternae (as in Yeast), but might also result in the export of the matured component from a cisternae whose identity remains mostly constant over time, which could correspond to mammalian Golgi. This is a conceptual advance, which is permitted by the quantitative model we propose. Again, our point of view is to adopt simple rules and to see the variety of structures that can issue as a function of the parameters. These rules are surely not sufficient to reproduce the entire diversity of mechanisms at play in the Golgi, but our results are robust and well controlled. Any model of the Golgi that includes some form of biochemical conversion and inter-cisternal exchange will inevitably encounter the universal phenomenon uncovered by our model.

We have emphasized that the biochemical conversion mechanism as a hypothesis in the revised version of the manuscript. We also have stressed that this does not have to rely on the Rab cascade, but could happen in many ways. It could also correspond to the modification of lipid components by enzymes in the Golgi, as nicely reviewed in (McDermott and Mousley, 2016).

4) The authors' model predicts that there will be more cis/medial components associated with Golgi structures having low maturation rates, and more trans components for Golgi with high maturation rates. There is no evidence in the literature supporting this prediction. If anything, when one lowers the temperature to slow movement through the Golgi (which in the Sens' model would slow maturation), molecules tend to accumulate at the trans face of the Golgi, the opposite of their prediction.

It is difficult to find clear experimental quantification of what would be the result of a decrease of maturation rate that would not affect other processes. So we agree with the reviewer that several predictions we make are not (yet) supported by experiments. We don’t feel that we should be criticised for making these predictions anyway. In the paper, we briefly discuss the result of starvation (glucose deprivation) experiments which slowed cisternal maturation by a factor of 2.5 and coincides with the Golgi becoming more polarized (Levi et al., 2010). This agrees with the general trends we put forward, but is in no way a falsifiable test, since one cannot be certain that starvation slow-down is due to a decrease of the maturation rate alone. We have since then become aware of more direct observations of the effect of varying the maturation rate (Kim et al., 2016). Interpreting this publication’s results is less straightforward as they can tune independently multiple steps of Golgi maturation, which is not the case in the present model implementation where all maturation steps occur with the same rate *k*_m_. In this paper, they show that overexpression of Ypt1 (in Yeast) increases the transition rate from early to transitional Golgi, and increases the co-localization of early and late Golgi’s markers. We interpret this as a decrease of Golgi’s purity upon increasing the maturation rate by unbalancing the ratio of budding to maturation rates. This suggests that the wild-type Yeast Golgi is closed to the line *k*_m_ = *k*_b_ (see Figure 2C of the paper). In the same paper, they report that over-expression of Ypt31 has a more complex phenotype. It accelerates transition to Late Golgi identity, but also increases exit from the Golgi via clathrin vesicles. Within our model, this would correspond to a concomitant increase of *k*_m_ for the medial-trans conversion and increase the budding rate of trans species. We did not attempt to model this, which goes beyond our simplifying approximation that the different rates are the same for all identities.

We would not venture to predict the result of a temperature shift, which probably affects all the parameters of the model in a way hardly controllable. It is not clear to us why the reviewer says that decreasing temperature affect the maturation rate and not budding or fusion? Temperature decrease to 20^◦^C slows down ER to Golgi export, and slows done Golgi to TGN export even more dramatically, leading to large Trans compartment (Ladinsky et al., 2002). This indicates that exchange is affected as much as, if not more than maturation. Our prediction are actually not based on the value of individual rates, but rather on the value of ratios of rates. Regarding the prediction alluded to by the reviewer, our claim is that increasing the ratio of maturation rate over budding rate should result in larger Trans cisternae. This fact agrees with results of Arf1 depletion, which likely decreases the budding rate (Bhave et al., 2014), and would also agree with the temperature shift data provided budding is slowed more than maturation, which seems reasonable given our current knowledge.

The revised version of the manuscript now proposes additional experimental strategies to test our predictions, detailed in our response to reviewer 3’s last comment.

5) The authors state that cargo flux through the Golgi in yeast is two orders of magnitude faster than in mammalian Golgi. This is unlikely. Bonfanti's paper examining flux of VSVG through the Golgi by EM, which they cite as supporting evidence, is inappropriate for this calculation at many levels- from using repeated temperature shifts to counting a few immuno-gold particles per Golgi (a super crude quantification) to having to compare different fixed cells at different time points. The authors ignore the quantitative live cell imaging data of Golgi export of VSVG-GFP provided by Hirschberg et al., 1998, which measured VSVG export out of the Golgi in real time, with thousands of data points per cell. There, VSVG was shown to be transported out the Golgi at an enormous rate (~7,000 molecules leaving the Golgi per sec) at peak flux after release from the ER. Importantly, the Hirschberg paper also revealed that the amount of VSVG cargo leaving the Golgi is directly correlated with the amount of cargo in the Golgi, with a single rate constant characterizing VSVG efflux kinetics. Sens' paper does not integrate any of these findings into their analysis.

We were well aware of the data reported in Hirschberg’s paper, which was cited in the original version and is not in contradiction with our statement. We did not state that the flux going through the Golgi is smaller in mammalian cells, but that the Golgi dynamics – which includes rates of budding, fusion and maturation – is slower. The number of 7000 molecules per sec. leaving the Golgi is a flux. This is controlled by the parameter *J* in our model. Doubling this flux would double the steady-state Golgi size without changing its dynamics, which is characterised by rates. Hirschberg reports Golgi to plasma membrane export rates for VSVG of order 3.0% per min, which corresponds to an exit rate constant of 1*/*30min. This paper reports Golgi residence time around 40 min. This is consistent with iFRAP (Patterson et al., 2008) and RUSH (Boncompain et al., 2012) experiments, which consistently report Golgi transit time of order 20 to 40 min for transit cargo (much longer times for Golgi resident proteins). On the other hand, the dynamics of Yeast Golgi, characterised by the maturation of cisternae, is of order 1 min (Losev et al., and Matsuura-Tokita et al., Science 2006, both cited in the paper. We thus firmly stand by our claim that all available evidence we are aware of indicate a Golgi dynamics one to two orders of magnitude faster in Yeast than in mammalian cells. We have clarified our statement in the revised version, and included the references above (including Hirschberg et al.). As already discussed in our response to question 2, the cargo export kinetics is exponential, in agreement with Experimental observations. This is now discussed in the paper, with one added figure (Appendix 4—figure 2).

6) The few predictions made by the author's model are not supported by findings in the literature. For example, the authors' state that their model predicts that deletion of Arf1 will decrease the number of Golgi compartments and increase their size. Experimentally, this has already been tested with brefeldin A (which inactivates Arf1) or with Arf1 mutants. Contrary to the Sens' model prediction, when Arf1 is inactivated in cells, the Golgi disappears, being resorbed quickly into the ER with no increase in any Golgi subcompartment. This occurs through retrograde transport back to the ER, a pathway not incorporated into the Sens' model.

Retrograde transport back to the ER is definitely included in the model. The fusion with the entry (the ER) occurs according to the rules of homotypic fusion, and can be modulated through the parameter *α*_ER_ (see subsection “Model” and Appendix 7).

Nevertheless, this is an important question that is discussed in the revised version of the manuscript. BFA is fairly non-specific, and as far as we know, there is no clear explanation of how it affects the Golgi structure. However, a similar phenotype is observed with Golgicide A (GCA), which inhibit GBF1, a GEF for Arf1. This leads to the dissociation of COPI coat and induces disassembly of the Golgi and the TGN (Saenz et al., 2009). There are some subtle differences, as BFA induces TGN tubulation, while GCA apparently causes TGN dispersal into small vesicles. This shows how difficult it would be for a model, even a highly complex multi-parameter model, to capture these effects.

Nevertheless, we discuss this issue in a specific paragraph dedicated to the limitations of our model. One limitation is that we include a single exchange mechanism, based on vesicles. Our model does not include tubules. We argue in the revised version of the manuscript that transient tubular connections between compartments or tubules that fuse with one compartment after detachment (Sciaky et al., 1997,Trucco et al., 2004) can to some extent be modeled in similar ways than our vesicle exchange mechanism, with the need of defining tubule nucleation/fusion and scission rates. However the tubules induced by BFA are of a different nature. They are much more stable and do not detach. This results in the Golgi quickly redistributing in the ER as soon as a tubular connection is established between the Golgi and the ER, in a way suggesting a purely physical (tension-driven) mechanism (Sciaky et al., 1997). Our model does not include this possibility, as is now clearly stated in the revised version. We actually believe that these observations could be made consistent with our predictions by adding a size-dependent tubulation mechanism based on membrane mechanics. The force required to form membrane tubules increases with membrane tension (Der´enyi et al., 2002). We predict that Arf1 inactivation increases the cisterna surface area *S* (which is what we call the size in the model). It should also increase the cisterna volume *V* , and should increase the so-called surface-to-volume ratio *S*^3*/*2^*/V* ∼ *N*^1*/*2^ (where *N* is the number of vesicles that fused together to form the cisterna). The membrane tension of a vesicle decreases (in a fairly non-linear way) upon increase of surface-to-volume ratio (Evans and Rawicz, 1990). One thus expects tubulation to increase with cisterna size, which (with the help of our model) could explain why Arf1 inactivation (indirectly) induces massive cisterna tubulation and strongly increases the fusion probability with the ER. Since large cisternae are under low tension (probably lower than that of the ER (Upadhyay and Sheetz, 2004)), the tension difference would drive their absorption into the ER upon tubular contact. We could thus reproduce the BFA phenotype by saying that the ER fusion parameter *α*_ER_ is size dependent. The argument above is not supposed to be quantitative, and we are well aware that the effect of BFA could be multifactorial. It could for instance prevent the scission of tubules that form continuously in normal conditions but never get long enough to reach the ER. The argument is merely intended to show that the BFA treatment is not in contradiction with our prediction, but requires improvement of the model (and the use of more parameters). We have added a shorter version of the comment above near the end of the Discussion section.

I appreciate the efforts made by the Sens group in using modeling to describe self-organizing systems, but their attempt to describe the Golgi apparatus with this approach is missing key aspects of Golgi organization and dynamics. This has resulted in predictions by the model that are counter to experimental data.

We hope we could convince the reviewer that most of the data she/he thought were conflicting with our predictions are actually in agreement with them, or at least not inconsistent with them, but beyond the scope of such a minimal model. We do agree that the limitations of the model need to be discussed in more detail, and we have done so in the revised version with the help of the reviewer’s comments. We want to stress again that this is a conceptual model, not designed to reproduce specific experiments, but rather to make a conceptual point that Golgi structure and dynamics are intimately linked and cannot be studied separately. We find remarkable that completely different phenotypes can be explained within such a simple model solely by changing rates. We feel that it is important to convey this message to the Golgi community.

Reviewer #2:Poorly written, hard to understand.I was excited to read this paper since the work of one of the authors is known to me and I expected an understandable and well-crafted paper. I was rather crestfallen to find a poorly written paper, and one where I could not figure out what the authors were actually doing.

We are very disappointed that the reviewer found our paper difficult to read. We can assure her/him that this is not due to our sloppiness, as we actually worked hard, but apparently failed, to make our paper understandable without the need for an extensive theoretical background. We have endeavoured to improve the paper readability, with the help of the reviewers’ comments, in the revised version of the manuscript.

One major concern is that this paper is not at all easy to read. It is either i) not self-contained, or ii) sloppily put together. The authors refer to a "steady state" but do not at any time show that such a state exists, as far as I can see. There is no plot with the time evolution of any quantities being measured. In fact, they admit that there is no steady state, since the size of the system is not constant.

Several of the reviewer’s comment relate to the existence of a steady-state. Let us be very clear that a steady-state does exist in our system. We have added Appendix 4 to the paper including a figure (Appendix 4—figure 1) showing the evolution of the system’s size with time. The organelle forms de-novo, and reaches the steady-state – under any parameter values – after a time of order a few 1*/K_f_*. There is apparently a confusion in the reviewer’s mind between the variation of the system’s size with time and its variation with the parameters. We aimed at fixing the steady-sate systems size to a constant value (independent on budding, fusion and maturation rates) by adjusting the influx to the different parameter values, in order to precisely characterise compartmentalisation. Since this is done through an analytical model that is only rigorously valid under high budding rate, we observe a small deviation from the targeted system size (or order 10%) for low budding rates, as shown in Appendix 5—figure 1. Once the steady-sate is reached, the system’s size fluctuates around the steady-state value (characterized by the standard deviation – Appendix 5—figure 1) in a parameter-dependent way that we qualitatively understand, and explain in Appendix 5 (and also in the main text in the revised version). Fluctuations are large for small budding rate *k*_b_ because compartments are large and transient. This feature is realistic and relevant to Golgi organisation. We have expanded the Discussion of the steady-state fluctuations in the revised version of the paper.

They state that "the maximum number of time-steps is typically set to 10^6^ or 10^7^ in order to reach steady state and accumulate enough statistics on all the measured quantities", but that in and of itself is meaningless if one doesn't correctly specify the conditions underlying the measurement.

We agree with the reviewer that giving the typical number of simulation time-steps is not in itself sufficiently informative. We have modified Appendix 1 (and added a discussion of the transient regime in Appendix 4) to explain this better. The steady-state is reached after a time ∼ 1*/K_f_*. Depending on the parameter, this corresponds to between 10^3^ and 10^6^ simulation time-steps. In practice, we start the simulations with a system already containing the steady-sate amount of cis, medial and trans components to reduces the duration of the transient regime. Our simulations last at least 10^7^ time-steps, which is much longer than any correlation time that might exist in our system (see Appendix 4—figure 1 in the revised version). All quantities reported in the paper are averaged over the full simulation time, excluding the first 10^6^ time-steps to get rid fo the transient regime. We have checked that these quantities do not depend on the total simulation time.

The paper as a whole is an odd combination of being verbose AND telegraphic at the same time. Echoing my above comment, I do not feel that I am getting enough information, enough detail, despite the verbosity. An example: it takes a long time to get to the point of how heterogeneous compartments can form; they say "this homotypic fusion mechanism (relying on local interactions) allows vesicular transport between compartments of different identities – a process that may be regarded as heterotypic fusion. It merely requires that the receiving compartment contains some membrane patches of identity similar to that of the emitted vesicle". Why not just give the rule, rather than the metaphysical description? It would take up less space!

The Golgi field is a highly contentious one, in which there exist people that are quick to dismiss models that do not fit their preconceived ideas. We wrote this particular sentence related to homotypic fusion following comments made by Golgi experts that homotypic fusion was inconsistent with the fact that vesicles could be exchanged between compartments of different identities. While this is obviously false, it unfortunately can be sufficient for some people to reject the model. By trying to protect us against these types of criticisms, we apparently ran into the problem of verbosity. This sentence has been simplified, and the section describing the model has been modified in the revised version of the paper.

Does seemingly everything have to be in the Appendix? One can understand page limits, but this is ridiculous. More explanation of the model in the main text would have been very welcome, because in its present form, the paper is extremely difficult to understand.

We tried to give a qualitative feel for the model and to explain the main results in the main text, and to be rigorous in explaining the role of every parameters and of the model specific assumptions in the Appendices. In the revised version, we have made major changes to the “Model” section following the recommendations of the reviewers.

The question of system size should appear in the main text. As mentioned above, they cannot really keep it constant, so they just adjust the injection rate so it is more or less constant when they change K. They say that they aim for N ~ 300, but the actual system size (sometimes called size, sometimes mass) in the "steady state" changes significantly with K (see Appendix 1—figure 1B and Appendix 4). Moreover, the fluctuations are very large (since no time evolution plot is ever shown, one suspects that they are not really fluctuations…). Of course, the results would be highly dependent on system size, and one gets no justification for their choice of N ~ 300 until the Appendix.

There is apparently another confusion here regarding the system’s size (mentions of the system’s mass has been corrected in the revised version). Appendix 1—figure 1B shows how the size and purity phase diagrams varying the system’s size from *N* = 30 to *N* = 1000. The main message is that the structure of the phase diagrams does not depend on system size. The three regimes (mixed, sorted, vesicular) still exist and the transition between them happens for the similar values of parameters. Appendix 5 show a 10% variation of the system’s size with the parameter, which is small, and proves that we can successfully fix the system’s size while varying the parameters. The fluctuations characterised in Appendix 5 (previously Appendix 4) are real fluctuations (also observable in the time-variation of the system’s size shown in the new Appendix 4—figure 1), which depend on the parameter for reasons we understand and explain in Appendix 5 (and also in the main text in the revised version) (low budding rate, large compartments, hence large fluctuations related to compartment’s fusion with the exits). The standard deviation is indeed large in the small budding regime. It amounts to about a third of the system’s size, and corresponds to the typical size of compartments in this regime. This is explained better in the revised version, and is supported by the addition of Appendix 4—figure 1 showing the fluctuations in a time series. The revised version also includes a discussion of the effect of the system’s size in the main text.

The plots that are shown in the paper are not very useful. They rather tend to obfuscate. Figure 2A intends to show a power law for the size of the compartments with an exponential cutoff, but by plotting a bar diagram (even though it is a log-log scale), it is very difficult to see how well this works. They should plot actual points and a fit, or a parallel line showing an actual power law. In a similar vein, they overuse density plots that are not really quantitative (even in the Appendices). In Figure 2BC, they could at the very least plot some contour lines (perhaps dotted) for the same values that are plotted with a continuous line for the analytical approximation -- from the colour map, it is difficult to judge.

The figures of the revised version has been modified with the help of these suggestions. The new version of Figure 2 shows the power law as a dashed line. The power law is very accurate up to the cross-over size, shown (as before) as a vertical red bar. For Figure 2B and Figure 2C, discrete filled contour plots are used to ease the comparison with the analytical expression. We believe that the new figure clearly show that the continuous approximation qualitatively recapitulate the numerical behaviour quite well. The colors were chosen to be color blind friendly.

Nomenclature:It is confusing to keep track for reasons that are fully avoidable. For instance, their parameters are (K_i_,K_f_,K_b_,K_m_): the rates of injection, fusion, budding and maturation. All of them are normalised by Kf, which sets the time unit. But then they define k_m_ = K_m_/K_f_ and, instead of maintaining the use of lowercase k for the others, they then define K=K_b_/K_f_, J=K_i_/K_f_. As if the other problems with the paper weren't enough, now the reader has to remember what each means.

This has been fixed in the revised version. We use systematic notations with dimensionless quantities designed by lower case: *k*_m_ = *K*_m_*/K*_f_, *k*_b_ = *K*_b_*/K*_f_, *j* = *J/K*_f_…

Main result is obfuscated:The main result, in my opinion, is the existence of an intermediate "sorted" regime. It would be trivial to have a regime with a large mixed compartment or with small pure ones. The potentially interesting aspect of the model is that it can create pure compartments of at least intermediate size, if the parameters are chosen correctly. This is highly dependent on the way budding is modeled -- only explained in the Appendix, in an unclear way. In short, the budding flux for species i is defined asJ_b,_i = K_b_ \times n \times f(\phii)where n is the size of the compartment and phi_i_ the proportion of vesicles of type i (phi_i_ = n_i_ /n). This is already confusing nomenclature, since they use J_b_ for this flux, when J is also the normalised injection rate. They then say that if the budding rate for species i depends linearly on its concentration, soJ_b,i_ = K_b_ n \phi_i_ = K_n_ n_i_then you cannot get the intermediate regime. For that you need what they call "non-linear budding". In their words: "In order to reproduce this feature we choose a highly non-linear budding scheme f(phi_species_) = 1 if phi_species_ > 0 and f(phi_species_) = 0 otherwise". First of all, they are now using phi_species_ for what was before phi_i_.I can only imagine what biological readers might make of this labeled "highly non-linear scheme". It is, after all, just a constant!phi_i_ cannot be negative and if phi_i_ = 0, then there are no vesicles to bud. So the budding rate is just a constant, independent of everything and the same for all species in the same compartment.

The existence of a sorted regime with large and pure compartments is indeed an important result. The fact that structural transitions are directly related to transition in the direction of vesicular transport is another crucial result. Our original statement (that non-linear budding was “essential” to the existence of a sorted regime) was too strong. The existence of a well-defined sorted regime with large compartments is indeed promoted by non-linear budding. With a linear budding scheme, the purity transition appears for larger values of the budding rate (*K*_b_*/K*_f_ ' 3 Appendix 8—figure 1B, instead of *K*_b_*/K*_f_ ' 0.6 with the non-linear scheme, Figure 3C). Consequently, compartments are smaller at the transition with a linear budding scheme (*n* ' 10 instead of *n* ' 50 with non-linear budding when the system’s size is fixed at *N* = 300). The text has been revised to reflect this.

The rationale behind the non-linear budding scheme is based on the Michaelis-Menten kinetics, for which the rate of synthesis (the formation of a budded vesicle in our case) is linear with the substrate concentration at low concentration, and saturates at high concentration. This is a very common kinetics for enzymatic reactions in biological systems, and is relevant to the budding kinetics of coated vesicles. This can be shown with a model accounting for the kinetics of coat protein binding to the membrane, and the kinetics of selective recruitment of membrane components during coat formation (Vagne and Sens, 2018b). Non-linear budding basically corresponds to a Michaelis-Menten kinetics where the Michaelis constant is very low (high affinity between the coat and membrane components) so that saturation occurs for very small values of *n_i_*. In the revised version, we have changed the name to “saturated budding” and *φ*_species_ has been changed into *φ_i_*.

Problematic statistical analysis:The statistical analysis is problematic. First of all, because we don't know whether there is a steady state, can one even define averages? But even if there is a steady state, there are things like the figure in Appendix 4. There, they measure the "temporal standard deviation" of the mass and purity. How are they defining this standard deviation, when the individual measurements come from a correlated time series and are, therefore, not independent? Note that there is a very large dependence with K, which could be trivial for this reason (as the correlation time would obviously depend on K).

As said above, there is always a steady state at which exiting fluxes balance the incoming flux. Fluctuations are correlated in the low budding regime, since they come from the fact that large compartments build up over time, and disappear by fusion with the boundary. This correlation occurs over time scales of order the inverse fusion rate 1*/K*_f_ and does not challenge the calculation of the temporal standard deviation because this measurement is made on time series over thousands of 1*/K*_f_. The revised version contains a new Appendix 4 with a figure showing the fluctuations in a time series. We hope that this figure will make it clear that we are indeed looking at a steady-state, with real fluctuations which statistics is properly quantified. We have expanded the discussion on statistical analysis at the end of Appendix 1 in the revised version.

Conclusion:I cannot recommend publication of the paper – it is too flawed at present – and importantly, it is not possible to evaluate whether the research presented is even correct. A revised version, should the editors wish to solicit one, at the very least, must show some evidence of a steady state and explain how averages are calculated. The authors should pay special attention and care to make the paper more readable. Much more readable. In particular, the model (the actual model) could be fit in the text and justification/discussion of terms could be left for the Appendix.

We have followed these recommendations to prepare the revised version of the manuscript. We hope that this revised version and the response above will convince the reviewer that our statistical analysis of the model is entirely correct.

Reviewer #3:This paper explores a simple model for the dynamics of vesicles as they progress through the Golgi apparatus and shows that a number of observed phenomena can be understood based on a small number of biologically, and physically, well-motivated assumptions.I thoroughly enjoyed reading this paper, even though I am not an expert on the Golgi. I think it's a great example of theory at its best, applied to an interesting biological problem. The authors start with some simple observations that have been reported in the literature, which they turn them into a mathematical model that upon analysis yields some interesting and experimentally testable predictions. I am optimistic that their results will lead to some new experiments.The key result of the paper is the connection between the directionality of the vesicular transport and the chemical composition of the Golgi compartments, for which there seems to be experimental evidence. The connection is established within a very simple model whose assumptions seem reasonable. The paper also describes the emergence of a number of structural and dynamical properties of the Golgi, from the same small set of assumptions. A particularly interesting feature of the model is how different structures and dynamics can be accommodated within the model by changing one or more parameters (of which there are only four). For example, the model predicts that there is an inverse relationship between the speed of vesicle transport and fidelity of processing, which is consistent with published comparisons between the structure of Golgi in normal and starved yeast.The paper is very clearly written, and there were only a few places where I was not quite sure about what the authors had in mind.

We thank the reviewer for this positive comments. We are particularly pleased that the reviewer found our paper very clearly written considering the comments of reviewer 2. It is also comforting that he remarked on the small number of parameters involved in the model, which we hope will help temper the criticisms of reviewers 1 that some experimental observation are not directly explained by the model.

1) When describing the Budding mechanism, there is reference to a non-linear budding scheme that depends on the compartment size and contamination state. These are described in detail later, but I would recommend that the authors say a few words here to give the reader an intuitive sense of the thing without having to skip ahead.

The budding mechanism is described in more detailed in the main text of the revised version of the paper. More generally, the quantitative description the model, which was initially relegated to the Appendices, now also appears in the main text.

2) In describing maturation, the chemical transformations are described as going only one way. How is this justified? Is it known that the reverse reactions are very slow? Also, is the maturation rate independent of the composition of the compartment and if so, what is the justification for that?

We have discussed our model of maturation in our response to point 3) of reviewer 1 above. The motivation for the maturation scheme is the observation of irreversible maturation of Golgi cisternae in Yeast, which has been explain by the Rab cascade (Rivera-Molina and Novick, 2009). While there is no clear evidence that such mechanism is also at play in mammalian Golgi, most of the proteins involved in this mechanisms are conserved in mammals (Klute et al., 2011) and irreversible maturation (now called biochemical conversion) has been put forward as a possible organising principle there as well (Pfeffer, 2010). Our biochemical conversion scheme is designed to be generic, and could also represent the irreversible modification of lipid components, such as phosphoinositides, for which evidences exist in mammalian Golgi (McDermott and Mousley, 2016). We have expanded the paragraph justifying and describing our model of biochemical conversion in the revised version of the paper.

Whether the conversion rate could be cooperative and depend on the local composition is a very interesting question that we plan to examine in the follow-up paper. We have assumed a constant conversion rate, independent of the local composition. This is likely to be a simplification, as conversion could be a cooperative phenomenon. In a previous paper (Vagne and Sens, 2018a), we have shown that introducing cooperativity of biochemical conversion increases robustness. We choose not to include this possibility here because it requires the introduction of additional parameters. We discuss this (and cite this paper) in the revised version of the manuscript.

3) A typical compartment size is defined as the ratio of the second and the first moment of the size distribution. I was struck by the fact that this would give a "typical size" even for an exponential distribution of sizes, which I would not say defines a typical size (at least not from the point of view that the mean and standard deviation of such a distribution are equal). Might be worth a short comment on why this is a good definition of "typical size".

With our definition of the typical size as the second moment over the mean, there is indeed a typical size for an exponential distribution, which is the characteristic size over which this distribution decays. This is explained in Appendix 2 Eqs 5-7. This does not mean that most compartments will be of that size, but rather that it is unlikely to find compartments with a size larger that this typical size. The original version of the manuscript was imprecise in that respect, and has been corrected. In our case, the distribution is close to a power-law, with an exponential cutoff. As shown in by the red bars in Figure 2A, and mathematically in the Appendix 2, our way to define the typical size appropriately quantify the decay of the size distribution.

4) In the "Steady-state organization" section there is reference to a cut-off size, which I don't think was defined.

We have rephrased this. The cut-off size is intended to describe the size beyond which the distribution decays exponentially, which is also the characteristic size as defined by the ratio of second to first moment.

5) In the Discussion there is a conclusion that the spatial information is less crucial than biochemical composition for the organization of the Golgi. I am not sure this is warranted since a comparison between the two cannot be made in a model lacking spatial information.

This comment was intended to reflect on experimental observations, not on the model. This was indeed unclear and has been rephrased in the revised version. Data suggest that the transport dynamics is not strongly affected by major modifications of the spatial structure of the Golgi. Indeed, transport still occurs (although at a slightly lower rates) when cisternae are locked to mitochondria, more dispersed than in the usual Golgi stack, and unable to fuse with one another (Dunlop et al., 2017). This suggests that the spatial structure may not be crucial to the Golgi dynamics, and justify the relevance of our self-organisation model lacking spatial information. We have rephrased the text in the revised version.

In general I was left wondering whether simple testable quantitative predictions can be made from the model. I very much liked the qualitative observations that the model neatly explains, but I would be even more excited if there was a prospect for putting it to a much more stringent test provided by quantitative experiments, which the authors might propose for an intrepid experimentalist to do.

We thank the reviewer for this comment, which prompted us to think deeper about falsifiable tests of our theory by experimentally exploring the two-dimensional phase diagrams we provide. Our main prediction regarding the Golgi structure is that compartment size and purity are linked and should be affected in a correlated fashion by physiological perturbations (as shown in the phase diagrams, Figure 2 BC of the main text). Importantly, these 2D phase diagrams can in principle be fully explored by independently varying the ratios of maturation over fusion rate *k*_m_ and of budding over fusion rate *k*_b_. We present such possibility in Appendix 9 of the revised version of the manuscript.

Compartment purity can be altered without changing their size by acting on *k*_m_, while it should be correlated with a change of size by acting on *k*_b_. Experimentally, it was found that impairing COP-I activity (decreasing *k*_b_) decreases purity and increases size (Papanikou et al., 2015), as we predict (red arrow in Appendix 9—figure 1). We can predict further that this loss of purity phenotype could be reversed upon decreasing the maturation rate *k*_m_ (*e.g.* by impairing Ypt1 activity – as done in (Kim et al., 2016)) and that this would not change cisternae size (dashed red arrow in Appendix 9—figure 1). It was also found that Ypt1 over-expression (which increases the early to transitional maturation rate) decreases purity as well (Kim et al., 2016). We predict that this should not be associated to change of cisternae size (at steady-state) if altering Ypt1 does not modify the budding rate *k*_b_ (green arrow in Appendix 9—figure 1). We can predict further that the loss of purity phenotype could be reversed upon increasing the budding rate by over-expressing COP-I, but that this would be accompanied by an decrease of cisternae size (dashed green arrow in Appendix 9—figure 1). We will add this proposal to the revised version of the manuscript. Of course, the arrows drawn in Appendix 9—figure 1 are somewhat arbitrary, but we hope that this proposal gives a good feel for the way our model can be used to make new and non-trivial predictions.

[Editors’ note: what follows is the authors’ response to the second round of review.]

Revisions:1) Regarding prior work, the reviewers note that as the work of Patterson et al., 2008, provided a detailed quantitative model of the Golgi using one set of parameters, and furthermore assessed and tested those parameters by doing extensive quantitative live cell imaging experiments, it would be appropriate to seek a comparison between your model (with its dimensionless parameters) and those results. This would likely enlarge the scope of potential predictions for direct experimental tests. Workers in the Golgi field would benefit from some suggested experiments expected to skew the pathway in one direction or another, with predicted outcomes on the size of the compartment or the speed of traversal etc. For example, the cisternal progression model predicted that larger cargo export rates would be more sensitive to nocodazole dissolution of microtubules and resulting creation of ministacks, a prediction that turned out to be correct: PMID 25103235.

Regarding experimental suggestions, we already provide fairly detailed suggestions of experimental approaches to test some of our predictions, which are discussed in the Discussion section and in Appendix 9—figure 1. These experiments could test the correlation between compartment size and purity which we predict. The present paper focusses on Golgi self-organisation and does not investigate the different transport kinetics for different cargo. Such discussion requires additional assumptions regarding the coupling between the cargo and the local cisterna environment, and will be the subject of a subsequent paper (See also our response to Question 4). We strongly feel that adding results on cargo dynamics to the present paper could obfuscate our main message.

Regarding previous works, and in particular the rapid partitioning model (RPM). We would like to stress that while the RPM and our model share some similarities, they are fundamentally different in their scope. Our model describes de-novo Golgi formation and maintenance, through self-organized fusion and scission mechanisms between dynamic compartments, while the RPM focuses on the transport of lipids and cargo molecules in a pre-established Golgi apparatus with a prescribed structure. While we assume that cisternae identity results from a balance between exchange and irreversible biochemical maturation, the RPM assumes that lipid composition (which can be a proxy for cisternal identity) is tuned by lipid transport mechanisms.

In spite of these fundamental differences, the two models converge on key aspects of cargo transport and export mechanisms. In both the RPM and our model, export is allowed from every cisterna, which is crucial to reproduce the exponential kinetics observed in experiments. In addition, no directionality is assumed a priori for intra-cisternal transport in the RPM, which is in accordance with the budding/fusion mechanism in our model. Finally, in the RPM, the transport rate towards other organelles is different from the intra-Golgi transport rate. A similar effect is obtained in our model, through the parameters *α*_ER_ and *α*_TGN_ that tune the fusion rate with entry or exit compartments.

The RPM is much more precise than our model, in terms of the microscopic description of the biochemical processes. While we study here the effect of only 4 coarse-grained parameters, there are about 50 different parameters in the full version of the RPM. This makes further comparison difficult, but it shows how our simple theoretical model could evolve to become more realistic. Adding Golgi enzymes, cargo molecules, or lipids to our model, and describing their synthesis and/or sorting into membrane domains, as is done in the RPM, is the logical next step that will push forward our understanding of Golgi self-organisation.

A paragraph summarising the discussion above has been added to the Discussion section.

2) To help the general reader understand the context of this work better, we would like to see an introductory figure illustrating the structure of the Golgi (and perhaps some of the models for transport in the Golgi).

We followed the suggestion and we have extended Figure 1 that now presents an illustration of a classical representation of the Golgi structure, together with four sketches some of the main existing transport models.

3) The reviewers suggest that putting more of the model equations in the main text would streamline the argument; one way of making things more understandable might be to start by writing down simple mean-field equations (as in the early parts of Appendix 3), before delving into the analysis of the detailed model. The back-and-forth between the main text and the appendices makes for difficult reading.

The mean field model for the composition of the system in the well-sorted limit, and for the size and composition of compartments in the well-mixed limit are now presented at the beginning of the Results section.

4) An important concern is that the model shows how Golgi structure and vesicular transport can arise in a self-organised fashion, but it remains unclear to what extent the authors can claim that vesicular transport in actual Golgi is dominated by this transport arising from self-organisation. When reading the paper, we wondered about the contribution of spatial structure or biochemical processes (that might bias which vesicles cargo goes into, or which compartments a vesicle fuses into). It was only pages later, in the Discussion, that one finds a paragraph on the role of spatial structure: even if experiments show that "Golgi functionality is preserved under major perturbation of its spatial structure", how is transport affected? From a quick look at the reference (Dunlop et al., 2017), it seems that transport can be slowed down massively, hinting that spatial structure is just as important as self-organisation. Are there similar experimental results on biochemical effects?

There are two distinct questions above, one regarding the spatial structure, and one regarding transport biased through biochemical effect.

Regarding spatial structure, the fact that our model does not include space was not clearly discussed in the Introduction. This has been changed in the revised version of the manuscript. We have also made the statement in the Discussion section more precise. What we meant to say was that a Golgi with land-locked cisternae is still able to transport and process cargo. To quote Dunlop et al., 2017: “It is amazing that cargo processing and transport continue with normal efficiency and are merely slowed down when the normal topological relationships among Golgi cisternae are grossly altered […].” Our original statement lacked precision, and we now explicitly write that transport is only slowed down by a factor 2 (from 20 min; to 40 min. on average, see Table 3 in Dunlop et al., 2017), despite major spatial perturbation. In our opinion, this remarkable observation shows fairly convincingly that spatial proximity is not required for functional intra-Golgi transport, although it does help increasing its rate. We have modified the discussion of Dunlop et al., 2017 in the Discussion section.

Regarding biochemical bias of intra-Golgi transport, it is certainly included in the model as a key element for self-organisation through homotypic fusion of vesicles with compartment. The way we understand this question is whether Cargo transport through the Golgi is bound to follow the average vesicular transport which result from self-organisation. This is a very interesting question, which, as we said above, will be the subject of a follow-up paper. We have nevertheless added the following comment to the text: Note that vesicular transport of cargo through the Golgi is not bound to follow the net vesicular flux between compartments discussed above. Indeed, the net flux is the average of the flux of vesicles of the three identities. A given cargo follows this flux – on average – if it does not interact preferentially with membrane of particular identity. On the other hand, a cargo that is preferentially packaged into vesicles of *trans* identity (for example) will be transported toward *trans*-rich compartments even if the net vesicular flux is mostly retrograde.

5) One of the main results in the paper is the existence of different regimes of self-organisation, and in particular the existence of an intermediate, sorted regime with large pure compartments. However, the definition of these regimes remains rather qualitative. Perhaps some plots of mean compartment size against mean purity, to complement Figure 2, would put this part of the analysis on a clearer footing.

It is correct that the transition between the vesicular and mixed regimes is rather smooth, so that the definition of an intermediate, sorted regime is somewhat arbitrary. We now show a plot of the mean compartment purity vs. the mean size in Figure 2 E. This shows a clear inflexion point which corresponds to what we call the sorted regime.

6) The model shows very clear anterograde transport of cargo in the limit k_b_>>1, but the retrograde transport in the limit k_b_<<1 is less clear; in particular, the model appears to show transport from the trans to well-mixed compartments, but not into the cis compartments: rather, there is weak flow from the cis compartments to well-mixed ones, i.e. anterograde flow in part of the system. Is it possible to show that flow cannot be purely retrograde? – This might be an important constraint on the "cisternal maturation" mechanism. (Perhaps part of this issue is addressed in Appendix 6.)

It is correct that the net retrograde flux in the limit <inline-graphic mime-subtype="png" mimetype="image" xlink:href="media/image1.png" />1 in our model does not correspond to the stereotypical *trans* → *medial* → *cis* flux, but is dominated by fluxes “from trans-rich toward less mature compartments” as indicated in the original manuscript. The absence of net *medial* → *cis* flux in this limit can be understood by analysing the fluxes of vesicles of different identities, shown in Appendix 6. The compartments that contain the highest fraction of *medial* identity in this regime are well mixed (*φ_cis_*' *φ_medial_*' *φ_trans_*' 1*/*3). The vesicular flux leaving such compartment is equally split into *cis*, *medial* and *trans* vesicles, which fuse homotypically with compartments enriched in their identity, yielding a vanishing net flux averaged over the three identities. On the other hand, vesicles emitted from early compartments will either undergo back fusion with early compartments or anterograde fusion with more mixed compartments. Since back-fusion is not included in the vector plot of Figure 4, the vesicular flux from *cis* compartment is necessarily anterograde. In our model, flow is never either purely retrograde or purely anterograde. The flow of *cis* vesicles is always retrograde (except if a vesicle undergoes maturation before fusion) and the flux of *trans* vesicles is always anterograde, as can be seen in Appendix 6—figure 1. What matters really is the intensity of this flux, which, as the reviewer points out, is very small (essentially negligible relatively to other fluxes) in the low budding regime. This directly translate into a clear retrograde enrichment in Figure 3B.

In the classical “cisternal maturation” mechanism, retrograde transport is needed to recycle *cis* Golgi components. Within our model, this can in principle be achieved by targeting these components to *cis* vesicles budding from more mature compartments.

These points are now made clear on the new sketches of vesicular transport in the revised version of Figure 3 and are now discussed in the text.

**References**

Bhave, M., Papanikou, E., Iyer, P., Pandya, K., Jain, B. K., Ganguly, A., Sharma, C., Pawar, K., Austin, J., Day, K. J., Rossanese, O. W., Glick, B. S., and Bhattacharyya, D. (2014). Golgi enlargement in Arf-depleted yeast cells is due to altered dynamics of cisternal maturation. *Journal of Cell Science*, 127(1):250–257.

Der´enyi, I., Ju¨licher, F., and Prost, J. (2002). Formation and interaction of membrane tubes. *Phys. Rev. Let.*, 88:238101.

Dunlop, M. H., Ernst, A. M., Schroeder, L. K., Toomre, D. K., Lavieu, G., and Rothman, J. E. (2017). Land-locked mammalian Golgi reveals cargo transport between stable cisternae. *Nature Communications*, 8(1):432.

Evans, E. and Rawicz, W. (1990). Entropy-driven tension and bending elasticity in condensed-fluid membranes. *Phys. Rev. Let.*, 64(17):2094–2097.

Glick, B. S. and Luini, A. (2011). Models for golgi traffic: A critical assessment. *Cold Spring Harb Perspect Biol*, 3:a005215.

Klute, M. J., Melan¸con, P., and Dacks, J. B. (2011). Evolution and diversity of the Golgi. *Cold Spring Harbor Perspectives in Biology*, 3(8):a007849.

Ladinsky, M., Wu, C., McIntosh, S., McIntosh, R., and Howell, K. (2002). Structure of the golgi and distribution of reporter molecules at 20^◦^c reveals the complexity of the exit compartments. *Mol. Biol. Cell.*, 13:2810–2825.

Levi, S. K., Bhattacharyya, D., Strack, R. L., Austin, J. R., and Glick, B. S. (2010). The yeast grasp grh1 colocalizes with copii and is dispensable for organizing the secretory pathway. *Traffic*, 11:1168–1179.

Papanikou, E., Day, K. J., Austin, J., and Glick, B. S. (2015). COPI selectively drives maturation of the early Golgi. *eLife*, 4.

Rivera-Molina, F. and Novick, P. (2009). A rab gap cascade defines the boundary between two rab gtpases on the secretory pathway. *P Natl Acad Sci Usa*, 106:14408–14413.

Saenz, J., Sun, W., Chang, J. W., Li, J., Bursulaya, B., Gray, N., and Haslam, D. (2009). Golgicide A reveals essential roles for GBF1 in golgi assembly and function. *Nat Chem Biol*, 5:157–165.

Upadhyay, A. and Sheetz, M. (2004). Tension in tubulovesicular networks of golgi and endoplasmic reticulum membranes. *Biophys. J.*, 86:2923–2928.

Vagne, Q. and Sens, P. (2018b). Stochastic model of vesicular sorting in cellular organelles. *Phys. Rev. Lett.*, 120:058102.